# 3D organoid-derived human glomeruli for personalised podocyte disease modelling and drug screening

Lorna J. Hale[1,2], Sara E. Howden[1,2], Belinda Phipson [1], Andrew Lonsdale[1], Pei X. Er[1], Irene Ghobrial[1], Salman Hosawi[3], Sean Wilson [1], Kynan T. Lawlor[1], Shahnaz Khan[1], Alicia Oshlack [1,4], Catherine Quinlan[1,2,5], Rachel Lennon [3] & Melissa H. Little [1,2]

The podocytes within the glomeruli of the kidney maintain the filtration barrier by forming interdigitating foot processes with intervening slit diaphragms, disruption in which results in proteinuria. Studies into human podocytopathies to date have employed primary or immortalised podocyte cell lines cultured in 2D. Here we compare 3D human glomeruli sieved from induced pluripotent stem cell-derived kidney organoids with conditionally immortalised human podocyte cell lines, revealing improved podocyte-specific gene expression, maintenance in vitro of polarised protein localisation and an improved glomerular basement membrane matrisome compared to 2D cultures. Organoid-derived glomeruli retain marker expression in culture for 96 h, proving amenable to toxicity screening. In addition, 3D organoid glomeruli from a congenital nephrotic syndrome patient with compound hetero-zygous NPHS1 mutations reveal reduced protein levels of both NEPHRIN and PODOCIN. Hence, human iPSC-derived organoid glomeruli represent an accessible approach to the in vitro modelling of human podocytopathies and screening for podocyte toxicity.

[1] Murdoch Children's Research Institute, Flemington Rd, Melbourne 3052 VIC, Australia. [2] Department of Paediatrics and Department of Anatomy and Neuroscience, The University of Melbourne, Melbourne 3010 VIC, Australia. [3] Wellcome Trust Centre for Cell Matrix Research, University of Manchester, Manchester M13 9PT, UK. [4] School of BioScience, The University of Melbourne, Melbourne 3010 VIC, Australia. [5] Department of Nephrology, Royal Children's Hospital, Melbourne 3052 VIC, Australia. Correspondence and requests for materials should be addressed to M.H.L. (email: melissa.little@mcri.edu.au)

The human kidney regulates fluid homoeostasis, electrolyte balance, and waste product removal by filtering the blood via glomeruli, the specialised filtration unit within each nephron. The average human kidney contains one million nephrons[1], each including a glomerulus. Blood enters the glomerulus from an afferent arteriole and passes through a fenestrated endothelial capillary bed surrounded by specialised glomerular epithelial cells, the podocytes. Podocytes are post-mitotic cells with a highly specialised morphology[2]. They possess elaborate interdigitating cellular processes which are anchored to the glomerular basement membrane (GBM) via a network of integrins and dystroglycans. The major processes (primary and secondary) are supported by microtubules and vimentin intermediate filaments, while the smaller terminal foot processes contain actin filaments which form a complex contractile apparatus that helps to counteract the expansive forces of the underlying capillary [3]. Neighbouring foot processes are connected by specialised cell–cell junctions, known as slit diaphragms which, in conjunction with the GBM, form a two-step filtration barrier to soluble plasma protein components [4].

In order to maintain intact barrier function, the GBM consists of unique cellular and extracellular matrix (ECM) components[5], some provided by the podocytes and others by both the podocytes and the endothelial cells. Collagen IV and laminin isoform switches are known to occur during glomerulogenesis and maturation of the GBM[6]. Initially the GBM contains the α1α1α2 type IV collagen network, but then changes as the glomerular capillaries begin to form and the podocytes begin to secrete α3α4α5 trimers[7]. Laminin trimer deposition also occurs during development, transitioning from α1β1γ1 to α5β1γ1 and finally α5β2γ1. The timing of these isoform switches and when the individual protomers oligomerise and fuse into mature trimers is not well understood.

A number of kidney diseases leading to proteinuria and/or haematuria, including congenital nephrotic syndrome (CNS) and Alport syndrome, result from defects in the GBM, or functional and structural alterations to the podocyte that lead to foot process effacement and loss of slit diaphragms[8]. The clinical manifestation of podocytopathies and glomerulopathies relies upon the cellular identity of the component podocytes, and in some instances the formation of a genuine endothelial interaction capable of inducing a glomerular basement membrane. The genetic basis of many podocytopathies has now been elucidated[9]. These include mutations in genes encoding components of the podocyte actin cytoskeleton, slit diaphragm, and GBM. However, there are still many instances in which no apparent genetic aetiology is evident.

Understanding the basis of human podocytopathies was initially hampered by the limited proliferative nature and architecturally constrained morphology of primary podocytes[10]. The generation of a temperature-sensitive SV40 conditionally immortalised podocyte cell line, which allows proliferation at 33 °C and terminal differentiation at 37 °C in vitro[11], began to address this challenge. Studies using this two-dimensional podocyte model, primary human podocyte cultures and murine podocyte cell lines have substantially advanced our understanding of podocyte biology. However, the lack of three-dimensional context and the absence of interacting endothelial and mesangial cells potentially restricts the suitability of such podocyte models. Indeed, a recent study which compared in vivo glomerular datasets with proteomics data from podocyte cell cultures highlighted the limitations of the available immortalised podocyte cell culture models[12]. As it is difficult to acquire and culture primary patient-derived human podocytes, validation of novel disease-associated mutations is most often performed in animal models which may not always replicate the human condition.

The advent of induced pluripotent stem cell (iPSC) technology provides insights into a window of human development ex utero that has previously been difficult to access. It is now possible to induce the stepwise differentiation of human iPSC through identifiable embryological milestones to re-create multicellular tissues that represent accurate models of developing human organs, including optic cup, cerebral cortex, intestine and stomach[13]. We and others have described protocols for the directed differentiation of human iPSCs to kidney cell types[14–17]. Using our protocol, we have demonstrated the formation of kidney organoids that are comprised of renal epithelial, endothelial and interstitial cell types and show transcriptional congruence with human foetal kidney[17,18]. Preliminary analysis of the glomeruli within these kidney organoids showed tightly interdigitated podocytes while transcriptional profiling of whole organoids revealed the induction of podocyte gene expression, including *NPHS1, NPHS2* and *PODXL*[17], implicating this approach as a promising source of human podocytes.

In this study, we describe the isolation and comprehensive transcriptional and proteomic characterisation of organoid-derived glomeruli (OrgGloms) as well as primary podocytes cultured out of these glomeruli (OrgPods). This reveals that OrgGloms display a greater degree of maturation than isolated podocytes or immortalised lines. Hence, OrgGloms represent an accurate and reproducible three-dimensional model of podocytes within the human glomerulus. Importantly, we show that OrgGloms can be easily accessed from kidney organoids using conventional glomerular sieving techniques and hence can be readily generated in substantial numbers from any individual. With increasing time in organoid culture, OrgGloms show proteomic evidence of laminin and collagen switching suggestive of maturation. OrgGloms isolated from organoids generated using CNS patient iPSCs displayed disruption of podocyte polarisation and reduced levels of slit diaphragm proteins. As a single kidney organoid contains in the order of 100 glomeruli[17,18] and one iPSC differentiation can generate up to 30 organoids, the utility of OrgGloms for high content toxicity screening is also demonstrated. As such, OrgGloms represent a tractable 3D model of the human glomerulus in the normal and diseased state.

## Results

**Isolation of intact glomeruli from human iPSC kidney organoids**. Within human iPSC-derived kidney organoids, we observed the formation of podocyte clusters that were resistant to enzymatic dissociation. These arose between day 11 and day 18 of organoid development (d7 + 11, d7 + 18), as per our previous protocol[17]. We hypothesised that these clusters were glomeruli, and hence adapted a commonly used glomerular sieving technique previously applied to human postnatal kidney[5,19] in order to isolate them. Enzymatic dissociation of organoids at day 11 post-aggregation (d7 + 11) yielded a predominantly single cell population, while enzymatic dissociation of more mature d7 + 18 organoids yielded 3D aggregates of podocytes representing forming glomeruli (OrgGloms). These structures could be consistently isolated in large numbers (Fig. 1a, top left), yielding approximately 1500–3000 OrgGloms per differentiation (50–150 OrgGloms per organoid; 20–30 organoids per single well differentiation in a six-well plate). Close examination of OrgGlom structure (Fig. 1a, right) demonstrated a striking resemblance to whole glomeruli isolated from post-natal human tissue (Fig. 1a, bottom left). Average OrgGlom diameter was between that of glomeruli from late trimester three human foetal kidney and adult human glomeruli (Fig. 1b, see Source Data file). OrgGloms at d7 + 18 were comprised of an involuting podocyte layer surrounded by a Bowman's capsule (Fig. 1c) suggestive of the

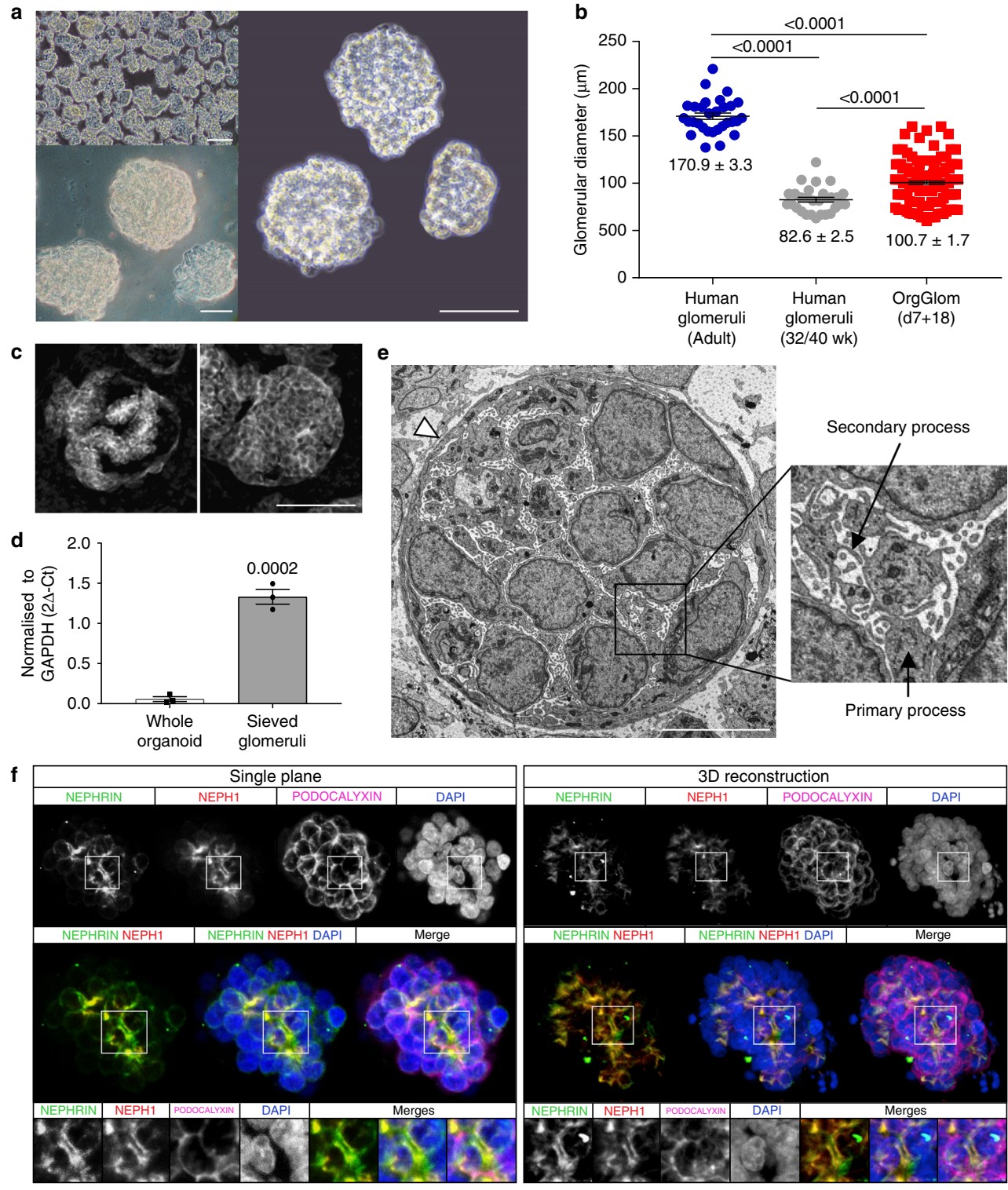

capillary loop stage of glomerular development[20], with enriched *NPHS1* gene expression in sieved glomeruli compared with whole organoids (Fig. 1d, see Source Data file). Transmission electron microscopy of OrgGloms in situ revealed podocyte cell bodies connected by primary and secondary processes and surrounded by an outer layer of parietal epithelial cells (Fig. 1e). This likely reduced enzymatic dissociation, facilitating the isolation of Org-Gloms by sieving. Immunofluorescence showed appropriate apicobasal polarity of the podocytes within the OrgGloms (Fig. 1f). PODOCALYXIN was shown to localise to the apical surface of the cells, with co-localisation of NEPHRIN and NEPH1 proteins at the intracellular junction between adjacent podocytes, as would be expected in the slit diaphragm[9] (Supplementary Video 1).

**Primary podocytes cultured from kidney organoid glomeruli.** Isolated OrgGloms placed into culture adhered within hours of plating, with migration of an organoid-derived podocyte cell population (OrgPods) from these glomeruli observed shortly

**Fig. 1** Intact glomeruli can be isolated from human iPSC-derived kidney organoids. **a** Top left: Pure populations of whole glomeruli can be isolated from kidney organoids at scale by sieving. Scale bar 200 μm. Right: These organoid-derived glomeruli (OrgGloms) when examined in detail (scale bar 100 μm) show similarity to adult human glomeruli (Bottom left, scale bar 100 μm). **b** OrgGloms are formed in the appropriate size range, comparable to both human glomeruli at 32 weeks gestation[70], and adult human glomeruli[62]. One-way ANOVA $p < 0.0001$; error bars = SEM; significant difference assessed by Tukey's multiple comparisons test; F-value = 176.9; DF = 2; biological replicates $n = 30$ adult glomeruli, $n = 30$ neonatal glomeruli, $n = 155$ OrgGlom. **c** Glomeruli developed within kidney organoids show comparable morphology to capillary loop stage human glomeruli, highlighted by in situ immunofluorescent staining of serial sections with the podocyte protein NEPHRIN, which shows infolding of the glomerular surface layer. Scale bar 10 μm, central cross-section (left), surface (right). **d** Quantitative PCR analysis of isolated glomerular fractions compared to whole organoid show a significant enrichment in *NPHS1* gene expression in sieved glomeruli. Error bar = SEM; significant difference assessed by Student's unpaired *t*-test $p = 0.0002$, *t*-value = 13, DF = 4, $n = 3$ biological replicates shown by symbols. **e** Transmission electron microscopy of kidney organoid glomeruli show podocyte cell bodies connected by primary and secondary processes (inset), surrounded by an outer layer of parietal epithelial cells (arrowhead). Scale bar 10 μm. **f** Immunostaining of whole OrgGloms shows appropriate apicobasal cell polarity. A single cross-sectional plane is shown on the left, and a 3D reconstruction using all Z-stack images acquired (444 sections) on the right. Expression levels of single channels shown in greyscale to preserve maximum contrast, merged images shown in colour. Full 3D reconstruction video of the Z-stack can be found as Supplementary Video 1

thereafter. By 18 h post-plating, distinct OrgPod populations were apparent surrounding isolated glomeruli. These cells possessed long, thin arborized projections (TAPs) (Fig. 2a) composed of F-actin (Fig. 2b) stretching between cells to initiate the formation of cell contacts. Over time, these cells continued to spread (Supplementary Fig. 1A), with immunofluorescent staining at 36 h post-plating revealing strong staining for SYNAPTOPODIN and PODOCALYXIN in OrgGloms surrounded by migrating Org-Pods (Fig. 2c). By 48 h post-plating, a diffuse OrgPod cell population was observed, displaying a large, flat, arborized morphology with processes connecting adjacent cells, reminiscent of podocyte processes in vivo (Fig. 2d, arrow) and comparable to previously reported fully differentiated conditionally immortalised human podocytes (ciPods) (Supplementary Fig. 1B). SYNAPTOPODIN, a key marker of differentiated, post-mitotic podocytes[11,21,22], is not observed in undifferentiated or de-differentiated podocytes[23,24] but has been shown to be expressed in ciPods after 14 days of induced differentiation. SYNAPTO-PODIN staining was also observed in OrgPods with appropriate localisation in conjunction with actin stress fibres (Fig. 2c, e). No de-differentiated podocytes, identified by a classical 'cobblestone' morphology[23,24], were observed in these OrgPod cultures. Org-Pods also appropriately expressed a number of classical podocyte proteins, including CD2-ASSOCIATED PROTEIN (CD2AP), WILMS' TUMOUR PROTEIN (WT1), PODOCIN and P-CADHERIN (Supplementary Fig. 1C)[23]. To investigate the capacity of OrgPods to endocytose albumin[25], we first confirmed the expression of the neonatal Fc receptor (FcRN). FcRN, which facilitates both IgG and albumin transport in podocytes[26,27], was present in a punctate distribution across the OrgPod cell surface (Fig. 2f, top panel). Live OrgPods actively endocytosed fluorescein isothiocyanate (FITC)-labelled albumin, resulting in discrete areas of FITC accumulation on the cell surface when in culture at 37 °C (Fig. 2f, central panel). This active process was halted when performed at 4 °C (Fig. 2f, lower panel). The facilitative glucose transporter, GLUT4, is known to translocate to the cell surface of podocytes to allow glucose uptake following exposure to insulin[28]. Insulin also directly remodels the actin cytoskeleton of the podocyte[29]. The cellular localisation of GLUT4 was found to be vesicular cytoplasmic in untreated OrgPods. After exposure to insulin for 10 min, GLUT4 translocated to the plasma membrane. This was accompanied by reorganisation of the actin cytoskeleton from a filamentous to a strongly cortical distribution (Fig. 2g). Together, these data demonstrate podocyte functional properties within OrgPod cultures.

**Improved podocyte identity in 3D isolated glomeruli.** To compare the validity of our 3D OrgGlom model compared to OrgPods and ciPods, transcriptional profiling was performed using RNA sequencing (RNA-seq)[30]. A principle component analysis (PCA) revealed that biological replicates of each cell type clustered tightly (Fig. 3a). A differential expression analysis was performed comparing all sample groups (false discovery rate <5% and absolute log2-fold-change >1, see Source Data file) with the greatest transcriptional variation seen when comparing 3D Org-Gloms with differentiated ciPods (Fig. 3b). An unbiased heatmap of the top 50 differentially expressed genes showed distinct differences between all 2D cultures and 3D OrgGloms, with many of the significantly upregulated genes in OrgGloms associated with the podocyte (Fig. 3c). Gene Ontology (GO) terms associated with slit diaphragm components, renal filtration cell differentiation and glomerular development were the most significantly enriched in OrgGloms compared with differentiated ciPods (Top 100 upregulated genes; Fig. 3d). Surprisingly, GO terms most significantly enriched between differentiated and undifferentiated ciPods included cell adhesion, regulation of signal transduction and apoptotic activity (Supplementary Table 1; Supplementary Fig. 2A), suggesting little evidence for podocyte maturation after temperature switching in this line.

A heatmap showing expression levels of key podocyte-associated genes demonstrated a clear distinction between the 3D glomeruli and the 2D cell models (Fig. 3e). This was validated using qPCR, which also showed enhanced gene expression in OrgGloms compared to the 2D models (Fig. 3f, see Source Data file). Again, ciPods showed significantly lower podocyte gene expression levels (two-way ANOVA $p < 0.0001$; with Tukey's multiple comparisons test) in both undifferentiated and fully differentiated states, in both our dataset and previously published data[30]. While OrgPods showed slightly improved expression levels in comparison to the immortalised line, they were also suboptimal compared to the OrgGloms. To confirm that the OrgGlom model was reproducible, we examined OrgGloms isolated from three independent iPSC lines. When comparing expression levels of podocyte-associated genes we found extremely tight congruence between all three clones (Supplementary Fig. 2B, see Source Data file). To relate our expression data back to human kidney, we defined a panel of 100 human glomerular-enriched genes by analysing publicly available adult glomerular gene expression data[31] (GSE21785). 'Glomerular expression scores' were determined for each sample, with OrgGloms found to be the most congruent with human glomeruli (Fig. 3g, see Source Data file).

Several previous studies reporting the generation of kidney organoids using iPSC described the formation of glomeruli[32,33]. Sharmin et al.[32] generated kidney structures used a NPHS1-GFP reporter line, which enabled enrichment for podocytes; however, this microarray-based data was not amendable to direct comparison with the current data. Kim et al.[33] transcriptionally

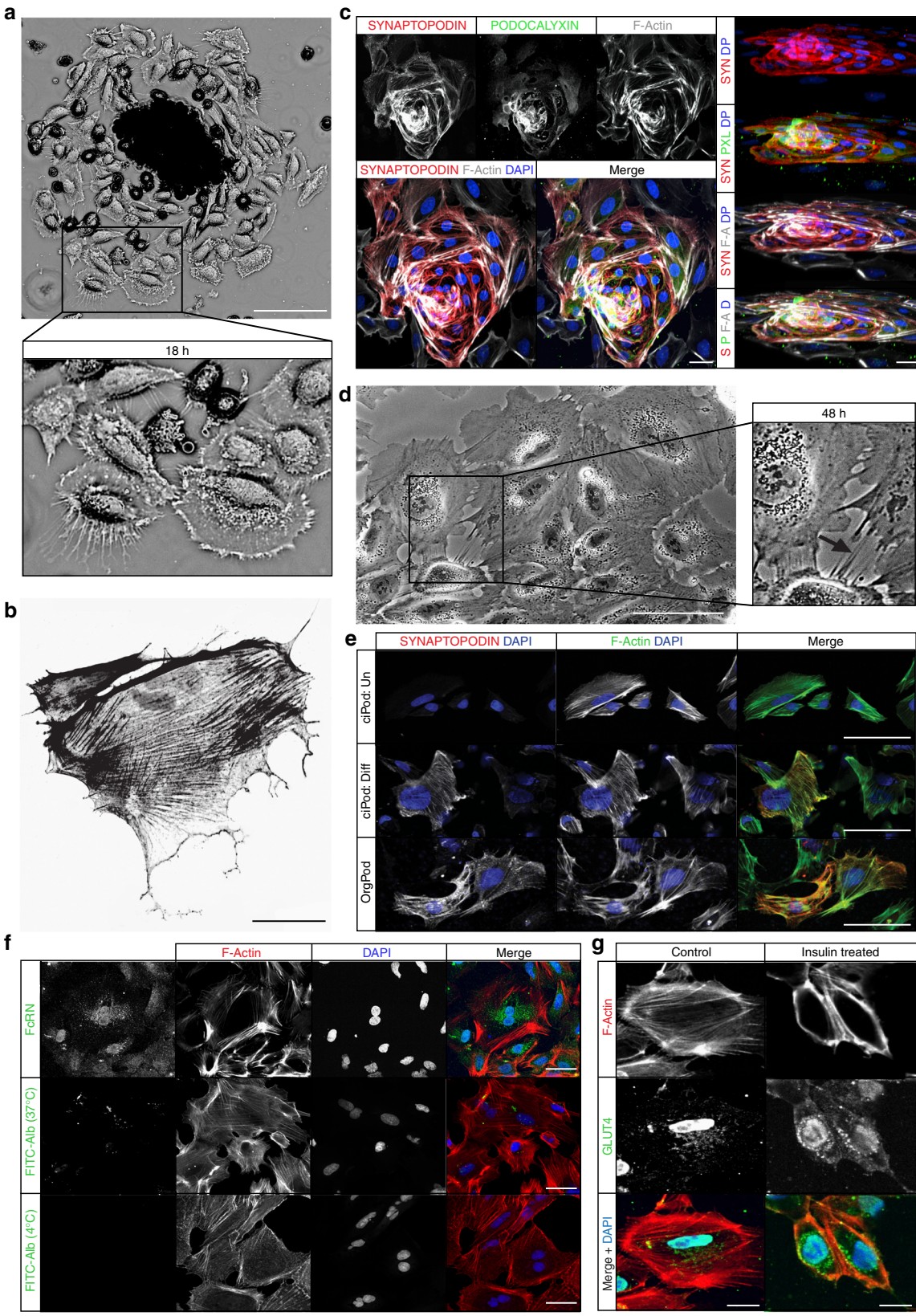

profiled undifferentiated iPSC lines (Supplementary Fig. 3A). While this may be feasible to characterise genes such as *PODXL*, which are expressed in undifferentiated iPSC (Supplementary Fig. 3B), this may not be a useful approach to characterising the role of genes only expressed in podocytes themselves. In conclusion, OrgGloms display improved transcriptional profiles and cellular identity to that of previously reported primary or immortalised podocyte lines, with profiling of OrgGloms, suggesting a clear benefit to 3D culture for the maintenance of podocyte identity in vitro.

**Fig. 2** Primary podocytes can be cultured from kidney organoid glomeruli. **a** Isolated organoid glomeruli show evidence of podocyte cell migration (OrgPods) displaying thin arborized projections (TAPs) (inset). Inverted image shown to provide maximum contrast, scale bar 100 μm. **b** TAPS from newly emerged podocytes are composed of F-actin shown by phallodin immunofluorescent staining. Inverted image, scale bar 50 μm. **c** Immunostaining at 36 h post-plating shows a strong positively stained 3D OrgGlom with a migrating 2D OrgPod population. Left panel 2D images, right panel 3D reconstruction of Z-stack. Scale bars 50 μm. **d** At 48 h post-plating OrgPods display a flattened, arborized morphology with processes connecting adjacent cells (arrow), scale bar 50 μm. **e** Immunostaining of ciPods for SYNAPTOPODIN showed expression is absent in undifferentiated cells (ciPod: Un), only becoming evident following 14 days induced differentiation at 37 °C (ciPod: Diff). OrgPods also display strong SYNAPTOPODIN protein expression, aligned with F-actin stress fibres. Scale bars 100 μm. **f** OrgPods express the neonatal Fc receptor (FcRN) and actively endocytose fluorescein isothiocyanate (FITC)-labelled albumin at 37 °C resulting in FITC-accumulation in endosomes on the cell surface. This is process halted when performed at 4 °C. Scale bars 50 μm. **g** OrgPods stimulated with insulin (10 mg/ml) for 10 min showed cortical reorganisation of their actin cytoskeleton with GLUT4 translocation from a vesicular to plasma membrane localisation. Scale bars 50 μm. All representative images reflect a minimum of three biological replicates. For immunofluorescence, images are shown in greyscale for single channels, and merged images in colour

**Collagen and laminin switching in organoid glomeruli.** The specific cues which initiate switches between immature and mature laminin and collagen IV isoforms during GBM maturation are unknown. While the podocytes are known to produce the appropriate collagen IV isoforms, crosstalk between podocytes and adjacent cells may provide cues for synthesis, secretion and ultimate assembly of matrix proteins[34]. The extracellular matrix composition of kidney organoids has not yet been examined. To investigate this further, mass spectrometry-based proteomic analysis of the isolated extracellular matrix (ECM) and cell lysate of OrgGloms was performed and compared to the matrices derived from organoid proximal tubular cells (OrgPT). Using our previously published method[5], serial chemical fractionation of Org-Gloms and OrgPTs was performed to derive fractions enriched for cellular and ECM components (Fig. 4a). Fractions C1 and C2 were predominantly cellular proteins, C3 nuclear proteins and C4 enriched for ECM proteins. A PCA demonstrated good separation between cellular and ECM fractions. There was close overlap of C1 fractions from both cell types, while C4 fractions showed close congruence in principle component 1, with greater differences seen in principle component 2 (Fig. 4b). Mapping of organoid proteomic data onto the human matrisome database (http://matrisomeproject.mit.edu/) identified 60 enriched matrix proteins, 30 found within the predominantly ECM-rich fraction (C4), 20 found in the predominantly cellular protein fraction (C1) and 10 matrix proteins common to both (Fig. 4c; Supplementary Tables 2 and 3). Using label-free quantification with ion intensities, the expression profile of these 60 matrix proteins was plotted and grouped by gene ontology (GO) classification. Core matrisome proteins were found to be primarily within the C4 fraction (Fig. 4d; Supplementary Tables 2 and 3; see Source Data file), while matrisome-associated proteins were predominantly within the cellular C1 fraction, which captures more soluble proteins (Fig. 4e; Supplementary Tables 2 and 3). Enrichment of mature GBM components was evident in OrgGloms. In particular, the laminin chains α5, β2 and γ1 were highly abundant, confirmed by immunostaining of laminin subunit alpha-5 (LAMA5) protein in OrgGloms (Fig. 4f). The presence of immature laminins was also observed in OrgGloms, which is unsurprising given the close similarity of kidney organoids to human foetal tissue at the gene level[17]. Type IV collagen chains α1 and α2 were also abundant and indicative of basement membrane formation in OrgGloms. The mature type IV collagen α5 and α6 chains were also highly abundant, these are expressed in the Bowman's capsule as the α5α5α6 network[35]; however, α3 and α4 chains were at the limit of detection, suggesting that alternative cues are required for assembly of the α3α4α5 network (Fig. 4d; Supplementary Tables 2 and 3). In addition, a number of key ECM-associated glycoproteins and proteoglycans were produced in OrgGloms which are essential for kidney development. The matrix components Fraser extracellular matrix subunit 1 (FRAS1) and Fraser extracellular

matrix subunit 2 (FREM2) known to have distinct roles in kidney development[36] were found to be significantly enriched in the OrgGlom C4 fraction. Also enriched in OrgGloms were hemicentin-2 (HMCN2), a key protein required for connecting adjacent basement membranes[37], and nidogen-1/2 (NID1, NID2) which bind collagen IV and laminin networks and are also subsequently important in membrane fusion[6]. In addition, agrin and heparan sulphate proteoglycan 2, two of the most abundant heparan sulphate proteoglycans in the GBM[6], were found to be highly expressed in OrgGloms (Fig. 4e). To achieve a wider perspective, the OrgGlom matrix data were compared to previously published proteomic data from human glomerular tissue[5] alongside the matrices of the conditionally immortalised human podocytes (ciPod)[11] and conditionally immortalised glomerular endothelial cells (ciGEnC)[38].

This comparison highlighted a number of glomerular proteins present within OrgGloms which are absent in immortalised cultures, including the mature type IV collagen α6. In summary, OrgGloms show greater congruence to human glomerular tissue by comparison to monoculture-derived matrix (ciPod or ciGEnC) or coculture-derived matrix (ciPod-ciGEnC).

**A reporter iPSC line profiles organoid glomerular differentiation.** The MAF bZIP transcription factor B gene (*MAFB*) is highly expressed in developing podocytes[39,40]. In order to monitor podocyte development within an organoid context, we generated a knock-in iPSC line that harbours the mTagBFP2 fluorescent reporter gene inserted at the start codon of the endogenous *MAFB* locus (MAFB^mTagBFP2/+) (Supplementary Fig. 4A and B)[40]. Kidney organoids were generated from MAFB^mTagBFP2/+ iPSCs, allowing podocyte development to be imaged in real time (Fig. 5a). Isolation of the mTagBFP2+ cells from dissociated organoids at day 10 post-aggregation (d7 + 10) by fluorescence-activated cell sorting (FACS) (Fig. 5b) followed by RT-PCR analysis confirmed *MAFB* expression was restricted to the mTagBFP2+ fraction and absent from mTagBFP2 non-expressing cells (Fig. 5c). Co-localisation of mTagBFP2 and NEPHRIN was also observed in whole organoids analysed by immunostaining at d7 + 18 (Supplementary Fig. 4C). Isolation of d7 + 18 OrgGloms from live organoids by sieving yielded an enriched mTagBFP2+ cell population (Fig. 5d, 0 h). However, 2D culture of OrgPods initially derived from mTagBFP2+ glomeruli demonstrated rapid reduction of reporter gene expression within 24 h (Fig. 5d, 24 h). In contrast, when sieved mTagBFP + Org-Gloms were cultured in suspension in a 3D format, strong reporter gene expression could be maintained for up to 96 h post-isolation (Fig. 5d, arrow) (Supplementary Fig. 4D). This observation strongly supports our prior conclusion that OrgGloms maintain a more accurate podocyte identity than the component cell types grown in 2D culture.

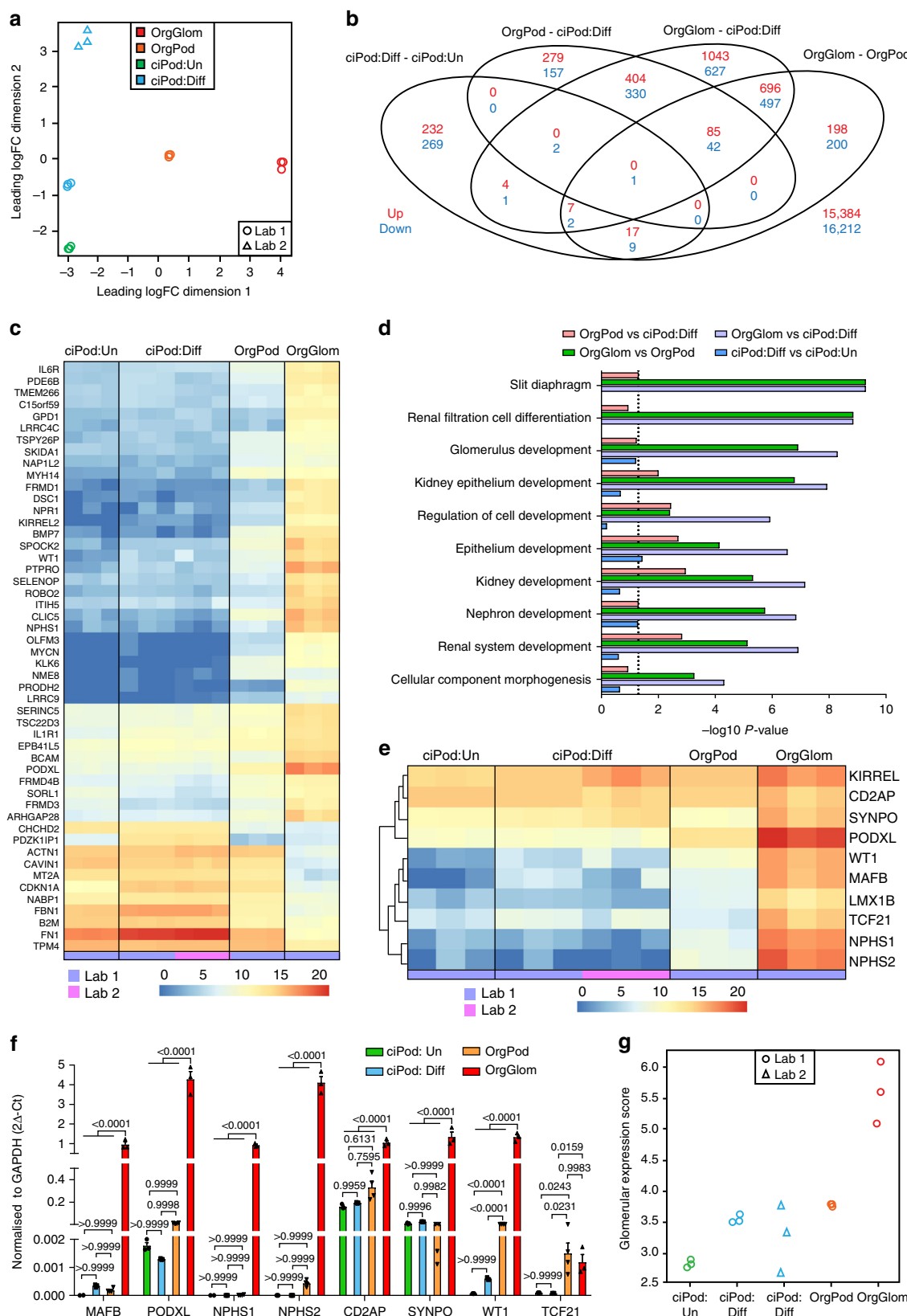

While the regulation of podocyte development has been studied in mice[41,42], in humans our understanding remains limited[31]. Through differentiation of MAFB$^{mTagBFP2/+}$ iPSCs to kidney organoids we were able to isolate MAFB-expressing cells at various developmental stages by virtue of the fluorescent reporter. We examined the transcriptional profile of MAFB+ cells after MAFB expression was first observed (d7 + 10), midway through organoid development (d7 + 14) and late in organoid development (d7 + 19). MAFB expression was first seen in the proximal end of the forming nephrons in organoids,

**Fig. 3** Organoid-derived glomeruli display improved podocyte identity. **a** Principle component analysis of RNA sequencing (RNA-Seq) data was performed on three biological replicates of OrgGloms, OrgPods and ciPods in both differentiated and undifferentiated states and compared to previously published data (Lab 2)[30]. **b** Venn diagram displaying the intersections of each comparison, upregulated genes are shown in red and downregulated genes are shown in blue. This shows the greatest number statistically significant upregulated genes were identified in the OrgGloms versus differentiated ciPods. There was little to no overlap of differentially expressed genes in differentiated ciPods versus undifferentiated ciPods compared with all other pairwise comparisons. **c** Heatmap showing the top 50 differentially expressed genes of each sample triplicate. A stark contrast in expression levels between the 3D OrgGloms and the 2D podocyte cultures is observed, particularly in genes associated with the podocyte. Log-normalised gene expression levels depicted. **d** Gene Ontology (GO) enrichment analysis of the top 100 upregulated genes differentially expressed between OrgGloms and differentiated ciPods found enrichment of GO terms associated with developmental processes, and slit diaphragm components including genes associated with podocyte foot processes and those of the podocyte actin cytoskeleton. The top 10 most statistically significant GO term categories are shown. $P$-value of 0.05 depicted as dotted line. **e** Heatmap representation of key podocyte-associated genes showed low expression levels in both undifferentiated and differentiated ciPods, regardless of the laboratory in which they were cultured. By contrast, significantly elevated gene expression levels were displayed in the OrgGlom samples. Log-normalised gene expression levels depicted. **f** Quantitative PCR analysis of key podocyte-associated genes validated the RNA-seq data showing enhanced gene expression in OrgGloms compared to 2D podocyte models. Two-way ANOVA $p < 0.0001$; error bars = SEM; significant difference was assessed by Tukey's multiple comparisons test; $n = 3$ biological replicates shown by symbols; F-value (Interaction) = 58.22, DF (Interaction) = 21; F-value (Gene) = 56.82, DF (Gene) = 7; F-value (Cell Type) = 544.4, DF (Cell Type) = 3. **g** A glomerular expression score was determined for each sample by calculating the average log expression across the top 100 upregulated genes identified from a human renal glomerulus-enriched gene expression dataset[31] (GSE21785). OrgGlom samples were found to have the highest scores for this gene set, showing greatest congruence to human glomerular isolates

as has been reported during nephron patterning in the developing human[43]. At this stage, the cells readily dissociate. However, as the glomerular structure matures, the glomeruli create a tightly packed ball of MAFB+ cells that can only be isolated by mechanical sieving. To capture all MAFB+ cells, isolation involved FACS sorting and glomerular sieving at all timepoints, with isolated MAFB+ cells combined for RNAseq profiling. A PCA was performed to assess the overall similarity between samples across time, with each triplicate found to cluster together closely (Fig. 5e). A differential expression analysis comparing all time points (false discovery rate <5% and absolute log2-fold-change >1, see Source Data file) showed few genes significantly up- or downregulated when comparing early to mid (d7 + 14 v d7 + 10) or mid to late time points (d7 + 19 v d7 + 14). The greatest difference was seen when comparing OrgGloms/BFP2+ podocytes (d7 + 19) to immature BFP2 + podocytes (d7 + 10) (Fig. 5f). Many of the 50 most upregulated genes showing increased expression across time were found to be members of the human glomerular ECM proteome (Fig. 5g). GO terms derived from the top 100 upregulated genes were associated with maturation of the ECM, cell adhesion and collagen trimer formation (Fig. 5h). When comparing these data to a panel of genes found to be enriched in adult human glomeruli[31], we found significant temporal upregulation in a number of non-podocyte glomerular genes (Fig. 5i), suggesting the accumulation with time of endothelial and mesangial cell types within the organoid glomeruli. This is supported by immunofluorescence of Org-Gloms for the endothelial marker PLATELET ENDOTHELIAL CELL ADHESION MOLECULE (PECAM1) and mesangial marker PLATELET-DERIVED GROWTH FACTOR RECEP-TOR BETA (PDGFRβ) (Supplementary Fig. 4E). In particular, KDR (VEGFR-2), which plays a critical and specific role in the maintenance and integrity of glomerular endothelial cells, showed a 24-fold upregulation between d7 + 10 and d7 + 19 (Fig. 5i; Supplementary Table 4). Endothelial cell markers (TEK, EMCN), mesangial markers (IGFBP5, TAGLN, MMP2), as well as critical GBM collagens (COL4A3, COL4A4) and genes essential for maintaining the renal vasculature (CXCL12, ANGPTL2) were all significantly upregulated with time (Fig. 5i; Supplementary Table 4). Hence, sieved OrgGloms contain increasing cellular complexity together with improved cellular maturity across organoid culture.

**Accurate modelling of congenital nephrotic syndrome (CNS) in vitro.** Congenital nephrotic syndrome (CNS) is characterised

by severe proteinuria evident in the first months of life[44]. Most cases of CNS result from mutations in genes encoding critical podocyte proteins, including NPHS1 (NEPHRIN) and NPHS2 (PODOCIN)[45]. To investigate whether OrgGloms could accurately model such a human podocytopathy, iPSC clones were derived from a patient with mutations in exon 10 (c.1235delG) and exon 27 (c.3481 + 4G > T) of the NPHS1 gene (compound heterozygote) (Fig. 6a; Supplementary Fig. 5). The exon 10 variant has not been previously reported and was predicted to create a frameshift, resulting in a premature stop codon with the mRNA produced likely targeted for nonsense mediated decay. The exon 27 mutation has been previously reported[46]. This single nucleotide substitution located in the donor splice site of intron 27 was predicted to result in exon 27 skipping. Both variants are likely pathogenic. Two iPSC clones were generated from this patient and differentiated into kidney organoids alongside a wild-type iPSC line of normal genotype (Supplementary Fig. 5, see Source Data file). Scanning electron microscopy of glomeruli within organoids from the CNS patient line showed evidence of large hypertrophied podocyte bodies, as has been described in NPHS1 mutant patients[47] (Supplementary Fig. 6A). OrgGloms were isolated for immunostaining and confocal Z-stack images were acquired through whole OrgGlom structures using matched parameters for image detection (Fig. 6b–d). OrgGloms differentiated from patient lines showed comparable protein expression levels (Supplementary Fig. 6B), with no significant difference between clones (Fig. 6f, see Source Data file). Differences in protein staining for NEPHRIN and PODOCIN were observed between both CNS clones and the control (Fig. 6e). Semi-quantitative analysis of fluorescence intensity within individual organoid glomeruli (Fig. 6f) revealed a significant reduction ($p < 0.0001$, two-way ANOVA with Sidak's multiple comparisons test) in both NEPHRIN and PODOCIN proteins in the patient Org-Gloms compared to control, whereas levels of CD2AP and NEPH1 were comparable. Reduced levels of NEPHRIN and PODOCIN proteins were quantitatively confirmed through western blot analysis of glomeruli from three independent biological replicates (Fig. 6g, see Source Data file).

**Use of 3D organoid glomeruli for toxicity screening.** The ability to reliably generate and isolate OrgGloms in large numbers from kidney organoids provides the opportunity for drug screening, both for toxicity and efficacy. As a proof of principle, individual MAFB-BFP2 OrgGloms (d7 + 18) were placed in a 96-well format and exposed to increasing

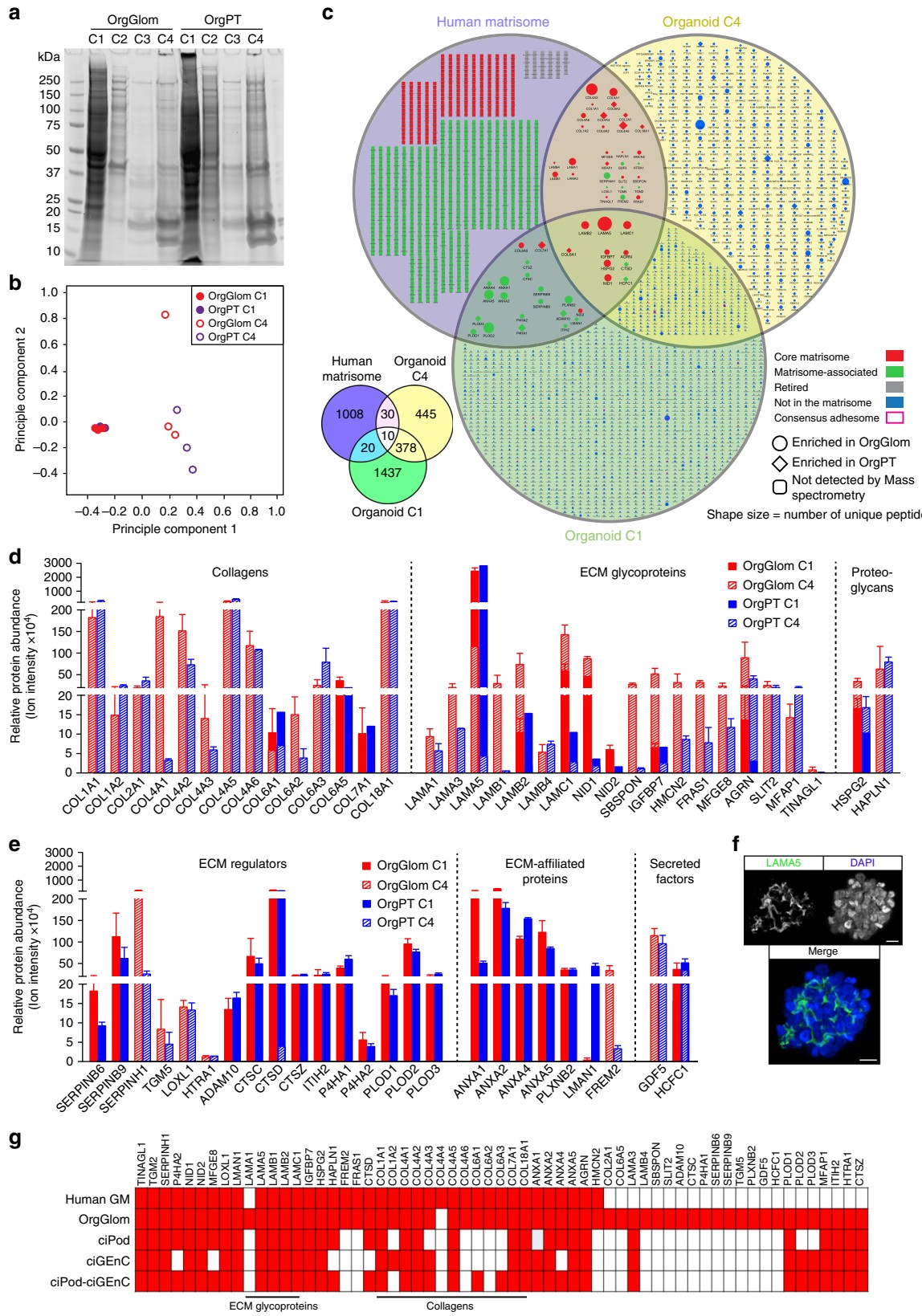

concentrations of doxorubicin. MAFB-BFP2 intensity was assessed via live imaging at 48 h post-treatment, revealing loss of BFP2 signal and fragmentation of glomeruli with increasing dose (Fig. 7a). Fixation of glomeruli followed by immunostaining for the apoptosis marker caspase-3 showed an increase in activity at the lower doses of doxorubicin, before cell death prevailed resulting in the destruction of glomeruli (Fig. 7b). MAFB-BFP2 intensity showed a dose-dependent decrease with a comparable reduction in glomerular size following doxorubicin treatment (Fig. 7c, see Source Data file).

**Fig. 4** Organoid glomeruli display a maturing GBM matrisome. **a** Serial chemical fractionation of organoid glomeruli (OrgGlom) and organoid proximal tubules (OrgPT) to derive fractions enriched for cellular and extracellular matrix (ECM) components, where C1 and C2 are predominantly cellular proteins, C3 nuclear proteins, and C4 enriched for ECM proteins. Representative blot showing one of the three biological replicates for each cell type. **b** Principal component analysis of mass spectrometry data from C1 and C4 fractions of both isolated glomeruli and tubules. **c** Mapping of organoid proteomic data onto the human matrisome database and the identification of 60 ECM proteins (Human matrisome resource: MatrisomeDB). **d, e** The expression profile of matrix proteins detected in OrgGloms and OrgPTs in the ECM enriched fraction (C4) and cellular fraction (C1). Gene ontology (GO) classifications (GO terms are based on the MatrisomeDB Gene Ontology divisions and categories) were used to group those detected into core matrix proteins (**d**), or proteoglycans and matrix-associated proteins (**e**). Error bars = SEM, n = 3 biological replicates. **f** Immunostaining of isolated OrgGloms shows basal expression of the GBM protein Laminin-α5. Representative image from >3 biological replicates. Expression levels of single channels shown in greyscale to preserve maximum contrast, merged image shown in colour. Scale bar 10 μm. **g** A comparison of matrix proteins identification in human organoid glomeruli (OrgGlom) versus: human glomerular matrix (GM), immortalised human glomerular endothelial cells (ciGEnC) (doi: 10.1681/ASN.2013070795), immortalised human podocytes (ciPod) (doi: 10.1681/ASN.2013030233) and GEnC and podocyte co-culture (doi: 10.1681/ASN.2013070795) based on previously reported data. Red bar denotes presence of this protein in the sample

## Discussion

The report in 1997[48] of the isolation and culture of primary podocytes in vitro and the subsequent derivation of conditionally immortalised podocyte cell lines[11,49] represented a major advance in our capacity to dissect podocyte biology. In this study, we have adapted a sieving approach for the isolation of intact glomeruli from human iPSC-derived kidney organoids and comprehensively characterised both the component podocytes either in 2D culture or as intact 3D glomeruli. Three-dimensional OrgGloms show comparable size and morphology to human glomeruli[5] and contain podocytes with appropriate apicobasal polarity and show a tighter transcriptional congruence to mature human glomeruli than that displayed by either a conditionally immortalised human podocyte cell line or organoid-derived podocytes cultured in 2D. Indeed, within 24 h of plating, component podocytes from within OrgGloms showed rapid downregulation of MAFB expression, highlighting the advantage of culture in 3D.

Proteomic analysis of OrgGloms revealed a distinct and complex ECM. OrgGloms showed evidence of maturing GBM components compared to 2D cultures, with an abundance of laminin-521. The 3D conformation of OrgGloms may promote synthesis of this more mature laminin network. The OrgGlom type IV collagen network showed enrichment of the α5(IV) chain; however, both α3 and α4(IV) chains were at the limit of detection, suggesting that additional cues such as blood supply are required for this isoform switch. FRAS1 and FREM2 are required for kidney development[36], while hemicentin is a component of the basement membrane adhesion complex B-LINK required for connecting adjacent basement membranes in *C. elegans*. These proteins were all enriched in OrgGloms, reflecting their embryonic nature. Of note, as expression of glomerular ECM components within OrgGloms became progressively enriched across time, this was accompanied by evidence for additional cells types, including both mesangial and endothelial populations. This reflects the proposed requirement for cellular crosstalk in glomerular maturation[50], with secreted growth factors and signalling ligands acting as paracrine and autocrine cross-talk effector molecules within and between glomerular cell types[51]. Equally, the ECM plays a crucial role in this process[52] and was shown to be maturing and complex in OrgGloms compared to podocytes in 2D culture. Hence, the OrgGlom model may facilitate studies into the mechanisms regulating collagen and laminin isoform switching and the role of additional components involved in GBM assembly, maintenance and repair.

Previous studies have reported the directed differentiation of human pluripotent stem cells to podocytes alone[53,54] or within an organoid setting[32,33]. However, most of these reports characterised their populations using 2D culture[33,53,54]. Profiling of podocytes in 3D within organoids has recently been performed via single cell RNA-seq of iPSC-derived organoids[55,56]; however, such transcriptional data are necessarily more shallow than the RNA-seq podocyte transcriptional profile we present here. Morphologically, OrgGloms arise at the proximal end of a developing nephron, apparently forming appropriate cell–cell interactions that facilitate foot process formation. Our profiling also suggests that, with time, minor endothelial and mesangial populations appear within these OrgGloms. OrgGloms also show an improved matrisome compared to 2D cultures, highlighting the importance of a 3D context for appropriate podocyte biology. Musah et al.[54] report the generation of 2D podocytes and seeding onto an organ-on-a-chip microfluidic device. The cells derived by this approach were compared to the same ciPods used in this study, and were found to be comparable, suggesting that these remain less accurate than OrgGloms. We also note that organoid-derived glomerular structures, generated using our protocol[17] and a previous differentiation protocol[32], draw in a recipient vasculature when transplanted in vivo[32,57], again supporting an appropriate identity for iPSC-derived glomeruli. Hence we conclude that OrgGloms represent a more accurate model of the podocyte than any 2D culture approach.

The modelling of disease by differentiating human iPSCs to kidney has been previously reported[15,33], using cell lines generated by CRISPR-Cas9 gene editing. In Kim et al.[33], characterisation was performed on *PODXL* null iPSCs. While applicable for genes expressed in undifferentiated iPSC, most podocyte genes are not expressed in this cell type (Supplementary Fig. 3). In this study, we have evaluated the utility of 3D OrgGloms to model CNS using patient-derived lines. As anticipated, OrgGloms isolated from patient kidney organoids displayed a reduction in NEPHRIN protein, validating the predicted degradation of truncated protein encoded by the exon 10 *NPHS1* variant which was hypothesised. Intracellularly, NEPHRIN binds to the adaptor protein PODOCIN, which facilitates its signalling function and proper localisation[58]. The exon 27 *NPHS1* variant was predicted to encode a protein with substantial loss of the 81 amino acids required to bind PODOCIN. Consequently, we observed a reduction in both PODOCIN and NEPHRIN protein in CNS OrgGloms derived from this patient. In contrast, the slit diaphragm protein NEPH1, which binds to the extracellular domain of NEPHRIN, was unaffected suggesting a potential compensation mechanism. In patients with the Finnish type (Fin-major) of CNS, in which there is homozygosity for an *NPHS1* nonsense mutation, patients have no detectable NEPHRIN protein[59], with slightly upregulated PODOCIN levels and significantly decreased NEPH1 levels[60]. This contrast to the heterozygous variants reported here emphasises the prospective utility of OrgGloms for the dissection of disease mechanisms, particularly in the case of variants of unknown significance.

In conclusion, our data demonstrate the accurate patterning and cellular identity of human organoid glomeruli compared to

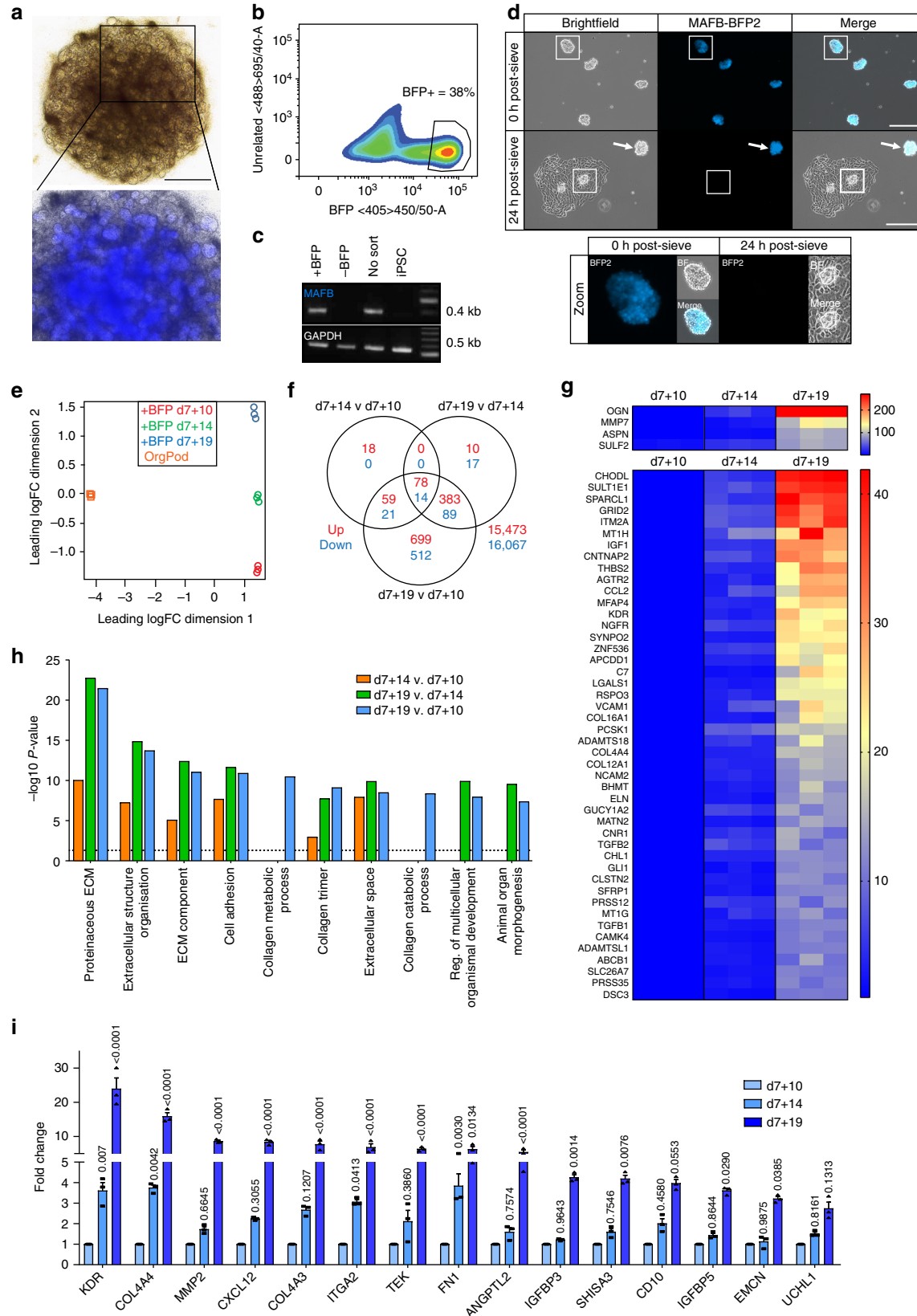

previous podocyte culture systems, highlighting the advantage conferred by the 3D format of OrgGloms. The capacity to readily generate large number of such an accurate model of the human glomerulus from iPSC in vitro will be a valuable tool to investigate both human glomerular development and disease. The

OrgGlom model will facilitate patient-specific functional genomics to validate novel podocytopathy genes and further interrogate the pathogenic mechanisms of existing podocytopathies and glomerular disease. This may also have utility for both drug efficacy and drug toxicity screening.

**Fig. 5** Temporal analysis of MAFB-expressing cells within organoids. **a** Differentiation of MAFB-BFP2 iPSC into kidney organoids was successful with blue fluorescent protein 2 (BFP2) expression observed in live organoids from d7 + 7 onwards. Scale bar 1000 μm. **b** FACS plot showing the BFP2-positive cell population isolated at d7 + 10. Representative plot of three biological replicates. **c** RT-PCR analysis of BFP2-positive and negative organoid cell fractions showed MAFB expression is only found in BFP2-positive cells. **d** Brightfield and BFP2-fluorescent live imaging of glomeruli isolated from d7 + 18 MAFB-reporter organoids at the time of plating and after 24 h culture. Strong BFP2 signal is observed within glomeruli when in suspension (0 h and 24 h arrow), but does not remain active when the glomeruli are adhered for culture, nor is it expressed in the migrating podocyte population (inset). Scale bar 200 μm. **e** A principle component analysis of RNA sequencing (RNA-Seq) data was performed on three biological replicates for d7 + 10, d7 + 14 and d7 + 19 BFP2-positive cell populations and compared to OrgPod samples. A clear separation between the 2D cultured organoid podocytes and MAFB BFP2-positive cells isolated from whole organoids was observed in dimension 1; in dimension 2 the variable of time was evident in the separation of the triplicates. **f** Venn diagram displaying the intersections of each comparison, upregulated genes are shown in red and downregulated genes are shown in blue. This shows the greatest number of statistically significant upregulated genes was identified in the OrgGlom (d7 + 19) vs immature podocyte (d7 + 10) MAFB-BFP2 population. **g** Heatmap showing the top 50 upregulated differentially expressed genes between d7 + 19 and d7 + 10, with many enriched genes found to transcribe proteins in the human glomerular ECM proteome. Fold change of log-normalised gene expression levels for each of the triplicate samples presented. **h** Gene Ontology (GO) enrichment analysis of the top 100 upregulated genes differentially expressed between d7 + 19 and d7 + 10 found enrichment of GO terms associated with extracellular matrix (ECM) maturation, collagen maturation and cell adhesion. The top 10 most statistically significant GO term categories are shown. *P*-value of 0.05 depicted as a dotted line. **i** Top 15 most-significantly upregulated genes with time, identified using a human renal glomerulus-enriched gene expression dataset[31]. Significant upregulation of genes associated with the maturation of additional glomerular cells, including endothelium and mesangial cells alongside specific GBM components and associated proteoglycans. Two-way ANOVA $p <$ 0.0001; error bars = SEM; significant difference between time points assessed by Tukey's multiple comparisons test; $n = 3$ biological replicates shown by symbols; *F*-value(Interaction) = 27.16, DF (Interaction) = 28; *F*-value (Gene) = 37.58, DF (Gene) = 14; *F*-value(Time) = 495.7, DF (Time) = 2

## Methods

**Differentiation of iPSC to kidney organoid.** Our previously reported protocol[18] was adapted to a feeder-free mechanism for use with cell lines cultured in Essential 8 Medium on a Geltrex cell substrate matrix (Thermo Fisher Scientific). The precise differentiation procedure remained unaltered. Briefly, iPSC were differentiated to kidney for 7 days, then separated to single cells and re-aggregated to form an organoid. Organoids were analysed at various time points following re-aggregation, described in this manuscript as d7 + 'x' number of days as an organoid, to a maximum of d7 + 18.

**Isolation of glomeruli from kidney organoids.** This method was adapted from protocols previously described for podocyte isolation from human kidney tissue[5,19,61]. iPSC-derived kidney organoids with an initial starting cell number of 200,000 differentiated iPSC were dissociated together in groups. A maximum of 12 organoids were used per group, this prevent blocking of the sieve mesh and single-cell contamination. Organoids were dissociated by incubation with TrypLE select enzyme (Thermo Fisher) for 12 min at 37 °C. Gentle mixing using a 1 ml pipette was applied every 3 min to aid dissociation, resulting in a homogeneous cell solution. Per group, a single 70 μm cell strainer (Falcon) was placed onto a 50 ml tube (Falcon) and the mesh hydrated with phosphate-buffered saline (PBS). The cell solution was added to the strainer in a stepwise manner using a 1 ml pipette, allowing flow through of the solution by gravity. Using the plunger from a 1 ml sterile syringe (Terumo) gently push the remaining cell solution captured on the strainer through the mesh, some matrices will remain. Wash the strainer thoroughly using PBS then discard, keep the cell flow through. Place a single 40 μm cell strainer (Falcon) onto a fresh 50 ml tube (Falcon) and hydrate the mesh. Apply the flow through cell mix collected this sieve, also in a stepwise manner allowing the single cells to flow through by gravity. Wash the sieve extensively with PBS to remove any remaining single cells, keep the flow through. To collect the largest glomeruli from the 40 μm cell strainer invert the sieve inside a 10 cm Petri dish, and wash the from below using PBS to flush out the captured glomeruli. Repeat this process using the final 30 μm cell strainer (Miltenyi Biotec) to collect the smaller glomeruli. Extensive washing of cell sieves throughout will minimise single cell contamination and allow isolation of a pure glomerular population.

**Isolation of kidney organoid proximal tubular cells.** Following organoid glomerular isolation, the remaining single cell organoid population was subjected to magnetic activated cell sorting (MACS) to isolate the proximal tubular fraction. Briefly, cells were centrifuged at 300 *g* for 5 min and resuspended in 300 μl of MACS buffer (D-PBS, 2 mM EDTA and 0.5% bovine serum albumin) containing 5 μl of LTL antibody (B-1325; Vector Laboratories). The cell suspension was incubated at 4 °C for 30 min, mixing gently every 10 min. Cells were rinsed with MACS buffer and centrifuged twice before resuspending in 100 μl of MACS buffer containing 35 μl of Streptavidin IgG Microbeads (Miltenyi Biotec) for 30 min at 4 °C. Cells were rinsed with MACS buffer and centrifuged twice before resuspending in 500 μL of MACS buffer and passage through an MS MACS column (one column per 6 organoids) according to the manufacturer's protocol (Miltenyi Biotec).

**Glomerular size analysis.** Glomerular diameter was determined using Feret's diameter, in concordance with previously published data[62].

**Culture of podocytes and sieved glomeruli.** Isolated glomeruli from iPSC-derived kidney organoids were plated on glass chamber slides coated with recombinant human laminin-521 (Biolamina) or onto tissue-culture coated plastic. For culture of glomeruli in suspension, sieved glomeruli were cultured low attachment culture plates (Corning). Cells were supplemented with previously defined podocyte growth media[11] and incubated in a humidified atmosphere at 37 °C plus 5% $CO_2$. Media was changed every other day post-seeding. Conditionally immortalised human podocytes were cultured as previously described[11].

**Transmission electron microscopy.** Samples were prepared by cutting approximately 1 mm² sections of organoid, fixing in 1.5% PFA plus 0.5% glutaraldehyde, then processed using the ROTO technique as previously described[63]. Samples were sectioned and observed on a Tecnai F30 at 200 kV.

**Scanning electron microscopy.** Glomeruli cultures were fixed in 2% paraformaldehyde (PFA), 2.5% glutaraldehyde in 0.1 M sodium cacodylate buffer and post-fixed in 2% osmium tetroxide in 0.1 M sodium cacodylate buffer. Following fixation, cells were rinsed in distilled water and dehydrated through a graded series of alcohols, then critical point dried using a Leica EMCPD300 automated critical point drier. Organoids were sputter coated in gold using an Emscope SC500 sputter coater and imaged in a JEOL Neoscope II SEM.

**Cellular uptake assay.** FITC-albumin uptake assays were performed as previously described[25]. Briefly, podocytes were pre-incubated for 60 min in Ringer solution, then exposed to 1.5 mg/ml FITC-labelled albumin (Sigma) in Ringer solution for 60 min at either 4 °C or 37 °C. Cells were then washed in ice-cold PBS and fixed with 2–4% PFA followed by immunofluorescent staining to mark F-actin and cell nuclei with visualisation by confocal microscopy (described in Immunohistochemical analysis).

**Insulin stimulation assay.** Podocytes were stimulated with 10 mg/ml human insulin (Sigma) for 10 min at 37 °C as previously described[28]. Following stimulation cells were immediately washed in ice-cold PBS and fixed with 2–4% PFA followed by immunofluorescent staining to mark GLUT4, F-actin and cell nuclei with visualisation by confocal microscopy (described in Immunohistochemical analysis).

**Immunohistochemical analysis.** Samples were fixed in 2–4% PFA, then washed and stored in PBS at 4 °C until needed. Immunofluorescent staining of organoids[18] and podocytes[29] was performed as previously described. Haematoxylin and eosin (H&E) staining was performed on organoid serial sections cut with a microtome at a thickness of 4 μm. Immunofluorescence was visualised with an LSM780 confocal microscope (Carl Zeiss) or a Dragonfly Spinning Disc Confocal (Andor Technology). 3D modelling software (Imaris 8, Bitplane, Connecticut) was utilised to reconstruct serial Z-stack images acquired via confocal microscopy.

**Evaluation of fluorescence intensities within glomeruli.** Fluorescence intensity of isolated OrgGloms, from both immunostaining or MAFB-BFP2 reporter fluorescence, was analysed in Fiji using the Multi Otsu Threshold algorithm[64].

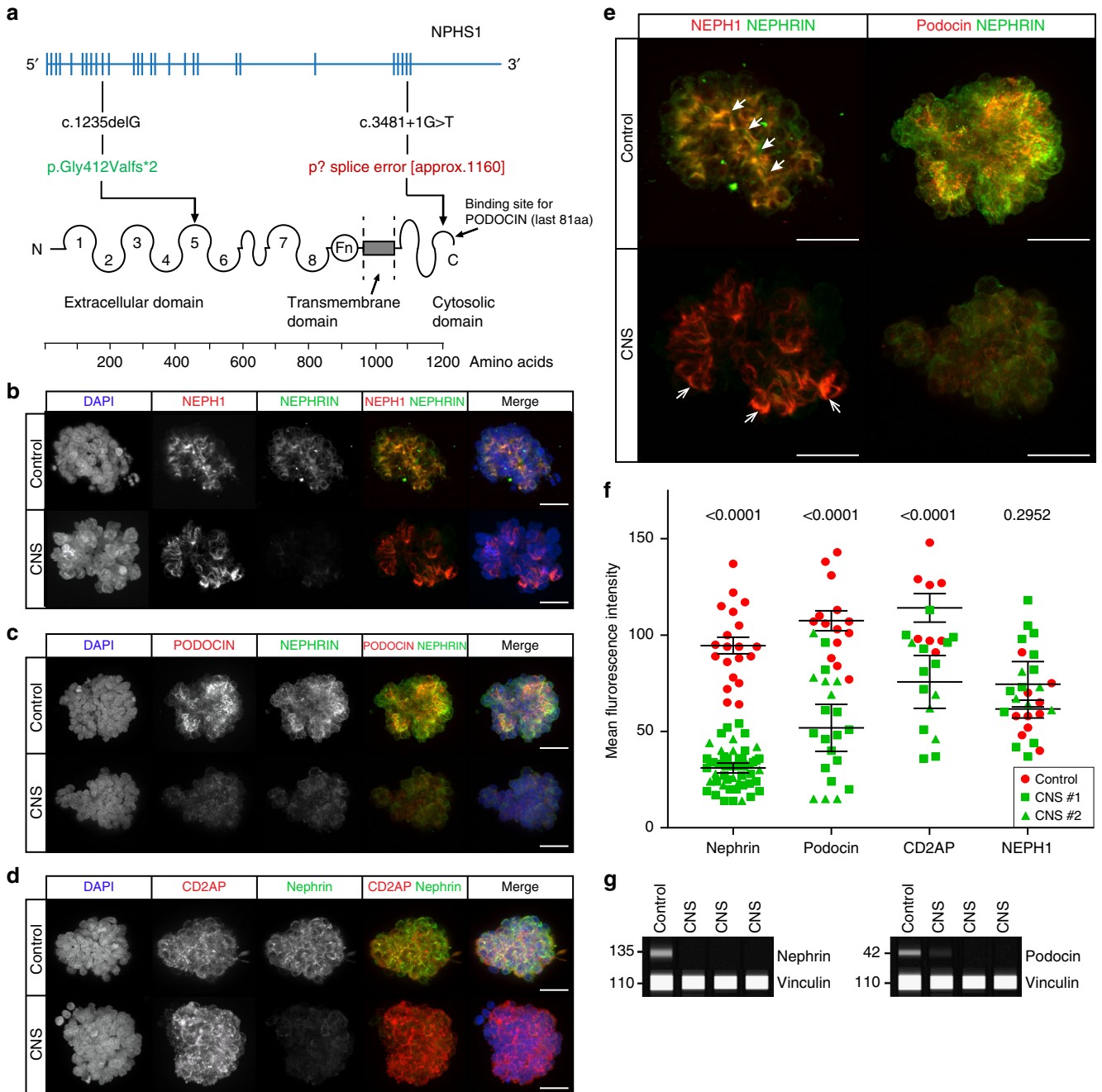

**Fig. 6** Organoid Glomeruli model of congenital nephrotic syndrome in vitro. **a** Description of the *NPHS1* variants identified in the patient modelled, diagnosed with congenital nephrotic syndrome (CNS). **b–d** Immunostaining of OrgGloms isolated from control organoids and CNS patient organoids show reduced NEPHRIN and PODOCIN protein levels in the organoids derived from patient-iPSC, representative images shown of >3 biological replicates. Scale bars 10 μm. **e** Higher power immunofluorescent images show the polarised co-localisation of NEPHRIN with NEPH1 (solid white arrowheads) and PODOCIN in control OrgGloms. This polarisation is lost in CNS OrgGloms due to the absence of NEPHRIN (white arrows). Scale bars 10 μm. **f** Quantitative analysis of fluorescence intensities from independent OrgGlom biological replicates performed using one control and two distinct patient-derived CNS iPSC clones. Organoid glomeruli generated from both patient-derived iPSC clones show significant reduction in NEPHRIN and PODOCIN protein levels. Two-way ANOVA $p < 0.0001$; error bars = SEM. Biological replicates. NEPHRIN (controls, $n = 20$; CNS, $n = 56$); PODOCIN (controls, $n = 14$; CNS, $n = 22$); CD2AP (controls, $n = 8$; CNS, $n = 15$); NEPH1 (controls, $n = 10$; CNS, $n = 17$). Significant difference assessed by Sidak's multiple comparisons test between cell lines; $F$-value $= 112$; DF $= 1$. NEPHRIN: control vs CNS#1, $p < 0.0001$; control vs CNS#2, $p < 0.0001$; CNS#1 vs CNS#2, $p > 0.9999$. PODOCIN: control vs CNS#1, $p < 0.0001$; control vs CNS#2, $p < 0.0001$; CNS#1 vs CNS#2, $p = 0.9995$. CD2AP: control vs CNS#1, $p = 0.0007$; control vs CNS#2, $p = 0.0016$; CNS#1 vs CNS#2, $p = 0.9980$. NEPH1: control vs CNS#1, $p = 0.5320$; control vs CNS#2, $p = 0.9994$; CNS#1 vs CNS#2, $p = 0.9992$. **g** Quantitative western blot analysis of NEPHRIN and PODOCIN protein levels within independent biological replicates confirms the significant depletion of these proteins in OrgGloms derived from CNS iPSCs

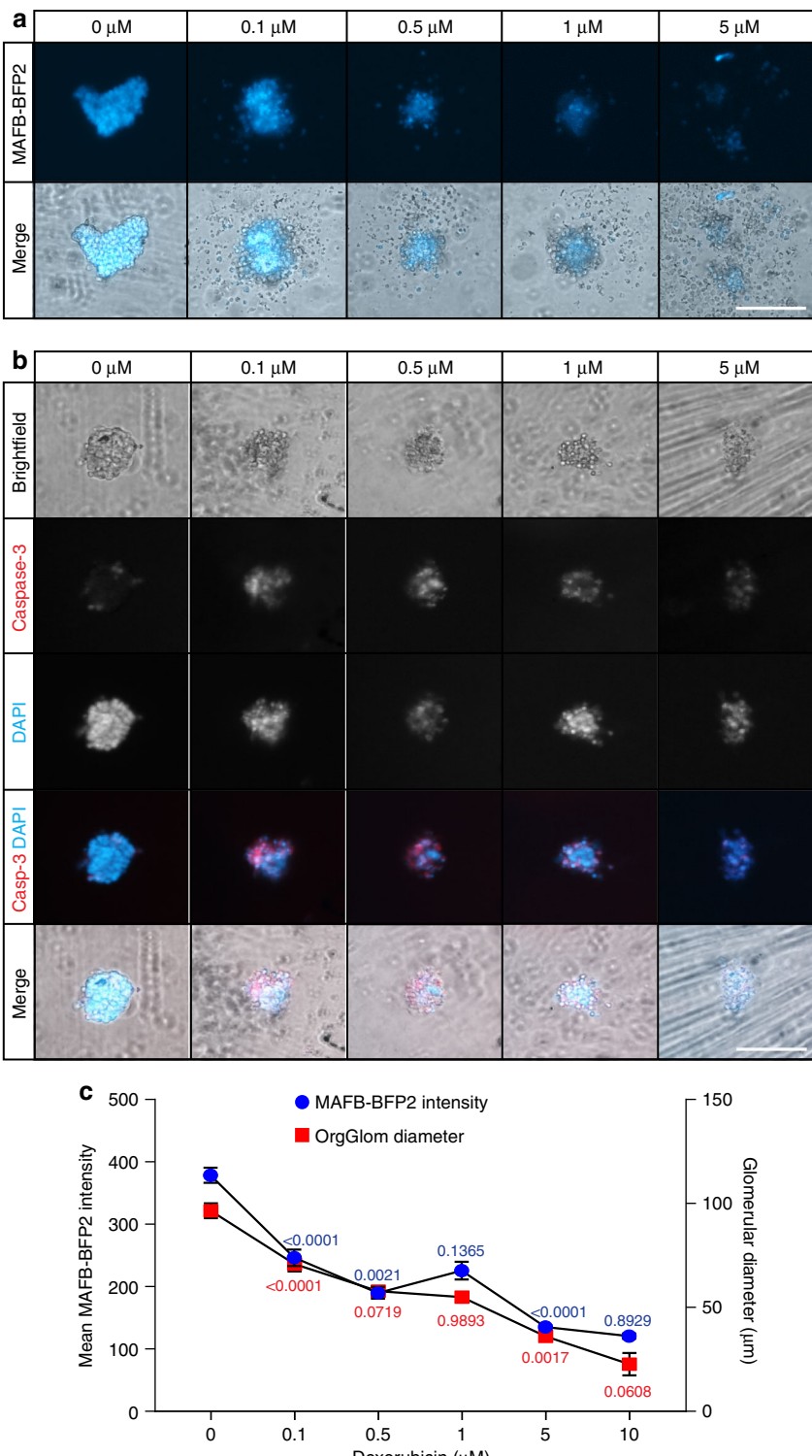

**Fig. 7** Toxicity screening using cultured organoid glomeruli. **a** MAFB-BFP2 OrgGloms can be cultured in isolation in a 96-well format. Live imaging of glomeruli at 48 h post-treatment with doxorubicin showed a dose-dependent decrease of BFP2-reporter intensity. **b** Fixed glomeruli immunolabelled with Caspase-3 showed activation of this pro-apoptotic pathway following doxorubicin treatment. Scale bar 100 μm. **c** Semi-quantitative analysis of BFP2-reporter intensity (blue) alongside OrgGlom diameter (red) showed a dose-dependent reduction in both glomerular diameter and BFP2-reporter intensity. One-way ANOVA $p < 0.0001$; error bars = SEM; significant difference between sequential doses assessed by Tukey's multiple comparisons test. BFP2-reporter intensity: $F$-value = 80.3; DF = 5; biological replicates $n = 16$ per dose. OrgGlom Diameter: $F$-value = 60.94; DF = 5; biological replicates $n = 16$ per dose

Mean intensity values for each individual glomerulus were plotted alongside standard error.

**Antibodies**. The following primary antibodies were used for immunofluorescence at a concentration of 1:200: CD2AP (sc-9137; Santa Cruz), NEPHRIN (AF4269; R&D Systems), PODOCIN (P0372; Sigma Aldrich), SYNAPTOPODIN (sc-21537; Santa Cruz), PODOCALYXIN (39-3800; Thermo Fisher), WT1 (m3561; Dako), P-CADHERIN (sc-7893; Santa Cruz), GLUT4 (MAB1262; R&D systems), FcRN (sc-46328; Santa Cruz), NEPH1 (HPA030458; Atlas Antibodies), Caspase-3 (9661; Cell Signalling), E-CADHERIN (610181; BD Biosciences), LTL (B-1325; Vector Laboratories), LAMA5 (ab77175; Abcam). Secondary antibodies were conjugated with Alexa 488, 568 or 647 and used at 1:400 alongside Phalloidin and DAPI where appropriate, both used at 1:1000 (Life Technologies). The following primary antibodies were used for western blotting: NEPHRIN (DB017-005; Acris), PODOCIN (P0372; Sigma Aldrich), VINCULIN (V9131; Sigma-Aldrich). Secondary antibodies included Goat Anti-Mouse Secondary HRP conjugate (042-205; ProteinSimple) and Goat Anti-Rabbit Secondary HRP conjugate (042-206; ProteinSimple). These were titrated to achieve a dynamic range as described in the ProteinSimple online methods.

**Simple western blot**. Sieved OrgGloms from six iPSC-kidney organoids per sample were pooled and then lysed in RIPA buffer with protease inhibitor (1:500) for 15 min on ice. Lysates were vortexed for 45 s and centrifuged at 13.3 $g$ for 10 min at 4 °C. The supernatant was collected and diluted 1:4 with 1× Sample Buffer (ProteinSimple). Protein quantification was performed using a 12–230 kDa 25-lane plate (PS-MK15; ProteinSimple) in a ProteinSimple Wes Capillary Western Blot analyser according to the manufacturer's instructions. See Source Data file for raw data.

**Quantitative RT-PCR analysis**. RNA was isolated using an RNeasy mini or micro kit (Qiagen) and then reverse transcribed using the SensiFAST™ cDNA synthesis kit (Bioline). Quantitative PCR was performed using the 7500 Real Time PCR System Thermal Cycler (Applied Biosystems) and SensiFAST™ SYBR® No-ROX cDNA synthesis kit (Bioline). All samples were normalised against GAPDH expression. OrgGlom fractions were isolated from 12 individual kidney organoids per sample and pooled in order to obtain sufficient material.

**RNA-sequencing profiling and statistical analysis**. RNA was extracted, in triplicate, from isolated glomeruli sieved and pooled from d7 + 18 kidney organoids, alongside organoid podocytes cultured from glomeruli for 72 h and both undifferentiated and differentiated conditionally immortalised podocytes (GEO accessions: GSE99583 and GSE111992). Additionally, RNA was extracted from organoids generated using a MAFB reporter line[57] using both FACS sorting and glomerular sieving. All isolations were performed in triplicate at d7 + 10, d7 + 14 and d7 + 19, respectively. At all timepoints, MAFB-mTAGBFP+ cells were enriched using FACS sorting and glomerular sieving then pooled for RNA isolation. Sequencing libraries were prepared using Illumina's TruSeq stranded total RNA protocol with RiboZero rRNA depletion. In total, 24 samples were sequenced using an Illumina NextSeq 500 sequencer (GEO accession: GSE111992). Transcript-level abundances were quantified from the 75 bp paired end reads using Salmon v0.8.2 and the ENSEMBL GRCh38 version of the human transcriptome. Gene level counts were obtained using the tximport R package[65]. For the publicly available RNA-Seq data on differentiated conditionally immortalised podocytes (GEO accession: GSE80651), the fastq files were downloaded from SRA (SRP073810) and processed as described above. Genes that had expression levels of at least 1 count per million in at least three samples in the organoid to conditionally immortalised comparisons, with the same requirement between the reporter line time course samples, were kept for further statistical analysis. The data were TMM normalised[66] and Voom transformed[67]. Differentially expressed genes were identified using moderated *t*-tests followed by a TREAT test from the R Bioconductor limma package[68]. Genes that had a false discovery rate less than 5% and an absolute log2-fold-change of greater than 1 were called significantly differentially expressed. Gene ontology analysis was performed using the goana function in limma on the top 100 most up-regulated and down-regulated genes for each comparison between conditionally immortalised and organoid samples, and between samples in the time course, separately (Fig. 3d). A publicly available Affymetrix microarray data set of human glomerular and tubular samples (GEO accession: GSE21785) was normalised using the gcrma algorithm with differentially expressed genes identified using moderated *t*-tests, including patient as a fixed effect in the model. From this, a glomerulus expression signature was defined as including the top 100 most up-regulated genes when comparing glomerular samples to tubular samples. A 'glomerular expression score' was then generated by evaluating expression for each signature gene in each sample cell type and creating a mean value (Fig. 3g, see Source Data file). For the publicly available RNA-Seq data on control iPSC-derived kidney (GEO accession: GSE103547) raw counts were downloaded. For every gene the raw counts for each biological replicate were normalised to their respective GAPDH raw counts. Matched analysis was performed on in-house RNA-seq data to allow comparison.

**Extracellular matrix isolation and enrichment**. ECM enrichment was carried out as previously described[5]. Briefly, the organoids were lysed in Tris-buffer (10 mM Tris, 150 mM NaCl, 1% [vol/vol] Triton X-100, 25 mM EDTA, 25 mg/ml leupeptin, 25 mg/ml aprotinin, and 0.5 mM 4-(2-aminoethyl)-benzenesulfonyl fluoride hydrochloride) to extract soluble proteins, and the samples were centrifuged at 14,000×$g$ for 10 min at 4 °C. The supernatant was kept on ice (fraction C1), and the pellets were resuspended in alkaline detergent buffer for 30 min (20 mM NH$_4$OH and 0.5% [vol/vol] Triton X-100 in PBS) to solubilise and disrupt the ECM interaction with the remaining cellular components. The samples were centrifuged at 14,000×$g$ for 10 min at 4 °C, and the supernatant was kept on ice (C2). To degrade the genomic materials (DNA/RNA) in the remaining pellets, the pellets were resuspended in a deoxyribonuclease buffer (10 mg/ml deoxyribonuclease I [Roche, Burgess Hill, UK] in PBS) for 30 min. The samples were centrifuged at 14,000×$g$ for 10 min at 4 °C, and the supernatant was kept on ice (C3). The residual pellet contains most ECM components, which were resuspended in reducing sample buffer (50 mM TrisHCl, pH 6.8, 10% [wt/vol] glycerol, 4% [wt/vol] SDS, 0.004% [wt/vol] bromophenol blue, and 8% [vol/vol] b-mercaptoethanol) to yield the ECM fraction (C4). Samples were heat denatured at 95 °C for 10 min.

**Mass spectrometry data acquisition and analysis**. Protein samples were run into SDS-PAGE gels for 3 min to concentrate proteins in the gel top and these were visualised by InstantBlue staining. Gel tops from C1 and C4 fractions were subjected to in-gel trypsin digestion as described previously[5]. Liquid chromatography–tandem MS analysis was performed using a nanoACQUITY UltraPerformance liquid chromatography system (Waters, Elstree, UK) coupled online to an LTQ Orbitrap mass spectrometer (Thermo Fisher Scientific, Waltham, MA). Peptides were concentrated and desalted on a Symmetry C18 preparative column (20-mm length, 180-mm inner diameter, 5-mm particle size, 100-Å pore size; Waters). Peptides were separated on a bridged ethyl hybrid C18 analytical column (250-mm length, 75-mm inner diameter, 1.7-mm particle size, 130-Å pore size; Waters) using a 45-min linear gradient from 1 to 25% (vol/vol) acetonitrile in 0.1% (vol/vol) formic acid at a flow rate of 200 nl/min. Peptides were selected for fragmentation automatically by data-dependent analysis. Tandem mass spectra were extracted using ProteoWizard msConvert[69] executed in Mascot Daemon (version 2.5.1; Matrix Science, London, UK). Peak list files were searched against Swiss-Prot and Trembl Database. Carbamidomethylation of cysteine was set as a fixed modification; oxidation of methionine and hydroxylation of proline and lysine were allowed as variable modifications. Only tryptic peptides were considered, with up to one missed cleavage permitted. Monoisotopic precursor mass values were used, and only doubly and triply charged precursor ions were considered. Mass tolerances for precursor and fragment ions were 10 ppm and 0.5 Da, respectively. MS quantification and proteomic data analyses were performed using Progenesis QI software (version 2.3; Nonlinear Dynamics, Newcastle upon Tyne, UK), where the sample was submitted to automatic alignment and reference selection at default settings, and peptides with a rank greater than 4 were removed. The proteins were quantified using Hi-N relative quantification with a minimum of three peptides per protein, and the normalised abundance calculated within Progenesis QI was used for further data analysis. PRIDE: px-submission #244453.

**Flow cytometry**. iPSC-derived kidney organoids were dissociated by incubation with TrypLE select enzyme (Thermo Fisher Scientific) for 10 min at 37 °C, with gentle pipetting every 2 min to aid dissociation. Cells were then passed through a series of cell strainers with sequentially smaller mesh sizes, ranging from 100 to 40 μM (pluriSelect), in order to obtain a single cell population. mTagBFP2-positive OrgGloms were captured and isolated from within the sieves. The remaining single cell population was then sorted for mTagBFP2 signal using a FACSAria Fusion flow cytometer (BD Biosciences). See Source Data file.

**MAFB-mTagBFP2 glomerular toxicity screen**. MAFB-mTagBFP2 glomeruli were isolated from iPSC-derived kidney organoids and manually transferred to individual wells of a low-bind 96-well plate microplate (Corning). Glomeruli were supplemented with previously defined podocyte growth media[11] containing a serial dilution of doxorubicin from 0 to 5 μM, and incubated in a humidified atmosphere at 37 °C plus 5% $CO_2$ for 48 h with rotation at 60 rpm. At 48 h post-treatment glomeruli were live imaged for BFP2 intensity with an Apotome.2 fluorescent microscope (Carl Zeiss), then fixed in 4% PFA and immunofluorescently stained for Caspase-3 activity (described in Immunohistochemical analysis).

**Clinical case summary and histopathology**. The proband was born at 35/40 weighing 1.83 kg. She presented to medical services at 7 weeks with oedema, ascites, hypoalbuminuria (serum albumin 19 g/l) and proteinuria (urinary protein:creatinine ratio 9042 mg/mmol). All other features were normal. Her parents were of European/Australian origin with no history of consanguinuity or family history of renal disease. A unilateral nephrectomy was performed at 3 months, resulting in a reduction in GFR and decreased albumin requirements. She is currently awaiting transplant with an estimated GFR by the Schwarz method of 10 ml/min/1.73 m$^2$.

Light microscopy examination of tissue removed at the time of transplantation revealed mild to moderate mesangial prominence involving most glomeruli (Supplementary Fig. 5A and B). Occasional epithelial proliferation was associated

with sclerosis. A moderate number of tiny glomeruli were seen beneath the capsule. Transmission electron microscopy revealed severe global effacement of podocyte foot processes with widespread, marked microvillous change (Supplementary Fig. 5C). Basement membranes appeared globally thin but without lamination or splitting, as considered normal for age. There was some segmental mesangial hypercellularity without convincing matrix expansion which may correspond to the microglomeruli identified on light microscopy. One glomerulus showed marked Bowman's capsule thickening with an apparent increase in number of epithelial cells. Histology was consistent with a diagnosis of CNS.

**Patient genome sequencing and analysis**. This research was conducted with approval from the human research ethics committees of the Royal Children's Hospital Melbourne (HREC/15/QRCH/126) including research governance approval (SSA/15/RCHM/90). Written informed consent was obtained from the participant. The patient presented with CNS requiring nephrectomy, and histology consistent with diffuse mesangial sclerosis (see Supplementary Data). The proband was tested for 37 genes associated with nephrotic syndrome using a HaloPlex target enrichment and Next Generation Sequencing via the Bristol Genetics Laboratory. Genes tested included ACTN4, ALG1, ALMS1, APOL1, ARHGAP24, ARHGDIA, CD151, CD2AP, COL4A3, COL4A4, COL4A5, COQ2, COQ6, CYP11B2 (association), E2F3, INF2, ITGA3, ITGB4, LAMB2, LMX1B, MYH9, MYO1E, NPHS1, NPHS2, PDSS2, PLCe1, PMM2, PTPRO, SCARB2, SMARCAL1, TRPC6, WT1 and ZMPSTE24. Analysis of the NPHS1 gene identified a novel heterozygous single base-pair deletion variant, c.1235delG, p.(Gly412Valfs*2) in exon 10 and a heterozygous splice-site variant, c.3481 + G > T in exon 27.

**Derivation of patient iPSC**. PBMCs were isolated from whole blood using density gradient centrifugation. Once isolated, 500,000 PBMCs were cultured in Erythroid expansion medium (Stem Cell Tech; Cat# 09605 and 02692) for 7 days in a humidified atmosphere at 37 °C plus 5% CO$_2$, with media replenished every other day. At day 7, cultured PBMCs were counted, 500,000 cells pelleted and then re-suspended in erythroid expansion medium + Sendai virus (Life Tech; Cat# A167517) with MOIs of 5, 5 and 5 (i.e., KOS MOI = 5, hc-Myc MOI = 5, h-Klf4 MOI = 5) plus the addition of polybrene (4 μg/ml) to a final concentration of 500,000 cells/ml. Cells were added to a single well of a 12-well plate and centrifuged at 2000 RPM for 30 min at room temperature. After 24 h in culture, a medium change with fresh erythroid expansion medium was performed (pellet at 300 RCF for 5 min at RT) to remove the Sendai virus and cells returned to the original well. After a further 48 h (Day 3 post transduction), cells are plated onto MEFs in 50:50 mix of StemSpan SFEM II (Stem Cell Tech; Cat #09605) and hiPSC media. Additional hiPSC media was added 2 days later, and a full media change at Day 7 post transduction. Daily media changes were performed thereafter until colonies are ready to be picked (Day 21 post transduction). SNP analysis was performed by the Victorian Clinical Genetics Service using an Infinium CoreExome-24 v1.0 DNA microarray (Illumina) (see Source Data file). No aneuploidies were detected. iPSC clones tested negative for mycoplasma contamination. Pluripotency of iPSC lines was confirmed by flow cytometry after staining with antibodies to pluripotency markers (CD9, SSEA4, TRA-1-81) (Supplementary Fig. 5D and E). Sequencing of the NPHS1 gene was performed in both patient-derived iPSC clones to verify the presence of compound heterozygous mutations in NPHS1 (Supplementary Fig. 5F). The capacity of each clone to differentiate into kidney organoids was also evaluated (Supplementary Fig. 5G).

**Reporting summary**. Further information on experimental design is available in the Nature Research Reporting Summary linked to this paper.

## Data availability

GEO accessions of newly data generated for the manuscript: GSE111992, GSE99583. GEO accessions of publicly available data referenced within the manuscript: GSE80651, GSE21785, GSE103547. PRIDE accession of newly data generated for the manuscript: PXD244453. PRIDE accessions of publicly available data referenced within the manuscript: PXD000456, PXD000643. The source data underlying Figs. 1b, 1d, 3a–g, 4a–d, 5b, d–h, s, 6f, g, 7c, Tables 1–4, Supplementary Figs. 2, 3 and 5 are provided as a Source Data file. A reporting summary for this article is available as a Supplementary Information file.

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

## Acknowledgements

We thank Moin Saleem, University of Bristol for the provision of the conditionally immortalised podocyte cell line. RNA sequencing was performed by the Translational Genomics Unit, Murdoch Children's Research Institute. We thank the family of the patient described in this study and we acknowledge the KidGen consortium, particularly Andrew Mallett, for recruitment ethics. iPSC patient line derivation was performed by iPSC Core Facility, Murdoch Children's Research Institute. Mass spectrometry was performed in the Biomolecular Analysis Core Facility, University of Manchester, UK and supported by fellowship funding (to R.L.) from the Wellcome Trust (202860/Z/16/Z). M. C.R.I. is supported by the Victorian Government's Operational Infrastructure Support Program. We also thank the Centre for Advanced Histology and Microscopy at the Peter MacCallum Cancer Centre for assistance with scanning electron microscopy. A.O. is an NHMRC Career Development Fellow. M.H.L. is an NHMRC Senior Principal Research Fellow (GNT1136085). This work was supported by the National Health and Medical Research Council of Australia (GNT1098654), the National Institutes of Health (DK107344-01) and the Royal Children's Hospital Foundation for funding of the ReGeniPS program.

## Author contributions

M.H.L. and L.J.H. planned the project, designed the experiments, analysed and interpreted data and wrote the manuscript. L.J.H. performed experiments. S.E.H. derived the iPSC reporter line. P.X.E., I.G. and S.K. maintained iPSCs and generated organoids. S.K. prepared material and reviewed SEM data. B.P., A.L. and A.O. analysed bioinformatics data. R.L. and S.H. performed proteomic data analysis and provided interpretation. S.W. assisted in confocal image modelling and western blot analysis. K.T.L. assisted in computer-based objective analysis of OrgGloms. C.Q. recruited the patient studied and assisted in clinical interpretation.

## Additional information

**Competing interests:** M.H.L. is an inventor on a patent around the differentiation protocol; this patent has been licenced to Organovo Inc. The remaining authors declare no competing interests.

