## [Peer Review File · Nature Communications]

Reviewers' comments:

Reviewer #1 (Remarks to the Author):

This is an exciting paper describing characteristics of podocytes and glomeruli isolated from organoids that have been generated from human iPSC cell lines. There are two major advances reported:

1) Description of a technique to generate whose transcriptional profile more faithfully recapitulates the profile found in human podocytes isolated from tissues, thus providing a new model to explore podocyte biology.

2) Description of a technique to develop vascularized glomeruli using a chick allantoic (CAM) procedure. The vascularized glomeruli that form in this setting provide an new technique to study podocyte development, explore human genetic podocytopathies in the lab and provide access to a system to study parietal epithelial cells.

Some comments for consideration by the authors:

1) A very interesting question is raised by the findings of vascularization – why does implantation using CAMs promote development and invasion of human-derived endos into glomeruli (Sox17-mCherry endos)?

2) Some discussion re: limitations of the comparison of RNAseq data from historical, published samples would be helpful. Some additional discussion is valuable e.g. discussion of clonal differences between immortalized podocyte cell lines should be considered.

3) Some discussion of use of primary podocytes for studies that might demonstrate a more faithful recapitulation of podocyte expression profile (an alternate to conditionally immortalized podocytes) could be included.

4) Some comment about limitations of organoid technique – i.e. large scale proteomics or epigenetic studies will not currently be possible with organoid derived-gloms but are possible with immortalized human podocytes

5) Some discussion (even if speculative) about reason for lack of vascularization in organoids would be valuable

6) There is quite a lot known about the switch of isoforms of laminins during development – appropriate references should be cited

7) Fig 6 – any chance to do immunogold to see slit diaphragms? How frequently are slit diaphragms found?

8) What % of endos are fenestrated? Is it possible to quantify? Some molecular characterization of the glomerular endos would be of interest

9) Fig 4 and general comment – what about profiling vascular growth factor- promoting genes?

Included in sup material but given focus on vascularization – some discussion about pro-angiogenic factors in ciPods vs. organoids would be helpful

Reviewer #2 (Remarks to the Author):

In the present study, Hale and colleagues describe a series of studies with human organoids and organoid-derived podocytes that led them to conclude that human iPSC-derived glomeruli facilitate accurate modelling of podocytopathies. Given the prevalence of podocytopathies, this is an important area of investigation. The seminal work by the Little lab, which led to the generation of human kidney organoids, was a breakthrough on the way to achieve this important goal. The current manuscript is the beginning of a nice body of work but is incomplete. Additionally, the claims are overstated given the evidence presented. Overall, the manuscript lacks novelty, because key findings, such as the vascularization of transplanted organoids, has already been published by two other groups (Ref. 45 and Xinaris et al, JASN, 2016: Functional Human Podocytes Generated in Organoids from Amniotic Fluid

Stem Cell). The chicken chorioallantoic membrane transplant model data establishes an alternative to previously published mouse and rat transplant models. However, it does not fit with the rest of the manuscript unless data is presented that demonstrates its utility in human disease modeling. At a minimum, it would be interesting to see the histopathology of transplanted MAFB^{-/-} organoids.

Lack of proper controls. To claim that a new in vitro system is superior than the current “gold standard” in modeling an in vivo derived human glomerulus or podocytes, the new system must be compared, at all levels, to the in vivo counterpart and to the true gold standard in the field, even if that is an in vitro system with differentiated mouse podocytes. In this particular case, the new system described needs to be compared to the immortalized mouse podocytes, which are the current gold standard, the immortalized human podocytes, which allow for species-specific comparisons to the new system, and the in vivo derived counterpart, which sets the bar against which the new system will be measured. In addition, it is well known in the stem cell field that iPSC derivatives are developmentally immature. This needs to be taken into consideration when interpreting the results. Ideally, there would be both fetal and adult control samples in the data presented.

Thus, to provide compelling support for their claim that OrgPods display increased differentiation, the authors should compare the OrgPods not only to the human immortalized cell line described by Saleem and colleagues (Ref. 12), but also to other podocyte cell culture systems described by many investigators, including iPSC-derived podocytes (Song et al. PLoS One, 2012; Ciampi et al. Stem Cells 2016), human primary podocytes from adult normal kidney (Li et al. JASN ,2017, Yang et al, Exp Cell Res 2017, Mundel et al. JASN 1997), human podocyte cell lines established from urine (Sakairi, AJPPhysiol 2010), as well as differentiated mouse podocyte cell lines (Mundel et al. Exp Cell Res 1997). Such a study would go a long way when trying to show “superior” differentiation of OrgPods. The results of such a comprehensive study would also be of great importance for the field, especially if such studies could confirm the postulated higher degree of differentiation of OrgPods. As a final point on proper controls, it is also well known in the field that structure affects function. The authors themselves even present supportive data in Figure 4C. This fact needs to be taken into consideration when choosing the proper controls and interpreting results.

A major advantage of podocyte cell lines is the possibility to grow up large batches of cells. Does the approach described here enable the generation of large enough quantities of OrgPods from organoid-derived isolated glomeruli that are required for biochemical and functional studies?

The title of the manuscript suggests that the focus is on modeling human podocytopathies with patient iPSCs. To explore this hypothesis, the authors generate iPSCs lacking the transcription factor MAFB/Kreisler, which when deleted in mice, lead to FSGS-like disease and postnatal lethality. The authors then assess changes in gene expression resulting from the deletion of MAFB/Kreisler and conclude that the observed changes are relevant to human pathology. This is somewhat surprising, as to this date, neither genetic nor clinical data have been published that would link MAFB/Kreisler to human podocytopathies. Why engineer a MAFB mutant iPSC line instead of generating an iPSC line from a patient with podocytopathy? Without human-relevant data, e.g. knockin of a human podocytopathy gene mutation or studying a patient-derived iPSC line, followed by the analysis of the resulting phenotype, the title of the manuscript and the conclusion of the abstract are not supported by data.

The manuscript also contains a set of studies using organoid transplantation onto chicken chorioallantoic membranes. In these experiments, the authors show that the organoid transplantation results in glomerular vascularization, filtration barrier assembly and slit diaphragm maturation. The results of these studies are neither new nor surprising. Similar grafting studies in rats and mice have already been published (Xinaris et al, JASN, ref. 46 Sharmin et al.) and both studies showed that

iPSC-derived glomeruli in the transplanted organoids exhibit many morphologic features of glomeruli in vivo, including a well-developed GBM, highly organized podocyte foot processes and slit diaphragms, which appear more mature than those junctions shown in this study (Fig. 6c). The latter look more like the “ladder-like” junctions that are found between developing podocytes (Reeves et al., *Lab Invest*, 1978), confirming their immature character.

The chorioallantoic membrane system appears to be of limited use for probing the function of the glomerular filtration barrier. Taking advantage of tracer studies with FITC- and RITC-conjugated dextrans of different molecular sizes, Xinaris and colleagues showed the functional maturation of the filtration barrier in transplanted human/mouse hybrid organoids. This approach appears preferable to the chicken chorioallantoic membrane model, in which filtration studies or hemodynamic studies cannot be conducted. This is of concern, as the critical role of glomerular hemodynamics in the development and progression of glomerular diseases is well established. It would therefore be of great interest to see the degree of functional maturation of the glomerular filtration barrier after transplanting the authors' iPSC-derived human organoids under the capsule of rat or mouse kidneys.

Figure 1 describes the isolation of capillary loop stage glomeruli from the organoids. These glomeruli are still developing and not fully mature. Is it possible to isolate (more) mature glomeruli if the organoids are maintained for a longer time in culture before isolation or by treating them with VEGF?

The authors report that a population of pure glomeruli with a diameter in the range of 50- 90 μm can be isolated (Fig. 1a_{ii}); this is markedly smaller than that of mature human glomeruli, which is about 200 μm . What is the size distribution of these isolated glomeruli, how many glomeruli can be isolated from an individual organoid and how reproducible is the procedure? What percentage of the isolated glomeruli contain a Bowman's capsule versus being acapsular? How does the glomerular size change after isolation from chorioallantoic membrane transplants?

The TEM analysis (Fig. 1c) should be supplemented with high resolution images of the podocyte (foot) processes and cell-cell junctions. The upregulation of nephrin mRNA expression (Fig. 1d) should be confirmed at the protein level by Western blotting.

The analysis of GBM components in Fig. 1e should include the analysis of the mature $\alpha 3$, 4 and 5 chain of Type IV collagen. The confocal images of the podocyte proteins are not very informative in the present version. The authors should apply superresolution imaging (Suleiman, *elife*, 2013) or conventional immunogold electron microscopy to determine the precise subcellular distribution of the SD proteins. Are they located at the cell-cell junctions/SD precursors or are they retained in the cytoplasm, arguing for immaturity? How does plating density affect junction formation and subcellular distribution of SD proteins? The authors should also assess the expression of WT-1 in the isolated glomeruli.

Figure 2: how many OrgPods can be obtained from a single isolated glomerulus? How do the authors in Fig. 2a differentiate between partially and fully differentiated cell by phase contrast microscopy? Are OrgPods smaller than ciPods or were the images taken at different magnifications? In Fig. 2b and 3a, a $\Delta\Delta\text{Ct}$ analysis comparing to a normal human podocyte or glomerulus control would be much more informative. The magnification of the confocal images in Figs. 2b,c is too low to allow a detailed assessment of the actin cytoskeleton; in Fig. 2b, the synaptopodin staining does not appear to be along stress fibers. It would generally be better to show not only the merged images but show each channel separate in b/w. From the images provided it appears that the OrgPods are largely devoid of stress fiber and present with a strong cortical actin cytoskeleton, a hallmark of undifferentiated podocytes. This is at odds with the authors' claim that OrgPods are more mature than ciPods. Here a more detailed analysis of the actin cytoskeleton is required. The mRNA data for synpo should be

validated at the protein level by Western blotting. For Fig. 2d a positive control is needed. Data for ciPod should also be shown. What is the functional relevance of the albumin uptake assay; do the OrgPods express the neonatal Fc receptor? Do they express TRCP6 channels and do they produce VEGF, a hallmark of differentiated podocytes (Barlett, *Annu Rev Physiol* 2016)?

How do the transcriptome data in Figure 3 and Table 1 compare to the data by Sharmin et al., who showed that typical transcriptional profiles were well conserved among their podocytes generated in vitro as well as mouse and human podocytes in vivo? How do the authors define "superior" gene expression and how are their data "superior" to those of Sharmin and colleagues? The postulated superior gene expression of podocyte identity and maturity is at odds with the cytoskeletal data in Fig. 2. Fig. 3a - Please comment on the possible causes of the large variation in differentiated ciPod from different labs. Fig. 3g - Protein level data would be more compelling than transcript level data. Figure 3h - It would be informative to see how an unrelated cell type scores, e.g. muscle, and how normal human podocytes or glomeruli score to show validation of this scoring system.

Figure 4 The author propose that MAFB/Kreisler deficient organoids recapitulate an anticipated podocyte phenotype, which is based solely on changes in mRNA expression for a small set of genes relevant for podocytes. From this limited analysis and the observation of similar change in gene expression in MAFB/Kreisler mutant mice (Ref. 34) they conclude that human iPSC kidney organoids can accurately model podocytopathies. As it stands, this far-reaching conclusion is not supported by compelling data. Podocytopathies are characterized by foot process effacement, proteinuria, and loss of podocytes (through detachment or cell death). To explore the relevance of their model, the authors should, at a minimum, test if they can find signs of foot process effacement and or podocyte loss in the MAFB/Kreisler mutant organoids. A more elegant approach would be to graft MAFB/Kreisler mutant organoids on the chorioallantoic membrane and compare the histopathology of these transplanted mutant organoids with the published histopathology of the kidneys in MAFB/Kreisler knockout kidneys. For Fig. 4d it would be helpful to see a band for a control gene like GAPDH.

Figure 5 - do the glomerular endothelial cells in the transplanted organoids express Tie-2? As glomerular cell-cell communication is critical for the development and maintenance of the glomerular filtration barrier (Barlett, *Annu Rev Physiol* 2016), the authors could consider analyzing the expression VEGF, which is required for maintaining filtration barrier function. Are the endothelial cells of human or chick origin? It would indicate if there are precursors in the organoids that need the proper environment to create the vasculature that is shown, or alternatively if the chick cells are vascularizing the human tissue. The images in fig. 5e are hard to interpret; it is not possible to see what the authors are claiming in the text.

Does the transplantation improve maturation of proximal and distal tubules and does it promote collecting duct formation?

Figure 6 - as mentioned above, the filtration barrier including the foot process morphology and the cell-cell junction look less mature than those of organoids transplanted under the kidney capsule (Sharmin et al., Xinaris et al.). What is the reason for this discrepancy? Can the author increase podocyte maturation by adding VEGF? Quantification would strengthen the observation; if possible, include the frequency at which a GBM is formed, mature podocytes are observed, and fenestrated endothelium is observed. If rigorous quantification is not possible, adding a comment in the text would suffice.

Minor comments

As the authors claim to have developed an accurate model of podocytopathies, it would be appropriate to offer a more nuanced discussion of the recently reported glomerular filter on chip by Musah et al. (Ref. 46). Whereas it may be true that the iPSC-derived podocytes described by Musah et al. may not be well differentiated, as claimed by the authors, the organ-on-a-chip approach (Ref. 46) offers a more direct approach to study -in a quantitative fashion- the function of the glomerular filter in health and disease. An obvious experiment of great value that would markedly increase the novelty of the manuscript, would be to assess the organoid-derived OrgPods in the glomerular filter on chip setting and explore if they are truly "superior" to the iPSC-derived human podocytes employed by Musah and colleagues.

Considering the richness of functional data that have been obtained with murine and human podocyte cell culture systems over the last two decades, the statement in the abstract "Studies into podocyte biology and disease have been hampered by a paucity of in vitro models of this non-proliferative cell type" is not justified and should be removed.

Similarly, the statements at the beginning of the second paragraph of the introduction "Understanding podocytopathies has been hampered by the non-proliferative nature and architecturally constrained morphology of this cell type" and "To date, the accepted in vitro model of the human podocyte has been the temperature-sensitive SV40 conditionally immortalised podocyte line, which allows proliferation at 33°C and terminal differentiation at 37°C in vitro [12]." are incorrect and should be removed.

Reviewer #3 (Remarks to the Author):

Hale et al. report that glomeruli sieved from human induced pluripotent stem cell (iPSC)-derived kidney organoids exhibit more abundant levels of podocyte markers compared with an immortalized podocyte cell line. The authors also induced MAFB-deficient iPSCs toward a podocyte lineage, which showed reduced levels of several MAFB target genes. They further transplanted the wild-type kidney organoids into chick embryos, which demonstrated vascularization of human glomeruli.

While transplantation of iPSC-derived kidney tissues into chick embryos and generation of MAFB-deficient iPSC lines are unique, the majority of findings were already reported by Sharmin et al. (J Am Soc Nephrol 27:1778, 2016). In addition, foot processes and slit diaphragms, which are characteristic podocyte structures, do not appear to be as mature as those in the previous report. Furthermore, Freedman et al. reported the generation of podocalyxin-deficient podocytes using CRISPR-Cas9 technology (Nat Commun 10:1058, 2015). However, the authors do not refer to these published results. The authors should treat the published literature fairly and clarify their own advances over previous results.

Major comments

1. The majority of findings have already been reported by Sharmin et al. (J Am Soc Nephrol 2016), including staining of typical podocyte markers, comprehensive gene expression analysis, isoform switches of collagen and laminin, and upon transplantation, glomerular vascularization, dual basement membrane formation, and fenestration of endothelial cells. In addition, Freedman et al. reported the characterization of podocalyxin-deficient podocytes generated using CRISPR-Cas9 technology (Nat Commun 10:1058, 2015). The authors should introduce published literature in the introduction section, and then clarify the advances and/or consistency of their own data in the following result and discussion sections.

2. The authors performed gene expression analysis using whole glomeruli, but not purified MAFB-GFP podocytes, while Sharmin et al. already showed microarray analysis of purified NPHS1-GFP podocytes (J Am Soc Nephrol 2016). It is critical to show whether podocytes induced by the authors' method are more mature than those described in the previous report and/or more similar to human podocytes in vivo.

3. Electron microscopic data of foot processes (secondary processes) and slit diaphragms (Fig. 1C, Fig. 6) are not convincing to claim maturity. While the authors should present more magnified pictures, the complexity of foot processes does not appear to be high and the effect of transplantation is unclear due to a lack of comparable data without transplantation. Slit diaphragms do not seem to be localized to the basal side of processes, as was achieved in the report by Sharmin et al. (J Am Soc Nephrol 2016). In support of this concern, NPHS1, a major component of slit diaphragms, is not localized close to the laminin/collagen+ basement membrane (Fig. 1e) in vitro, or after transplantation (Fig. 5d, e). Such immaturity could result either from protocol differences or observation at a shorter time-point after transplantation (5 days in chick vs 20 days in mice). Taking these points into account, the authors should clarify their advances over previously published results. In addition, organoids used for transplantation were day 7+7 and harvested at day 7+12, whereas organoids at day 7+11 in vitro were not suitable for podocyte analysis (day 7+18 were used for Fig. 1-4). The authors should explain why they could not wait until day 7+18, as this is likely related to limitations of the chick transplantation system.

4. The authors showed reduction of only three target genes in MAFB-deficient podocytes, and failed to show any key structural abnormalities, such as foot process effacement. Transplantation experiments were not performed using MAFB-deficient podocytes. Thus, it is not convincing at present that the established cell lines combined with the authors' protocol are useful for "accurate modeling of podocytopathy".

Minor comments

1. Organoid numbers used for transplantation, as well as actual numbers of vascularized glomeruli against those examined, should be described.

2. Examined numbers of MAFB heterozygous and homozygous clones, as well as numbers of organoids examined per clone, should be described. Data from multiple clones should be presented in Fig 4e.

Reviewer #1

Comment: This is an exciting paper describing characteristics of podocytes and glomeruli isolated from organoids that have been generated from human iPS cell lines. There are two major advances reported:

- 1) Description of a technique to generate whose transcriptional profile more faithfully recapitulates the profile found in human podocytes isolated from tissues, thus providing a new model to explore podocyte biology.
- 2) Description of a technique to develop vascularized glomeruli using a chick allantoic (CAM) procedure. The vascularized glomeruli that form in this setting provide a new technique to study podocyte development, explore human genetic podocytopathies in the lab and provide access to a system to study parietal epithelial cells.

Response: *We thank Reviewer #1 for this positive feedback. In response to the comments from other reviewers, we have removed the data on CAM co-culture. However, we have added substantial additional data on the application of organoid derived glomeruli for disease modelling using patient derived iPSC and for toxicity screening.*

Comment: Some discussion re: limitations of the comparison of RNAseq data from historical, published samples would be helpful. Some additional discussion is valuable e.g. discussion of clonal differences between immortalized podocyte cell lines should be considered.

Response: *This reviewer and other reviewers suggested that we should be providing comparisons between the transcriptional profiling performed here and previous studies in which some level of profiling has been performed. Indeed, we were also asked to compare our data with expression data from mouse and from studies in which data is provided as microarrays. It is not possible to do this accurately for the following reasons: 1) It is not feasible to compare levels of expression between species. 2) It is problematic to compare RNAseq data to microarray data. 3) It is difficult to compare RNAseq data between experiments unless the appropriate information has been uploaded to GEO. Having said that, we have gone to considerable lengths in this revised manuscript to better compare our analysis with previous studies. We would note that this is now the most comprehensive transcriptional analysis available of the human immortalised podocyte lines both in a differentiated and undifferentiated state. We have included in our analysis a direct comparison with a prior dataset for this line. We have also returned to the analysis of podocytes presented within Sharmin et al. This data was not readily comparable due to the transcriptional profiling approach used. We have also directly compared our OrgGlom data with that published in Kim et al, 2017 which was generated using a different protocol for the differentiation of human pluripotent stem cell lines. The data made available via GEO from this study included only raw counts. The raw data was not available, nor were any details around normalisation. However, we evaluated the level of expression of a wide set of podocyte genes normalised for available GAPDH expression levels and compared this directly to our own data presented in the same manner. This is now included as Supplementary*

Figure 4. What this shows is considerably improved podocyte identity using our own method. It also highlights significant sample to sample variation in the Kim et al data.

Comment: Some discussion of use of primary podocytes for studies that might demonstrate a more faithful recapitulation of podocyte expression profile (an alternate to conditionally immortalized podocytes) could be included.

Response: *We now refer to the challenges of primary podocytes for disease modelling in the introduction.*

Comment: Some comment about limitations of organoid technique – i.e. large scale proteomics or epigenetic studies will not currently be possible with organoid derived-gloms but are possible with immortalized human podocytes

Response: *We do not agree that organoids are any more limited for proteomic or epigenetic studies than immortalised podocytes. Indeed, we now present substantial data showing a capacity to derive detailed proteomic data on glomeruli from organoids. The concept that this will be better with immortalised lines because these are immortal is counterbalanced by the fact that organoids are derived from a pluripotent stem cell line which can be grown indefinitely and then differentiated into any cell type. We believe that the data we now provide shows this approach to be robust and superior to either primary or immortalised lines grown in 2D.*

Comment: Some discussion (even if speculative) about reason for lack of vascularization in organoids would be valuable

Response: *We have previously shown that endothelial cells are present within organoids, but the level to which they invest the glomeruli is poor. As we have removed the chicken data from this version of the manuscript, this discussion is probably not warranted. However we would note that we have now included a formal temporal evaluation of gene expression within the glomeruli isolated from organoids and show that intact OrgGlom begin to show expression of a number of markers suggestive of a degree of mesangial and endothelial cell presence within these structures.*

Comment: There is quite a lot known about the switch of isoforms of laminins during development – appropriate references should be cited

Response: *In response to this comment, we have included a more thorough discussion of the process of laminin and collagen switching in the glomerulus and have now performed an extensive proteomic analysis of the matrisome present in organoid-derived glomeruli.*

Comment: Fig 6 – any chance to do immunogold to see slit diaphragms? How frequently are slit diaphragms found?

Response: *We did not perform immunogold staining to look for slit diaphragms. As previously mentioned we have removed the CAM data from this manuscript which was in Figure 6.*

Reviewer #2

Comment: The current manuscript is the beginning of a nice body of work but is incomplete. Additionally, the claims are overstated given the evidence presented. Overall, the manuscript lacks novelty, because key findings, such as the vascularization of transplanted organoids, has already been published by two other groups (Ref. 45 and Xinaris et al, JASN, 2016: Functional Human Podocytes Generated in Organoids from Amniotic Fluid Stem Cell). The chicken chorioallantoic membrane transplant model data establishes an alternative to previously published mouse and rat transplant models. However, it does not fit with the rest of the manuscript unless data is presented that demonstrates its utility in human disease modeling. At a minimum, it would be interesting to see the histopathology of transplanted MAFB^{-/-} organoids.

Response: *This criticism was largely targeted at the data included in this manuscript around the vascularisation of organoids when cultured on the chicken chorioallantoic membrane. This was the first report taking this approach to vascularisation. In response to this reviewer's comment that this data does not fit with the rest of the manuscript, it has been removed. The comment that this manuscript lacked novelty has been substantially addressed via the inclusion of an extensive body of additional data both around the characterisation of the cells themselves and the application of organoid-derived glomeruli for disease modelling and toxicity screening.*

Comment: Lack of proper controls. To claim that a new in vitro system is superior than the current "gold standard" in modeling an in vivo derived human glomerulus or podocytes, the new system must be compared, at all levels, to the in vivo counterpart and to the true gold standard in the field, even if that is an in vitro system with differentiated mouse podocytes. In this particular case, the new system described needs to be compared to the immortalized mouse podocytes, which are the current gold standard, the immortalized human podocytes, which allow for species-specific comparisons to the new system, and the in vivo derived counterpart, which sets the bar against which the new system will be measured. In addition, it is well known in the stem cell field that iPSC derivatives are developmentally immature. This needs to be taken into consideration when interpreting the results. Ideally, there would be both fetal and adult control samples in the data presented.

Response: *We have taken the criticism of this reviewer very seriously, but we respectfully disagree with the statement that mouse podocytes represent a gold standard with which we should be comparing. It is not at all clear why, or even technically how, one would compare transcriptional profiles of human podocytes, whether from organoids or primary lines, with immortalised lines or in vivo glomeruli from mouse. Instead of this, we have now substantially extended our transcriptional profiling to include ciPods before and after differentiation, including data from another publication, and profiling of freshly sieved organoid-derived glomeruli in comparison to data from human isolated glomeruli. Using*

organoid glomeruli FACS isolated across organoid differentiation using a podocyte-specific reporter line, we also have a temporal profile of gene expression change within organoid glomeruli. We have also performed a complete proteomic analysis of the matrisome of organoid glomeruli and compared this to previous data from immortalised lines. While the reviewer argues that this may be an inferior model as the glomeruli are developmentally immature, we show more evidence for appropriate laminin and collagen proteins in the matrisomes of sieved organoid glomeruli than any line, primary or immortalised, cultured in 2D. This improved analysis strongly supports our claim that iPSC-derived podocytes, and even more so iPSC-derived organoid glomeruli are a superior model for the analysis of podocyte biology.

Comment: (To) provide compelling support for their claim that OrgPods display increased differentiation, the authors should compare the OrgPods not only to the human immortalized cell line described by Saleem and colleagues (Ref. 12), but also to other podocyte cell culture systems described by many investigators, including iPSC-derived podocytes (Song et al. PLoS One, 2012; Ciampi et al. Stem Cells 2016), human primary podocytes from adult normal kidney (Li et al. JASN ,2017, Yang et al, Exp Cell Res 2017, Mundel et al. JASN 1997), human podocyte cells lines established from urine (Sakairi, AJPhysiol 2010), as well as differentiated mouse podocyte cell lines (Mundel et al. Exp Cell Res 1997). Such a study would go a long way when trying to show “superior” differentiation of OrgPods. The results of such a comprehensive study would also be of great important for the field, especially if such studies could confirm the postulated higher degree of differentiation of OrgPods. As a final point on proper controls, it is also well known in the field that structure affects function. The authors themselves even present supportive data in Figure 4C. This fact needs to be taken into consideration when choosing the proper controls and interpreting results.

A major advantage of podocyte cell lines is the possibility to grow up large batches of cells. Does the approach described here enable the generation of large enough quantities of OrgPods from organoid-derived isolated glomeruli that are required for biochemical and functional studies?

Response: *As a result of this reviewers argument that the requirement by the podocyte for appropriate 3D cell-cell interactions has prompted us to extend our analyses to include a comparison of organoid-derived primary podocytes with intact organoid-derived glomeruli. These can be readily sieved and cultured as a 3D mass for a number of days without apparent loss of MAFB expression, In contrast, MAFB expression is lost within hours of plating podocytes. As a result, the message of our manuscript has now very much shifted to the value of the glomeruli as a whole rather than the podocytes alone. There are certainly enough glomeruli within an organoid for proteomic analyses, as we now show. It is also possible to seed individual glomeruli for toxicity screening, as we also show. Hence, there is no apparent barriers to biochemical or functional analyses. In response to the comment that we are using only one differentiation protocol, we have no intention of extensively comparing every other protocol reported for the generation of glomerular like structures / podocytes using iPSC. What we have provided here is an extensive dataset characterising our own protocol. A comparative level of characterisation has never been provided for any other iPSC-derived model and yet these have been published with claims of podocyte identity.*

Comment: The title of the manuscript suggests that the focus is on modeling human podocytopathies with patient iPSCs. To explore this hypothesis, the authors generate iPSCs lacking the transcription factor MAFB/Kreisler, which when deleted in mice, lead to FSGS-like disease and postnatal lethality. The authors then assess changes in gene expression resulting from the deletion of MAFB/Kreisler and conclude that the observed changes are relevant to human pathology. This is somewhat surprising, as to this date, neither genetic nor clinical data have been published that would link MAFB/Kreisler to human podocytopathies. Why engineer a MAFB mutant iPSC line instead of generating an iPSC line from a patient with podocytopathy? Without human-relevant data, e.g. knockin of a human podocytopathy gene mutation or studying a patient-derived iPSC line, followed by the analysis of the resulting phenotype, the title of the manuscript and the conclusion of the abstract are not supported by data.

Response: *The targeting of the MAFB locus was performed in order to generate a robust fluorescent tag for the identification and isolation of podocytes. In this revised manuscript, we use this line to good effect to facilitate and extensive temporal analysis of podocyte maturation within organoid glomeruli. As this reviewer was not convinced that this was a credible disease model, we have now generated an iPSC line from a patient with congenital nephrotic syndrome in which genome sequencing has definitively identified the nature of the mutations present. We clearly show aberrant protein levels and localisation of key podocyte proteins in glomeruli generated using this diseased line. We believe this is strong evidence that podocytopathies are amendable to analysis using iPSC-derived organoid glomeruli.*

Comment: The manuscript also contains a set of studies using organoid transplantation onto chicken chorioallantoic membranes. In these experiments, the authors show that the organoid transplantation results in glomerular vascularization, filtration barrier assembly and slit diaphragm maturation. The results of these studies are neither new nor surprising. Similar grafting studies in rats and mice have already been published (Xinaris et al, JASN, ref. 46 Sharmin et al.) and both studies showed that iPSC-derived glomeruli in the transplanted organoids exhibit many morphologic features of glomeruli in vivo, including a well-developed GBM, highly organized podocyte foot processes and slit diaphragms, which appear more mature than those junctions shown in this study (Fig. 6c). The latter look more like the “ladder-like” junctions that are found between developing podocytes (Reeves et al., Lab Invest, 1978), confirming their immature character.

Response: *More negative comments were made by this reviewer about this aspect of the study. As noted previously, this has been removed from the revised manuscript.*

Comment: Figure 1 describes the isolation of capillary loop stage glomeruli from the organoids. These glomeruli are still developing and not fully mature. Is it possible to isolate (more) mature glomeruli if the organoids are maintained for a longer time in culture before isolation or by treating them with VEGF?

Response: *We have now extensively characterised the transcriptional profile of organoid glomeruli across time in organoid culture and show with time the presence of endothelial and mesangial markers. We have not investigated the effect of addition of VEGF, however we have published data showing that the glomeruli within organoids are making VEGF (Van den*

Berg et al, 2018).

Comment: The authors report that a population of pure glomeruli with a diameter in the range of 50- 90 μm can be isolated (Fig. 1a_{ii}); this is markedly smaller than that of mature human glomeruli, which is about 200 μm . What is the size distribution of these isolated glomeruli, how many glomeruli can be isolated from an individual organoid and how reproducible is the procedure? What percentage of the isolated glomeruli contain a Bowman's capsule versus being acapsular? How does the glomerular size change after isolation from chorioallantoic membrane transplants?

Response: *We have now quantified the diameter of sieved organoid glomeruli and compared this with fetal and adult glomeruli. We have not examined size change after CAM co-culture as this data is no longer included in this manuscript.*

Comment: The TEM analysis (Fig. 1c) should be supplemented with high resolution images of the podocyte (foot) processes and cell-cell junctions. The upregulation of nephrin mRNA expression (Fig. 1d) should be confirmed at the protein level by Western blotting.

Response: *We have now included high resolution images of the podocyte processes in the TEM analysis. We have also shown nephrin protein expression via immunofluorescence in whole organoids and isolated glomeruli. In particular we have shown appropriate co-localisation with Neph1 in glomeruli.*

Comment: The analysis of GBM components in Fig. 1e should include the analysis of the mature alpha 3, 4 and 5 chain of Type IV collagen. The confocal images of the podocyte proteins are not very informative in the present version. The authors should apply superresolution imaging (Suleiman, *elife*, 2013) or conventional immunogold electron microscopy to determine the precise subcellular distribution of the SD proteins. Are they located at the cell-cell junctions/SD precursors or are they retained in the cytoplasm, arguing for immaturity? How does plating density affect junction formation and subcellular distribution of SD proteins? The authors should also assess the expression of WT-1 in the isolated glomeruli.

Response: *In reference to the analysis of GBM components, we have now provided substantive data detailing the abundance of GBM components and associated proteins via mass spectrometry of organoid glomeruli. To demonstrate the appropriate polarity of podocyte protein in organoid glomeruli we now show appropriate basal co-localisation of Neph1 with Neph1 and apical expression of Podocalyxin.*

Comment: Figure 2: how many OrgPods can be obtained from a single isolated glomerulus? How do the authors in Fig. 2a differentiate between partially and fully differentiated cell by phase contrast microscopy? Are OrgPods smaller than ciPods or were the images taken at different magnifications? In Fig. 2b and 3a, a delta delta Ct analysis comparing to a normal human podocyte or glomerulus control would be much more informative. The magnification of the confocal images in Figs. 2b,c is too low to allow a detailed assessment of the actin

cytoskeleton; in Fig. 2b, the synaptopodin staining does not appear to be along stress fibers. It would generally be better to show not only the merged images but show each channel separate in b/w. From the images provided it appears that the OrgPods are largely devoid of stress fiber and present with a strong cortical actin cytoskeleton, a hallmark of undifferentiated podocytes. This is at odds with the authors' claim that OrgPods are more mature than ciPods. Here a more detailed analysis of the actin cytoskeleton is required. The mRNA data for synpo should be validated at the protein level by Western blotting. For Fig. 2d a positive control is needed. Data for ciPod should also be shown. What is the functional relevance of the albumin uptake assay; do the OrgPods express the neonatal Fc receptor? Do they express TRCP6 channels and do they produce VEGF, a hallmark of differentiated podocytes (Barlett, Annu Rev Physiol 2016)?

Response: *With respect to the specific questions about numbers, this information is now provided and clearly illustrates the large number of individual OrgGloms available from a single differentiation experiment. Indeed, a single differentiation would be sufficient to seed 50 x 96 well plates for screening purposes. We have also now shown the progressive migration of OrgPods from organoid glomeruli to give an indication of cell number. However, as the main conclusion of our paper argues that 3D glomeruli are more favourable to 2D culture we did not analyse specific numbers per glomeruli. The phrasing of 'partially differentiated' and 'fully differentiated' in respect to the ciPods is now clarified in the figure legend, 'partially' being mid-way through the published time frame of differentiation for these cells, 'fully' being upon completion. Scale bars are included in this figure which address the reviewers query with regard to cell size. We do not have access to human kidney tissue, therefore we are unable to acquiesce to the request for this as a control for qPCR analysis. We disagree that the original confocal images did not show co-localisation of synaptopodin along actin stress fibres, as this is shown clearly in figure 2c and indicated with an arrow. However, we have now included new confocal imaging showing appropriate synaptopodin distribution in both glomeruli and OrgPods in conjunction with F-actin staining, depicting the extensive stress fibres. In addition, as requested we have shown the expression of the neonatal Fc receptor in OrgPods, and shown their responsiveness to insulin at both the level of actin reorganisation and GLUT4 translocation. As previously stated, we have published data showing that the podocytes within organoids are making VEGF (Van den Berg et al, 2018).*

Comment: How do the transcriptome data in Figure 3 and Table 1 compare to the data by Sharmin et al., who showed that typical transcriptional profiles were well conserved among their podocytes generated in vitro as well as mouse and human podocytes in vivo? How do the authors define "superior" gene expression and how are their data "superior" to those of Sharmin and colleagues? The postulated superior gene expression of podocyte identity and maturity is at odds with the cytoskeletal data in Fig. 2. Fig. 3a - Please comment on the possible causes of the large variation in differentiated ciPod from different labs. Fig. 3g - Protein level data would be more compelling than transcript level data. Figure 3h - It would be informative to see how an unrelated cell type scores, e.g. muscle, and how normal human podocytes or glomeruli score to show validation of this scoring system.

Response: *Respectfully, the transcriptional profiling presented in Sharmin et al was limited and it was not possible to directly compare our data with that data. We have already*

addressed the fact that OrgPods do in fact have an appropriate actin cytoskeleton, which was an inaccurate observation. The variation seen between ciPods from different labs is in actual fact a small change. This is highlighted by the close congruence observed in gene expression levels shown in the heatmaps in which each biological replicate from each laboratory is shown. We agreed with the reviewers comment that protein level data would be more compelling to examine collagen and laminin isoforms and hence have now included an in depth analysis of the OrgGlom matrixome. This has never before been performed on any iPSC/ES-derived glomerular population. We disagree with the suggestion to include unrelated cell types into the scoring system in Figure 3G. This was included to depict the similarity between OrgGloms and human glomerular tissue at the gene level, which has now also been substantiated by comparing to human glomeruli at the protein level. The addition would of unrelated cell types would be no additional benefit.

Comment: Figure 4. The author propose that MAFB/Kreisler deficient organoids recapitulate an anticipated podocyte phenotype, which is a based solely on changes in mRNA expression for a small set of genes relevant for podocytes. From this limited analysis and the observation of similar change in gene expression in MAFB/Kreisler mutant mice (Ref. 34) they conclude that human iPSC kidney organoids can accurately model podocytopathies. As it stands, this far-reaching conclusion is not supported by compelling data. Podocytopathies are characterized by foot process effacement, proteinuria, and loss of podocytes (through detachment or cell death). To explore the relevance of their model, the authors should, at a minimum, test if they can find signs of foot process effacement and or podocyte loss in the MAFB/Kreisler mutant organoids. A more elegant approach would be to graft MAFB/Kreisler mutant organoids on the chorioallantoic membrane and compare the histopathology of these transplanted mutant organoids with the published histopathology of the kidneys in MAFB/Kreisler knockout kidneys. For Fig. 4d it would be helpful to see a band for a control gene like GAPDH.

Response: *As discussed addressing a previous comment, we have now generated an iPSC line from a patient with congenital nephrotic syndrome in which genome sequencing has definitively identified the nature of the mutations present as an alternative to the MAFB homozygous line. We have also added in the GAPDH band into the figure as requested.*

Comment: Figure 5 - do the glomerular endothelial cells in the transplanted organoids express Tie-2? As glomerular cell-cell communication is critical for the development and maintenance of the glomerular filtration barrier (Barlett, Annu Rev Physiol 2016), the authors could consider analyzing the expression VEGF, which is required for maintaining filtration barrier function. Are the endothelial cells of human or chick origin? It would indicate if there are precursors in the organoids that need the proper environment to create the vasculature that is shown, or alternatively if the chick cells are vascularizing the human tissue. The images in fig. 5e are hard to interpret; it is not possible to see what the authors are claiming in the text. Does the transplantation improve maturation of proximal and distal tubules and does it promote collecting duct formation?

Response: *This question is not relevant as the data has been removed.*

Comment: Figure 6 - as mentioned above, the filtration barrier including the foot process

morphology and the cell-cell junction look less mature than those of organoids transplanted under the kidney capsule (Sharmin et al., Xinaris et al.). What is the reason for this discrepancy? Can the author increase podocyte maturation by adding VEGF? Quantification would strengthen the observation; if possible, include the frequency at which a GBM is formed, mature podocytes are observed, and fenestrated endothelium is observed. If rigorous quantification is not possible, adding a comment in the text would suffice.

Response: *This question is not relevant as the data has been removed.*

Minor comments: As the authors claim to have developed an accurate model of podocytopathies, it would be appropriate to offer a more nuanced discussion of the recently reported glomerular filter on chip by Musah et al. (Ref. 46). Whereas it may be true that the iPSC-derived podocytes described by Musah et al. may not be well differentiated, as claimed by the authors, the organ-on-a-chip approach (Ref. 46) offers a more direct approach to study -in a quantitative fashion- the function of the glomerular filter in health and disease. An obvious experiment of great value that would markedly increase the novelty of the manuscript, would be to assess the organoid-derived OrgPods in the glomerular filter on chip setting and explore if they are truly “superior” to the iPSC-derived human podocytes employed by Musah and colleagues.

Response: *This manuscript contains an extensive dataset focussed on characterising the validity of iPSC-derived glomeruli as a model for the in vivo situation. Our intent was not to evaluate the organ-on-a-chip approach. Indeed, we believe we have shown utility of organoid-derived glomeruli in and of themselves.*

Comment: Considering the richness of functional data that have been obtained with murine and human podocyte cell culture systems over the last two decades, the statement in the abstract “Studies into podocyte biology and disease have been hampered by a paucity of in vitro models of this non-proliferative cell type” is not justified and should be removed.

Similarly, the statements at the beginning of the second paragraph of the introduction “Understanding podocytopathies has been hampered by the non-proliferative nature and architecturally constrained morphology of this cell type” and “To date, the accepted in vitro model of the human podocyte has been the temperature-sensitive SV40 conditionally immortalised podocyte line, which allows proliferation at 33°C and terminal differentiation at 37°C in vitro [12].” are incorrect and should be removed.

Response: *We accept this criticism and have reworded the opening comments. The intent of the manuscript is not to negate the information now available as a result of immortalised models in both mouse and human but to highlight the improvements provided by an iPSC-derived podocyte source. Indeed, having extended our dataset significantly since the prior submission, we can now strongly show the benefits of leaving the organoid podocytes in the 3D glomerular confirmation and the added utility of culture in this format for disease modelling and drug screening. The large number of glomeruli able to be generated in this way, the renewable nature of the cell lines and the capacity to derive such lines from patients without biopsy are compelling argument for the adoption of this approach.*

Reviewer #3

Comment: Hale et al. report that glomeruli sieved from human induced pluripotent stem cell (iPSC)-derived kidney organoids exhibit more abundant levels of podocyte markers compared with an immortalized podocyte cell line. The authors also induced MAFB-deficient iPSCs toward a podocyte lineage, which showed reduced levels of several MAFB target genes. They further transplanted the wild-type kidney organoids into chick embryos, which demonstrated vascularization of human glomeruli.

While transplantation of iPSC-derived kidney tissues into chick embryos and generation of MAFB-deficient iPSC lines are unique, the majority of findings were already reported by Sharmin et al. (J Am Soc Nephrol 27:1778, 2016). In addition, foot processes and slit diaphragms, which are characteristic podocyte structures, do not appear to be as mature as those in the previous report. Furthermore, Freedman et al. reported the generation of podocalyxin-deficient podocytes using CRISPR-Cas9 technology (Nat Commun 10:1058, 2015). However, the authors do not refer to these published results. The authors should treat the published literature fairly and clarify their own advances over previous results.

Response: *We have addressed these omissions by including a clear discussion of what has been previously reported. However, we would respectfully argue that this dataset, particularly after this extensive revision, is the most comprehensive characterisation of iPSC-derived podocytes / glomeruli ever provided and that the data presented around utility for disease modelling, particularly using patient-derived iPSC lines, and toxicity screening is novel.*

Comment: The majority of findings have already been reported by Sharmin et al. (J Am Soc Nephrol 2016), including staining of typical podocyte markers, comprehensive gene expression analysis, isoform switches of collagen and laminin, and upon transplantation, glomerular vascularization, dual basement membrane formation, and fenestration of endothelial cells. In addition, Freedman et al. reported the characterization of podocalyxin-deficient podocytes generated using CRISPR-Cas9 technology (Nat Commun 10:1058, 2015). The authors should introduce published literature in the introduction section, and then clarify the advances and/or consistency of their own data in the following result and discussion sections.

Response: *Addressed above. We now refer to and discuss this prior literature. We have also provided substantial additional data that clearly differentiates this study from previous datasets.*

Comment: The authors performed gene expression analysis using whole glomeruli, but not purified MAFB-GFP podocytes, while Sharmin et al. already showed microarray analysis of purified NPHS1-GFP podocytes (J Am Soc Nephrol 2016). It is critical to show whether podocytes induced by the authors' method are more mature than those described in the previous report and/or more similar to human podocytes in vivo.

Response: *This is incorrect. In the original manuscript, we profiled podocytes from sieved glomeruli. We have now also profiled podocytes isolated using FACS and also profiled whole*

glomeruli after sieving from organoids. We also provide extensive proteomic analyses. The data provide in Sharmin et al was microarray data. What we present is RNA-seq. Hence, our profiling is more comprehensive.

Comment: Electron microscopic data of foot processes (secondary processes) and slit diaphragms (Fig. 1C, Fig. 6) are not convincing to claim maturity. While the authors should present more magnified pictures, the complexity of foot processes does not appear to be high and the effect of transplantation is unclear due to a lack of comparable data without transplantation. Slit diaphragms do not seem to be localized to the basal side of processes, as was achieved in the report by Sharmin et al. (J Am Soc Nephrol 2016). In support of this concern, NPHS1, a major component of slit diaphragms, is not localized close to the laminin/collagen+ basement membrane (Fig. 1e) in vitro, or after transplantation (Fig. 5d, e). Such immaturity could result either from protocol differences or observation at a shorter time-point after transplantation (5 days in chick vs 20 days in mice). Taking these points into account, the authors should clarify their advances over previously published results. In addition, organoids used for transplantation were day 7+7 and harvested at day 7+12, whereas organoids at day 7+11 in vitro were not suitable for podocyte analysis (day 7+18 were used for Fig. 1-4). The authors should explain why they could not wait until day 7+18, as this is likely related to limitations of the chick transplantation system.

Response: *This comment refers to the CAM co-culture which has been removed from the manuscript.*

Comment: The authors showed reduction of only three target genes in MAFB-deficient podocytes, and failed to show any key structural abnormalities, such as foot process effacement. Transplantation experiments were not performed using MAFB-deficient podocytes. Thus, it is not convincing at present that the established cell lines combined with the authors' protocol are useful for "accurate modeling of podocytopathy".

Response: *Addressed above we now show disease modelling of congenital nephrotic syndrome using an iPS line from a NPHS1 compound heterozygous patient from which organoids were generated and glomeruli isolated.*

Minor comments

1. Organoid numbers used for transplantation, as well as actual numbers of vascularized glomeruli against those examined, should be described.

No longer required.

2. Examined numbers of MAFB heterozygous and homozygous clones, as well as numbers of organoids examined per clone, should be described. Data from multiple clones should be presented in Fig 4e.

No longer required. Disease modelling now performed using CNS patient line.

Reviewers' comments:

Reviewer #1 (Remarks to the Author):

The authors have provided thoughtful responses to my questions.

A few comments outlined below:

1. While the addition of the matrisome data is valuable and highlights the utility of iPSC-derived organoids for proteomic analysis, larger and more complex proteomic studies that may involve biochemical assays and/or a variety of conditions still require very large starting material amounts that will only be possible using large-scale cell based methods. Some comment/caveat could be included in the discussion. Podocyte or heterologous cell lines will still remain the only option for some of these studies currently.
2. The addition of a model from iPSC derived organoids from a patient with NPHS1 mutation is valuable. The demonstration of reduced expression of NPHS1 and Podocin does demonstrate utility (especially as the cells come directly from a patient without need for genetic manipulation). Some description of limitations should be included as well – i.e. given lack of full differentiation, it is unlikely that it will be possible to see broadening of foot processes or lack of a slit diaphragm – as is typically observed in biopsies from patients with congenital nephrosis of the Finnish variety. A comment recognizing this limitation could be included in the paper.
3. The authors have removed the allantois data – however, it would still be valuable to know whether or not SDs form at all in the organoids – immunoEM should be possible to look at a slit diaphragm protein such as Nephrin. Even if the SDs do not form – this would be valuable information for the field.
4. There was a recent pre-publication on bioRx from the Humphrey's group describing transcriptomic comparisons between 2 organoid protocols and adult human kidney – this pre-print could be cited.

Reviewer #2 (Remarks to the Author):

The revised manuscript is greatly improved, and the authors have done an amazing job to adequately address all of my key concerns.

The characterization of OrgGloms generated from a congenital nephrotic syndrome patient with compound heterozygous NPHS1 mutations is a highlight of the revised paper!

The observation that isolated organoid-derived glomeruli synthesise mature components of the human glomerular basement membrane is another highlight of the revised manuscript.

Finally, the image analysis of the cytoskeleton is markedly improved.

Remaining minor issue

The second sentence of the abstract "Due to the non-proliferative nature of primary podocytes studies into human podocytopathies have relied upon the use of immortalised human cell lines." is an

overstatement and should be removed from the final manuscript.

Reviewer #3 (Remarks to the Author):

Reviewer #3

Response: We have addressed these omissions by including a clear discussion of what has been previously reported. However, we would respectfully argue that this dataset, particularly after this extensive revision, is the most comprehensive characterisation of iPSC-derived podocytes / glomeruli ever provided and that the data presented around utility for disease modelling, particularly using patient-derived iPSC lines, and toxicity screening is novel.

New comment:

The authors have deleted the electron microscopic data upon transplantation, emphasizing the usefulness of the sieved glomeruli from in vitro organoids. However, much of the additional new information is still preliminary.

Regarding patient-derived iPSCs, data to support the reproducibility of the experiments (i.e. numbers of clones established and tested, numbers of independent experiments per clone, information and suitability of iPSCs used as a control) are lacking. Western blots for NEPRHIN, PODOCIN, and NEPH1 should be presented to show the specific reductions in the first two proteins. Importantly, detailed histological abnormalities (i.e. apico-basal distribution of slit diaphragm-related proteins, electron microscopic analysis of foot processes and slit diaphragms) should be presented. Podocytes are characterized by a unique morphology that is tightly associated with their functions and diseases, and it is known that NPHS1 mutations result in the absence of the slit diaphragms (Ruotsalainen et al Am J Pathol 157, 1905, 2000). These data are therefore critical to the argument that the glomeruli sieved from the in vitro organoids model podocytopathies.

Regarding the toxicity experiments, the glomeruli appear to simply die after drug treatment. Are the glomeruli more sensitive to the drug than other lineages in the organoids, such as renal tubules? Again, morphological abnormalities, such as foot process effacement, should also be examined.

Original Comment : The authors performed gene expression analysis using whole glomeruli, but not purified MAFB-GFP podocytes, while Sharmin et al. already showed microarray analysis of purified NPHS1-GFP podocytes (J Am Soc Nephrol 2016). It is critical to show whether podocytes induced by the authors' method are more mature than those described in the previous report and/or more similar to human podocytes in vivo.

Response: This is incorrect. In the original manuscript, we profiled podocytes from sieved glomeruli. We have now also profiled podocytes isolated using FACS and also profiled whole glomeruli after sieving from organoids. We also provide extensive proteomic analyses. The data provide in Sharmin et al was microarray data. What we present is RNA-seq. Hence, our profiling is more comprehensive.

New comment:

Regarding the new Figure 5: it is unclear how the samples for RNA-seq were harvested at each time point. At a first glance, all the samples appear to represent podocytes, because MAFB-GFP iPSCs were used. However, the d7+10 sample is FACS-purified podocytes, while the d7+19 sample is whole glomeruli (d7+14 remains unclear). If this is the case, it is not appropriate to compare podocytes with whole glomeruli during the time-course experiments, given that whole glomeruli contain non-podocyte lineages. Comparison of purified podocytes at each stage would be more straightforward and informative.

In addition, it is not clear if endothelial cells are actually present within the glomeruli or if the cells surrounding the glomeruli are contaminated in the sieved samples. These possibilities could be easily

tested by staining the OrgGloms with endothelial markers. Notably, there were few endothelial cells in the glomeruli in the authors' previous study (Takasato et al., Nature 2015).

Importantly, although the authors compare their organoids with 2D podocytes and immortalized cell lines, a comparison with human samples in vivo is lacking. Figure 5F and Fig 1G do not address the critical point of how similar the OrgGloms are to their counterparts in vivo. Fig 4F only shows the presence or absence of each ECM and does not show the relative expression levels, and even in this setting, the OrgGloms look different from those in vivo, including in the expression of COL4A4 (add COL4A4 data in Fig. 4D and F). The new phrase added to the abstract "comparable with human glomeruli in vivo" is therefore misleading and should be deleted.

Original Comment: The authors showed reduction of only three target genes in MAFB-deficient podocytes, and failed to show any key structural abnormalities, such as foot process effacement. Transplantation experiments were not performed using MAFB-deficient podocytes. Thus, it is not convincing at present that the established cell lines combined with the authors' protocol are useful for "accurate modeling of podocytopathy".

Response: Addressed above we now show disease modelling of congenital nephrotic syndrome using an iPS line from a NPHS1 compound heterozygous patient from which organoids were generated and glomeruli isolated.

New comment:

Please see the above comment. Histological examination is required, even though the cell line has been changed from MAFB-GFP to CNS iPSCs.

Original comment: Examined numbers of MAFB heterozygous and homozygous clones, as well as numbers of organoids examined per clone, should be described. Data from multiple clones should be presented in Fig 4e.

Response: No longer required. Disease modelling now performed using CNS patient line.

New comment

Please see the above comment. Data to support reproducibility (such as clonal variations, sample numbers per clone, type and suitability of the control iPSCs) are required, even though the cell line has been changed from MAFB-GFP to CNS iPSCs.

Reviewer #1 (Remarks to the Author):

The authors have provided thoughtful responses to my questions.

A few comments outlined below:

1. While the addition of the matrisome data is valuable and highlights the utility of iPSC-derived organoids for proteomic analysis, larger and more complex proteomic studies that may involve biochemical assays and/or a variety of conditions still require very large starting material amounts that will only be possible using large-scale cell based methods. Some comment/caveat could be included in the discussion. Podocyte or heterologous cell lines will still remain the only option for some of these studies currently.

Response: We were readily able to generate the matrisome data from sieved glomeruli. On average, we were able to obtain 4mg of protein from a single OrgGlom isolation. This is from a standard diff of x24 organoids at 200k size (one 6-well plate of organoids). This was far in excess of what was required for matrisome characterisation or Western blotting. For a Western blot you typically load 10-100ng protein, and for the identification of proteins in complex mixtures by Mass Spectrometry the amount of protein should be upwards of 10ug. Importantly, this approach makes it possible to perform proteomics from any patient and not simply be restricted to an immortalised line with suboptimal gene expression and morphology. Reviewer 1 seems to be suggesting that a single isolated OrgGlom within a well of a 96 well screen may not be sufficient for proteomic / matrisome analysis. While we do not demonstrate this here, new generation Mass Spec approaches such as are now available, including the Q Exactive platform (Kelstrup et al, 2018; Hoyer et al, 2018) , are able to perform proteomics, phosphoproteomics and cell surface proteomics on a tissue biopsy or something as small as a single cancer organoid, hence we do not believe that this will be a barrier in the future. We would also note that screening is more likely to involve rapid high content imaging outputs and not proteomics in the first instance. What is critical is that what we could provide using isolated organoid derived glomeruli is a major advance over anything currently available for the analysis of distinct patient podocytopathies.

2. The addition of a model from iPSC derived organoids from a patient with NPHS1 mutation is valuable. The demonstration of reduced expression of NPHS1 and Podocin does demonstrate utility (especially as the cells come directly from a patient without need for genetic manipulation). Some description of limitations should be included as well – i.e. given lack of full differentiation , it is unlikely that it will be possible to see broadening of foot processes or lack of a slit diaphragm – as is typically observed in biopsies from patients with congenital nephrosis of the Finnish variety. A comment recognizing this limitation could be included in the paper.

Response: We accept that this is a model and not the real tissue. We also accept that while it may be able to model primary podocytopathies, it is not structurally mature enough for glomerular basement membrane diseases. However, we believe that we have clearly demonstrated that this is a far better model than anything currently available. We have included comments within the Discussion with respect to the limitations.

3. The authors have removed the allantois data – however, it would still be valuable to know whether or not SDs form at all in the organoids – immunoEM should be possible to look at a slit

diaphragm protein such as Nephlin. Even if the SDs do not form – this would be valuable information for the field.

Response: We do not anticipate the formation of mature slit diaphragms in OrgGloms. Indeed, no slit diaphragms form in 2D podocyte cultures. We did not perform immunoEM on the CAM cultures. The entire section on chicken CAM culture has been removed at the request of the reviewers and because we found this a much slower and more costly approach to generating models of human glomeruli. This approach is also never going to be applicable to high content screening, which we will be able to perform using isolated glomeruli.

4. There was a recent pre-publication on bioRx from the Humphrey's group describing transcriptomic comparisons between 2 organoid protocols and adult human kidney – this pre-print could be cited.

Response: We also have our own BioRX publication on single cell profiling (Combes et al) of organoids. Neither this nor Wu et al shows anything more than what we show here. Indeed, the single cell analysis is not as deep as the RNA Seq provided here. Nevertheless, we have referred to both datasets in the revised Discussion. We would add that single cell profiling is at present less sensitive with respect to transcripts detected and more expensive. Hence this would never be used as a disease modelling tool. By isolating the OrgGloms, we have purified down to the specific cell types of interest.

Reviewer #2 (Remarks to the Author):

The revised manuscript is greatly improved, and the authors have done an amazing job to adequately address all of my key concerns.

Response: We appreciate the positive feedback from this reviewer and are pleased that they have recognised the extensive additional data that has been provided.

The characterization of OrgGloms generated from a congenital nephrotic syndrome patient with compound heterozygous NPHS1 mutations is highlight of the revised paper!

Response: Thankyou

The observation that isolated organoid-derived glomeruli synthesise mature components of the human glomerular basement membrane is another highlight of the revised manuscript.

Response: Thankyou

Finally, the image analysis of the cytoskeleton is markedly improved.

Response: Thankyou

Remaining minor issue:

The second sentence of the abstract “Due to the non-proliferative nature of primary podocytes studies into human podocytopathies have relied upon the use of immortalised human cell lines.” is an overstatement and should be removed from the final manuscript.

Response: We have reworded this sentence given that this reviewer feels that this will be seen as a negative statement about the state of the field. We do believe, however, that the difficulty of

isolating primary podocytes from patients has resulted in the lack of direct disease modelling of podocytopathies. Hence our excitement about the data we provide in this manuscript.

Reviewer #3

Our previous response: We have addressed these omissions by including a clear discussion of what has been previously reported. However, we would respectfully argue that this dataset, particularly after this extensive revision, is the most comprehensive characterisation of iPSC-derived podocytes / glomeruli ever provided and that the data presented around utility for disease modelling, particularly using patient-derived iPSC lines, and toxicity screening is novel.

New comment (responses in red):

- The authors have deleted the electron microscopic data upon transplantation, emphasizing the usefulness of the sieved glomeruli from in vitro organoids. However, much of the additional new information is still preliminary.
We respectfully disagree.
- Regarding patient-derived iPSCs, data to support the reproducibility of the experiments (i.e. numbers of clones established and tested, numbers of independent experiments per clone, information and suitability of iPSCs used as a control) are lacking.
 - **We have now provided additional data generated using 2 distinct iPSC lines generated from the same patient. The data is consistent.**
 - **The iPSCs used as a control were a standard wildtype line with no podocyte gene variants.**
 - **All data provided for the analysis of quantitative immunofluorescence were the result of 3 independent experiments (biological replicates) from each iPSC line tested. Statistical analysis is provided.**
- Western blots for NEPHRIN, PODOCIN, and NEPH1 should be presented to show the specific reductions in the first two proteins.
 - **Western blot data is now provided that includes protein lysates prepared from OrgGloms isolated from 3 independent differentiation experiments (biological replicates) compared to a wildtype control. Protein levels of the most significantly downregulated proteins (NEPHRIN and PODOCIN) are included.**
- Importantly, detailed histological abnormalities (i.e. apico-basal distribution of slit diaphragm-related proteins, electron microscopic analysis of foot processes and slit diaphragms) should be presented. Podocytes are characterized by a unique morphology that is tightly associated with their functions and diseases, and it is known that NPHS1 mutations result in the absence of the slit diaphragms (Ruotsalainen et al Am J Pathol 157, 1905, 2000). These data are therefore critical to the argument that the glomeruli sieved from the in vitro organoids model podocytopathies.
 - **We do not claim that OrgGloms have slit diaphragms *in vitro*. We would note that slit diaphragms are also not present in any podocyte monolayer culture currently used to model human podocyte disorder.**
 - **Severely truncating NPHS1 mutations result in the absence of slit diaphragms as the NEPHRIN protein is integral to that structure. Milder NPHS1 mutations however do not necessarily result in loss of slit diaphragms, rather reduction in protein levels due to aberrant trafficking to the cell membrane for example. Our data shows the significant reduction in NEPHRIN protein expression within two distinct iPSC clones derived from a congenital nephrotic syndrome patient. The ease of analysis of OrgGloms by**

immunofluorescence would allow the interrogation of protein mis-localisation associated with disease.

- We do not claim that an OrgGlom is a perfect example of a glomerulus in vivo. However, it is not possible to access material from patients to characterise the mature structure. We do not claim that OrgGloms can model defects of the GBM. If the presence of a slit diaphragm was imperative to a specific disease manifestation, organoids could be transplanted under the renal capsule of mice to further maturation of the OrgGloms (van den Berg et al, 2018) prior to OrgGlom isolation by sieving. We should not be expected to show this in this manuscript as that would represent another 12 months of work and we believe what we present to date is already of importance to the research community.
- Regarding the toxicity experiments, the glomeruli appear to simply die after drug treatment. Are the glomeruli more sensitive to the drug than other lineages in the organoids, such as renal tubules? Again, morphological abnormalities, such as foot process effacement, should also be examined.
 - This experiment was performed to demonstrate not only the utility of OrgGloms at scale for screening purposes, but that the reporter gene expression decreases in response to even a very low dose of cytotoxic drugs. Very high doses resulted in OrgGlom destruction which is unsurprising, however an effect was evident at the lowest dose. Analysis of foot process effacement in a high content fashion is not feasible, but we have shown in Figure 6 a capacity to measure changes in protein localisation and staining intensity using imaging analysis that would be amenable to high content screening. We believe this is sufficient evidence of potential utility.

Original Reviewer Comment: The authors performed gene expression analysis using whole glomeruli, but not purified MAFB-GFP podocytes, while Sharmin et al. already showed microarray analysis of purified NPHS1-GFP podocytes (J Am Soc Nephrol 2016). It is critical to show whether podocytes induced by the authors' method are more mature than those described in the previous report and/or more similar to human podocytes in vivo.

Our previous Response: This is incorrect. In the original manuscript, we profiled podocytes from sieved glomeruli. We have now also profiled podocytes isolated using FACS and also profiled whole glomeruli after sieving from organoids. We also provide extensive proteomic analyses. The data provide in Sharmin et al was microarray data. What we present is RNA-seq. Hence, our profiling is more comprehensive.

New comment:

- Regarding the new Figure 5: it is unclear how the samples for RNA-seq were harvested at each time point. At a first glance, all the samples appear to represent podocytes, because MAB-GFP iPSCs were used. However, the d7+10 sample is FACS-purified podocytes, while the d7+19 sample is whole glomeruli (d7+14 remains unclear). If this is the case, it is not appropriate to compare podocytes with whole glomeruli during the time-course experiments, given that whole glomeruli contain non-podocyte lineages. Comparison of purified podocytes at each stage would be more straightforward and informative.
 - We accept that the explanation surrounding the precise isolation of cells at each time point was not clear. In response, we have provided a more precise description of the source of material for each timepoint in order to improve clarity on this point. The objective was to examine the changes in gene expression of podocytes matured within kidney organoids. In earlier cultures, the organoid glomeruli have not formed tightly packed structures and there are immature pre-podocytes that are already expressing MAFB. Once the organoid glomeruli form interdigitating foot processes, they become difficult to isolated by FACS but readily collected via sieving. Hence, at different timepoints, different approaches were used to capture all MAFB-BFP cells present. The samples at d7+10 were isolated by FACS by MAFB-reporter fluorescence as shown in

Figure 5B. Samples at d7+14 were also isolated by a combination of FACS (immature cells) and sieving (increasingly mature OrgGloms) in order to maximise the podocyte population. D7+19 OrgGloms were isolated by sieving alone. The purpose of this Figure is to depict the increasing maturity of the podocytes and other cell types within OrgGloms with time.

- In addition, it is not clear if endothelial cells are actually present within the glomeruli or if the cells surrounding the glomeruli are contaminated in the sieved samples. These possibilities could be easily tested by staining the OrgGloms with endothelial markers. Notably, there were few endothelial cells in the glomeruli in the authors' previous study (Takasato et al., Nature 2015).
 - This data has been provided in Supplementary Figure 3E which shows immunostaining of OrgGloms for PECAM1/CD31 marking the endothelium. Also shown is staining for PDGFR β marking the mesangial population. Hence, as they mature, a proportion of OrgGloms appear to have both endothelial and mesangial signatures suggesting cellular complexity.
- Importantly, although the authors compare their organoids with 2D podocytes and immortalized cell lines, a comparison with human samples in vivo is lacking.
 - We do not have access to human fetal or adult material for this purpose. We would note that we have compared our results to whatever published data is available with this highlighting that there is little available. Since the resubmission of our manuscript, Rinschen et al (Cell Reports, 2018) have published a comprehensive transcriptomic and proteomic analysis of NPHS2 GFP-positive mouse podocytes compared to non-podocytes from the glomerulus. They also do not provide any data from human glomeruli but concluded that existing human isolated podocyte lines are a poor model. We have provided data here on human itself, which we believe is of greater value to the field and worthy of publication.
- Figure 5F and Fig 1G do not address the critical point of how similar the OrgGloms are to their counterparts in vivo.
 - We have provided extensive data on human iPSC-derived podocytes showing them to be a valid model for human podocytes. The data is extensive and of significant value. As discussed above and previously, we do not have access to such human material. We have trawled the literature for all pre-existing data on podocytes within iPSC-derived kidney tissue and there is also nothing out there. We highlight in Suppl. Figure 5 the fact that data published in Stem Cells arguing a capacity to show a phenotype using an iPSC line with a PODXL mutation was actually profiled in the undifferentiated state, making it very evidence that this is not data from podocytes and hence is of questionable relevance to the field.
We note that there is no figure 1G.
- Fig 4F only shows the presence or absence of each ECM and does not show the relative expression levels, and even in this setting, the OrgGloms look different from those in vivo, including in the expression of COL4A4 (add COL4A4 data in Fig. 4D and F). The new phrase added to the abstract "comparable with human glomeruli in vivo" is therefore misleading and should be deleted.
 - It is not possible to present what is being requested here because of the nature of the prior data that was being used for comparison. The proteomic analysis in the previously published articles looking at human glomeruli and immortalised lines in culture was performed using spectral counting. The proteomics performed on the OrgGlom and OrgPT samples was analysed using the more advanced method of ion intensity. Consequently it was not possible to directly compare the precise protein levels detected between the two methods. Hence presence or absence was used.
 - As discussed in the text, the level of COL4A4 protein in the OrgGlom samples was at the limit of detection. A minimum of 3 unique peptides were used as a cut off for protein

identification to ensure only adequately expressed proteins are represented in the data. Figure 4D details all proteins identified. We have added COL4A4 into figure 4F to represent this.

Original reviewer comment: The authors showed reduction of only three target genes in MAFB-deficient podocytes, and failed to show any key structural abnormalities, such as foot process effacement. Transplantation experiments were not performed using MAFB-deficient podocytes. Thus, it is not convincing at present that the established cell lines combined with the authors' protocol are useful for "accurate modeling of podocytopathy".

Our previous response: Addressed above we now show disease modelling of congenital nephrotic syndrome using an iPS line from a NPHS1 compound heterozygous patient from which organoids were generated and glomeruli isolated.

New reviewer comment:

Please see the above comment. Histological examination is required, even though the cell line has been changed from MAFB-GFP to CNS iPSCs.

- We now provide scanning electron microscopy of the glomeruli present within organoids generated from a control line and a CNS patient-derived line. This data shows clearly the presence of flattened and hypertrophied podocytes on the glomeruli, as has been described by Kuusniemi et al (Kidney International, 2006). This new SEM data is included in Supplementary Figure 4.

Original reviewer comment: Examined numbers of MAFB heterozygous and homozygous clones, as well as numbers of organoids examined per clone, should be described. Data from multiple clones should be presented in Fig 4e.

Our previous response: No longer required. Disease modelling now performed using CNS patient line

New reviewer comment:

Please see the above comment. Data to support reproducibility (such as clonal variations, sample numbers per clone, type and suitability of the control iPSCs) are required, even though the cell line has been changed from MAFB-GFP to CNS iPSCs.

- We have repeated the immunofluorescence and quantitative imaging for a second iPSC clone derived from the same CNS patient. We provide statistical evidence for a distinct and statistically significant reduction in protein intensity of NEPHRIN and PODOCIN and CD2AP in the patient lines. The manuscript alone shows evidence for the capacity to generate isolated organoid glomeruli from at least 5 independent iPSC lines. We have data within our laboratory for a capacity to generate kidney organoids, which contain glomeruli, from more than 15 distinct iPSC lines. We believe this represents sufficient evidence of the utility of the approach and would argue that this provides better evidence of a capacity to characterise (gene and protein) podocytes and model a glomerular disease using an iPSC line than has been provided in the literature previously. Again we would note the transcriptional profiling data from Kim et al (Stem Cells, 2017), which reportedly modelled a podocytopathy, does not profile those podocytes but provides data only from the undifferentiated iPSC line (see Supplementary Figure 5).

Reviewers' comments:

Reviewer #1 (Remarks to the Author):

The authors have responded to my prior questions.

Reviewer #3

Our previous response: We have addressed these omissions by including a clear discussion of what has been previously reported. However, we would respectfully argue that this dataset, particularly after this extensive revision, is the most comprehensive characterisation of iPSC-derived podocytes / glomeruli ever provided and that the data presented around utility for disease modelling, particularly using patient-derived iPSC lines, and toxicity screening is novel.

New comment (responses in red):

The authors have deleted the electron microscopic data upon transplantation, emphasizing the usefulness of the sieved g

Reviewer #4 (Remarks to the Author):

I have reviewed the revised manuscript as a new reviewer because the reviewer #3 could not review the revised manuscript. Hence, I mainly checked whether the revised manuscript addressed reviewer #3's concerns. Here I copied the reviewer #3's comments and authors' replies. My comments are written in Bold after each authors' reply.

I have reviewed the revised manuscript as a new reviewer because the reviewer #3 could not review the revised manuscript. Hence, I mainly checked whether the revised manuscript addressed reviewer #3's concerns. Here I copied the reviewer #3's comments and authors' replies. My comments are written in **Bold** after each authors' reply.

Reviewer #3

Our previous response: We have addressed these omissions by including a clear discussion of what has been previously reported. However, we would respectfully argue that this dataset, particularly after this extensive revision, is the most comprehensive characterisation of iPSC-derived podocytes / glomeruli ever provided and that the data presented around utility for disease modelling, particularly using patient-derived iPSC lines, and toxicity screening is novel.

New comment (**responses in red**):

The authors have deleted the electron microscopic data upon transplantation, emphasizing the usefulness of the sieved glomeruli from in vitro organoids. However, much of the additional new information is still preliminary.

We respectfully disagree.

The manuscript is well-written as expected from the leading lab in nephrology. However, I agree with the reviewer #3 because some data is presented in a misleading way and interpretation of some results is not appropriate. Overall, the novelty of the manuscript is not so strong as a publication in Nature Communications. In terms of the method to isolate glomeruli and culture them, there is no advancement from published approaches. It is not even described whether isolated podocytes can proliferate and be used like as cell lines. If the podocytes cannot be expanded, usefulness of the isolation is questionable. In other words, the advantage of isolated podocytes over whole kidney organoids is unclear. Nishinakamura lab has already demonstrated transcriptome analysis with microarray in JASN [PMID: 26586691] which has lower impact factor than Nature Communications.

Regarding patient-derived iPSCs, data to support the reproducibility of the experiments (i.e. numbers of clones established and tested, numbers of independent experiments per clone, information and suitability of iPSCs used as a control) are lacking.

- **We have now provided additional data generated using 2 distinct iPSC lines generated from the same patient. The data is consistent.**
- **The iPSCs used as a control were a standard wildtype line with no podocyte gene variants.**
- **All data provided for the analysis of quantitative immunofluorescence were the result of 3 independent experiments (biological replicates) from each iPSC line tested. Statistical analysis is provided.**

First, the patient information should be listed in the manuscript. Did the patient exhibit proteinuria from what age? How was the renal biopsy result? Was podocin expression reduced in the patient? Second, the deep-seq results at the mutations sites need to be presented in the patient original cells and generated iPSCs. Spontaneous mutation correction and/or induction can happen during iPS generation from patient cells. Third, in figure 6F, as authors did 3 independent experiments, authors should use three different dots or colors to indicate results from three independent experiments. The number of organoids analyzed in each experiment should be also indicated. Forth, the podocin reduction in OrgGloms is not consistent with the published case [66] as authors discussed in page 17. Is there any case where podocin reduction was shown in patients of NPHS1 mutation?

Western blots for NEPRHIN, PODOCIN, and NEPH1 should be presented to show the specific reductions in the first two proteins.

○ Western blot data is now provided that includes protein lysates prepared from OrgGloms isolated from 3 independent differentiation experiments (biological replicates) compared to a wildtype control. Protein levels of the most significantly downregulated proteins (NEPHRIN and PODOCIN) are included.

The image of whole membranes of western blot should be presented.

Importantly, detailed histological abnormalities (i.e. apico-basal distribution of slit diaphragm related proteins, electron microscopic analysis of foot processes and slit diaphragms) should be presented. Podocytes are characterized by a unique morphology that is tightly associated with their functions and diseases, and it is known that NPHS1 mutations result in the absence of the slit diaphragms (Ruotsalainen et al Am J Pathol 157, 1905, 2000). These data are therefore critical to the argument that the glomeruli sieved from the in vitro organoids model podocytopathies.

○ We do not claim that OrgGloms have slit diaphragms *in vitro*. We would note that slit diaphragms are also not present in any podocyte monolayer culture currently used to model human podocyte disorder.

○ Severely truncating NPHS1 mutations result in the absence of slit diaphragms as the NEPHRIN protein is integral to that structure. Milder NPHS1 mutations however do not necessarily result in loss of slit diaphragms, rather reduction in protein levels due to aberrant trafficking to the cell membrane for example. Our data shows the significant reduction in NEPHRIN protein expression within two distinct iPSC clones derived from a congenital nephrotic syndrome patient. The ease of analysis of OrgGloms by immunofluorescence would allow the interrogation of protein mis-localisation associated with disease.

○ We do not claim that an OrgGlom is a perfect example of a glomerulus in vivo. However, it is not possible to access material from patients to characterise the mature structure. We do not claim that OrgGloms can model defects of the GBM. If the presence of a slit diaphragm was imperative to a specific disease manifestation, organoids could be transplanted under the renal capsule of mice to further maturation of the OrgGloms (van den Berg et al, 2018) prior to OrgGlom isolation by sieving. We should not be expected to show this in this manuscript as that would represent another 12 months of work and we believe what we present to date is already of importance to the research community.

I agree with the reviewer #3 that this part is important for the novelty as a publication in Nature Communications. The advantage of isolated OrgGlom over whole organoids is unclear.

Regarding the toxicity experiments, the glomeruli appear to simply die after drug treatment. Are the glomeruli more sensitive to the drug than other lineages in the organoids, such as renal tubules? Again, morphological abnormalities, such as foot process effacement, should also be examined.

○ This experiment was performed to demonstrate not only the utility of OrgGloms at scale for screening purposes, but that the reporter gene expression decreases in response to even a very low dose of cytotoxic drugs. Very high doses resulted in OrgGlom destruction which is unsurprising, however an effect was evident at the lowest dose. Analysis of foot process effacement in a high content fashion is not feasible, but we have shown in Figure 6 a capacity to measure changes in protein localisation and staining intensity using imaging analysis that would be amenable to high content screening. We believe this is sufficient evidence of potential utility.

I agree with the reviewer #3. It is not clear whether OrgGlom is better than other current tools to assess podocyte toxicity. Evaluation of apoptosis is not specific to assess podocyte-specific injury induced by doxorubicin. What is the mechanism of doxorubicin-induced podocyte injury? Due to lack of appropriate controls, the advantage of OrgGlom is not clear.

Original Reviewer Comment: The authors performed gene expression analysis using whole glomeruli, but not purified MAFB-GFP podocytes, while Sharmin et al. already showed microarray analysis of purified NPHS1-GFP podocytes (J Am Soc Nephrol 2016). It is critical to show whether podocytes induced by the authors' method are more mature than those described in the previous report and/or more similar to human podocytes in vivo.

Our previous Response: This is incorrect. In the original manuscript, we profiled podocytes from sieved glomeruli. We have now also profiled podocytes isolated using FACS and also profiled whole glomeruli after sieving from organoids. We also provide extensive proteomic analyses. The data provide in Sharmin et al was microarray data. What we present is RNA-seq. Hence, our profiling is more comprehensive.

I agree with the reviewer #3 that the current manuscript lacks definitive evidence that authors' podocytes are more matured than podocytes induced by other differentiation protocols. Humphreys lab indicated clear difference in maturation of podocytes differentiated by two different published protocols [Wu, bioRxiv, 20017], suggesting that authors' podocytes are less matured than Bonventre's. I agree that organoids contain more matured podocytes than human podocyte cell lines; however, lack of human podocyte data in figure 3 as a control makes it impossible to conclude that their induced podocytes are matured. To me, the figure 3 results demonstrate dedifferentiation of podocytes after isolation and culture because gene expression in OrgPod became similar to immortalized podocytes in comparison to OrgGlom. This result indicates that isolation of podocytes from organoids leads to less matured phenotypes than podocytes in organoids, suggesting the need of new culture methods to retain podocyte characteristics after isolation. The current manuscript does not contain any novel method for that purpose.

New comment:

Regarding the new Figure 5: it is unclear how the samples for RNA-seq were harvested at each time point. At a first glance, all the samples appear to represent podocytes, because MAB-GFP iPSCs were used. However, the d7+10 sample is FACS-purified podocytes, while the d7+19 sample is whole glomeruli (d7+14 remains unclear). If this is the case, it is not appropriate to compare podocytes with whole glomeruli during the time-course experiments, given that whole glomeruli contain non-podocyte lineages. Comparison of purified podocytes at each stage would be more straightforward and informative.

○ We accept that the explanation surrounding the precise isolation of cells at each time point was not clear. In response, we have provided a more precise description of the source of material for each timepoint in order to improve clarity on this point. The objective was to examine the changes in gene expression of podocytes matured within kidney organoids. In earlier cultures, the organoid glomeruli have not formed tightly packed structures and there are immature pre-podocytes that are already expressing MAFB. Once the organoid glomeruli form interdigitating foot processes, they become difficult to isolated by FACS but readily collected via sieving. Hence, at different timepoints, different approaches were used to capture all MAFB-BFP cells present. The samples at d7+10 were isolated by FACS by MAFB-reporter fluorescence as shown in Figure 5B. Samples at d7+14 were also isolated by a combination of FACS (immature

cells) and sieving (increasingly mature OrgGloms) in order to maximise the podocyte population. D7+19 OrgGloms were isolated by sieving alone. The purpose of this Figure is to depict the increasing maturity of the podocytes and other cell types within OrgGloms with time.

I agree with the reviewer #3 that it is not appropriate to compare gene expression in samples obtained by different methods. The revised manuscript is still confusing and misleading.

In addition, it is not clear if endothelial cells are actually present within the glomeruli or if the cells surrounding the glomeruli are contaminated in the sieved samples. These possibilities could be easily tested by staining the OrgGloms with endothelial markers. Notably, there were few endothelial cells in the glomeruli in the authors' previous study (Takasato et al., Nature 2015).

○ This data has been provided in Supplementary Figure 3E which shows immunostaining of OrgGloms for PECAM1/CD31 marking the endothelium. Also shown is staining for PDGFR β marking the mesangial population. Hence, as they mature, a proportion of OrgGloms appear to have both endothelial and mesangial signatures suggesting cellular complexity.

I suppose Supplementary Figure 3E meant Supplementary Figure 4E. Even if so, the data is not convincing enough to conclude that there are mesangial cells and endothelia in organoid glomeruli. PDGFR β staining appears positive in most of cells in glomeruli. PECAM1 staining also look positive in all cells in the structure. Co-staining with podocyte markers and quantification are necessary.

Importantly, although the authors compare their organoids with 2D podocytes and immortalized cell lines, a comparison with human samples in vivo is lacking.

○ We do not have access to human fetal or adult material for this purpose. We would note that we have compared our results to whatever published data is available with this highlighting that there is little available. Since the resubmission of our manuscript, Rinschen et al (Cell Reports, 2018) have published a comprehensive transcriptomic and proteomic analysis of NPHS2 GFP-positive mouse podocytes compared to nonpodocytes from the glomerulus. They also do not provide any data from human glomeruli but concluded that existing human isolated podocyte lines are a poor model. We have provided data here on human itself, which we believe is of greater value to the field and worthy of publication.

Authors demonstrated isolated glomeruli from adult human kidney in Figure 1Aiii. If the human samples are no

longer available, authors should tone down their conclusions throughout the manuscript. Figure 3 simply demonstrates that isolation of podocytes from kidney organoids leads to less matured podocytes similar to immortalized human podocytes.

Figure 5F and Fig 1G do not address the critical point of how similar the OrgGloms are to their counterparts in vivo.

○ We have provided extensive data on human iPSC-derived podocytes showing them to be a valid model for human podocytes. The data is extensive and of significant value. As discussed above and previously, we do not have access to such human material. We have trawled the literature for all pre-existing data on podocytes within iPSC-derived kidney tissue and there is also nothing out there. We highlight in Suppl. Figure 5 the fact that data published in Stem Cells arguing a capacity to show a phenotype using an iPSC line with a PODXL mutation was actually profiled in the undifferentiated state, making it very evidence that this is not data from podocytes and hence is of questionable relevance to the field.

We note that there is no figure 1G.

Figure 1 just says that glomeruli can be isolated from kidney organoids, so I think this is fine.

Fig 4F only shows the presence or absence of each ECM and does not show the relative expression levels, and even in this setting, the OrgGloms look different from those in vivo, including in the expression of COL4A4 (add COL4A4 data in Fig. 4D and F). The new phrase added to the abstract “comparable with human glomeruli in vivo” is therefore misleading and should be deleted.

○ It is not possible to present what is being requested here because of the nature of the prior data that was being used for comparison. The proteomic analysis in the previously published articles looking at human glomeruli and immortalised lines in culture was performed using spectral counting. The proteomics performed on the OrgGlom and OrgPT samples was analysed using the more advanced method of ion intensity. Consequently it was not possible to directly compare the precise protein levels detected between the two methods. Hence presence or absence was used.

○ As discussed in the text, the level of COL4A4 protein in the OrgGlom samples was at the limit of detection. A minimum of 3 unique peptides were used as a cut off for protein identification to ensure only adequately expressed proteins are represented in the data. Figure 4D details all proteins identified. We have added COL4A4 into figure 4F to represent this.

As authors say that it is not possible to compare values from the database to authors' result, the current Figure 4F is not appropriate. Authors should remove Figure 4F and "comparable with human glomeruli in vivo" as the reviewer #3 said. Then, Figure 4 will essentially show difference between OrgGlom and OrgPT whose conclusion will be quite different from the current one.

Original reviewer comment: The authors showed reduction of only three target genes in MAFBdeficient podocytes, and failed to show any key structural abnormalities, such as foot process effacement. Transplantation experiments were not performed using MAFB-deficient podocytes. Thus, it is not convincing at present that the established cell lines combined with the authors' protocol are useful for "accurate modeling of podocytopathy".

Our previous response: Addressed above we now show disease modelling of congenital nephrotic syndrome using an iPSC line from a NPHS1 compound heterozygous patient from which organoids were generated and glomeruli isolated.

New reviewer comment:

Please see the above comment. Histological examination is required, even though the cell line has been changed from MAFB-GFP to CNS iPSCs.

○ We now provide scanning electron microscopy of the glomeruli present within organoids generated from a control line and a CNS patient-derived line. This data shows clearly the presence of flattened and hypertrophied podocytes on the glomeruli, as has been described by Kuusniemi et al (Kidney International, 2006). This new SEM data is included in Supplementary Figure 4.

Kuusniemi et al demonstrated hypertrophied cell bodies by light microscopy but not by SEM. The SEM images in supplementary Figure 5A look quite different from what was published in Kuusniemi's paper. Overall, the NPHS1 mutant part in the manuscript appears immature as a publication in Nature Communications. More data is required.

Original reviewer comment: Examined numbers of MAFB heterozygous and homozygous clones, as well as numbers of organoids examined per clone, should be described. Data from multiple clones should be presented in Fig 4e.

Our previous response: No longer required. Disease modelling now performed using CNS patient line

New reviewer comment:

Please see the above comment. Data to support reproducibility (such as clonal variations, sample numbers per clone, type and suitability of the control iPSCs) are required, even though the cell line has been changed from MAFB-GFP to CNS iPSCs.

○ We have repeated the immunofluorescence and quantitative imaging for a second iPSC clone derived from the same CNS patient. We provide statistical evidence for a distinct and statistically significant reduction in protein intensity of NEPHRIN and PODOCIN and CD2AP in the patient lines. The manuscript alone shows evidence for the capacity to generate isolated organoid glomeruli from at least 5 independent iPSC lines. We have data within our laboratory for a capacity to generate kidney organoids, which contain glomeruli, from more than 15 distinct iPSC lines. We believe this represents sufficient evidence of the utility of the approach and would argue that this provides better evidence of a capacity to characterise (gene and protein) podocytes and model a glomerular disease using an iPSC line than has been provided in the literature previously. Again we would note the transcriptional profiling data from Kim et al (Stem Cells, 2017), which reportedly modelled a podocytopathy, does not profile those podocytes but provides data only from the undifferentiated iPSC line (see Supplementary Figure 5).

According to the following data from Kim et al (Stem Cells, 2017), the transcriptional profiling data is supposed to be generated from kidney organoids but not from undifferentiated iPSC line (I believe they used hESCs but not iPSCs). In any case, to compare gene expression profiles obtained from different sources (isolated glomeruli v.s. whole organoids) is not appropriate. Thus, supplementary Figure 3 should be removed.

<https://www.ncbi.nlm.nih.gov/geo/query/acc.cgi?acc=GSE103547>

There are other concerns such as glomeruli number. Authors say >3000 OrgGloms can be isolated from one experiment, but there is no actual data provided in the manuscript. How did they count the glomeruli number? Did they isolate kidney organoid area apparently by bright field? Did they modify differentiation protocols to the 15 distinct iPSC lines? Quality control of organoid generation is not also discussed. How do they choose successful differentiation samples?

Detailed responses to reviewer #4. Note, comments from reviewer 4 are in green and our new responses to reviewer 4 are in bold red.

Reviewer #4 comment: I have reviewed the revised manuscript as a new reviewer because the reviewer #3 could not review the revised manuscript. Hence, I mainly checked whether the revised manuscript addressed reviewer #3's concerns. Here I copied the reviewer #3's comments and authors' replies. My comments are written in Bold after each authors' reply.

Reviewer #3

Prior response to reviewer #3. We have addressed these omissions by including a clear discussion of what has been previously reported. However, we would respectfully argue that this dataset, particularly after this extensive revision, is the most comprehensive characterisation of iPSC-derived podocytes / glomeruli ever provided and that the data presented around utility for disease modelling, particularly using patient-derived iPSC lines, and toxicity screening is novel.

New comment by reviewer #3:

The authors have deleted the electron microscopic data upon transplantation, emphasizing the usefulness of the sieved glomeruli from in vitro organoids. However, much of the additional new information is still preliminary.

We respectfully disagree.

Reviewer #4: The manuscript is well-written as expected from the leading lab in nephrology. However, I agree with the reviewer #3 because some data is presented in a misleading way and interpretation of some results is not appropriate. Overall, the novelty of the manuscript is not so strong as a publication in Nature Communications.

Response: These are subjective comments without clarity and as such are difficult to definitively respond to. However, we completely disagree that we have been in any way misleading about what we have written. With respect to novelty, we would once again state that this is the first ever study to demonstrate that glomerular structures within kidney organoids are sufficiently robust to be isolated in an intact fashion and show that this provides a significantly improved population of podocytes upon which to study the basis of primary glomerular disease. We also show unequivocally that podocytes within organoid-derived glomeruli maintain an appropriate cellular identity superior to primary isolated, immortalised or organoid derived podocytes cultured in 2D. This is completely novel. We also show the capacity to use organoid derived glomeruli to screen response to compounds and to model disease. This is currently completely impossible using patient material and is clearly not going to be valid using primary cells cultured in 2D. By being iPSC-derived, these can be readily generated in large numbers from any individual. The argument that we could just as easily use organoids shows a lack of understanding of the requirements for screening. What is possible here is to generate many glomeruli from a single organoid, significantly increasing the options available for biological investigation. No other currently available approach to studying glomerular disease provides these advantages. As two other reviewers, who have reviewed this manuscript now on several occasions, regard it is as suitable for publication would suggest that they disagree with this reviewer.

Reviewer #4: In terms of the method to isolate glomeruli and culture them, there is no advancement from published approaches. It is not even described whether isolated podocytes can proliferate and be used like as cell lines. If the podocytes cannot be expanded, usefulness of the isolation is

questionable. In other words, the advantage of isolated podocytes over whole kidney organoids is unclear. Nishinakamura lab has already demonstrated transcriptome analysis with microarray in JASN [PMID: 26586691] which has lower impact factor than Nature Communications.

Response to reviewer #4: The nature of these statements would suggest that this reviewer has completely missed the point of the manuscript. Monocultured glomerular cells in 2D culture lack key phenotypic characteristics. At no point do we claim that sieving is a novel protocol. We clearly acknowledge that this is an approach that can be applied to adult kidney tissue. However, i) it is not possible to access sufficient material by biopsy from a patient to use isolated glomeruli as the screening tool and ii) no one has previously shown a capacity to apply sieving to organoids. The fact that the reviewer asks whether the podocytes proliferate is also a moot point. We show that isolated podocytes from glomeruli behave as well as, in fact transcriptionally slightly better than, the gold standard immortalised podocyte line and behave akin to primary podocytes isolated from human tissue. We note that the immortalised line does not proliferate when differentiated. Indeed, podocytes don't like to proliferate, which is one of the challenges of the field as they are terminally differentiated. The point of our manuscript, which is completely novel data, is that a 3D OrgGlom better retains podocyte structure and molecular identity without the need to proliferate as we can isolate many glomeruli from any given organoid. Hence the statement that the 'usefulness of isolation is questionable' is simply wrong. The reviewer then asks what the advantage of isolated podocytes is over whole organoids. We are arguing that isolated glomeruli are the asset here. To have to use an entire organoid to analyse a podocyte response is unnecessary. What we can generate is an entire plate of glomeruli sieved from a single organoid which provides us a capacity to screen, something that has never previously been possible.

The comment that Nishinakamura has already demonstrated transcriptome analysis with microarrays we have addressed previously. That study profiled the outcome of a completely different protocol. As it was performed using microarrays, it is impossible to compare this data with our own in an unbiased statistical fashion given the shift in platform. It also was not comprehensively compared to a gold standard nor analysed across the duration of the protocol. The level of analysis we present is far more focussed and extensive than that dataset. We note that we have also previously published RNAseq data on our own protocol for generating kidney structures (Combes et al, bioRxiv, 2017) but again that was not isolated glomeruli. The data in this manuscript provides the most robust analysis to date of the transcriptional profile of an organoid-derived glomerulus. We note that it also provides the most extensive analysis to date of the identity of the immortalised podocyte line commonly used in the literature in both the 'proliferating' (33C) and 'differentiated' (37C) state. It is our opinion that our manuscript is being harshly viewed because it calls into question the validity of that model. We note that we are not the only group to question this approach and have now included reference to the recent publication (Rinschen et al, Cell Reports, 2018) that also clearly shows that this is a suboptimal model of podocytes.

Reviewer 3: Regarding patient-derived iPSCs, data to support the reproducibility of the experiments (i.e. numbers of clones established and tested, numbers of independent experiments per clone, information and suitability of iPSCs used as a control) are lacking.

- We have now provided additional data generated using 2 distinct iPSC lines generated from the same patient. The data is consistent.
- The iPSCs used as a control were a standard wildtype line with no podocyte gene variants.
- All data provided for the analysis of quantitative immunofluorescence were the result of 3 independent experiments (biological replicates) from each iPSC line tested. Statistical analysis is provided.

Reviewer #4 comment: First, the patient information should be listed in the manuscript. Did the patient exhibit proteinuria from what age? How was the renal biopsy result? Was podocin expression reduced in the patient? Second, the deep-seq results at the mutations sites need to be presented in the patient original cells and generated iPSCs. Spontaneous mutation correction and/or induction can happen during iPSC generation from patient cells. Third, in figure 6F, as authors did 3 independent experiments, authors should use three different dots or colors to indicate results from three independent experiments. The number of organoids analyzed in each experiment should be also indicated. Forth, the podocin reduction in OrgGloms is not consistent with the published case [66] as authors discussed in page 17. Is there any case where podocin reduction was shown in patients of NPHS1 mutation?

Response: We have now provided a complete clinical report of this patient together with the outcome of clinical diagnostic sequencing, the details on clone derivation, tests of pluripotency and SNP reports after derivation, evidence that the lines can differentiate to kidney organoids and validation at the DNA level that the patient mutation remains present within both clones. We note that the initial identification of this mutation was performed within a clinically accredited diagnostic clinic and we have extensive experience in the derivation and characterisation of such lines from patients and are completely aware of what is required for appropriate quality control. Indeed, a third clone from this patient was not used as SNP analysis revealed spontaneous mutational events. All of this information is now provided as Supplementary data as there is no capacity to include this within what is already a large manuscript. See new Supplementary Figure 5.

While we concede the value in presenting such additional information, questions such as 'was podocin reduced in the patient' has been directly addressed in the manuscript using our approach of IF on isolated organoid glomeruli. The only alternative would be to make a primary cell line which we show will be an inferior approach. We were asked whether there is any specific case where podocin reduction has previously been shown in a patient with NPHS1 mutation. The answer is not that we are aware of. This reflects the paucity of cell biological understanding available due to the lack of a simple approach to study such patients at the individual level. However, there are known protein-protein interaction between NEPHRIN and PODOCIN and there are prior reports of reduced glomerular expression of slit diaphragm proteins in glomerular injury. Rather than doubting the data, the reviewer could have acknowledged that this represents a novel discovery for which we are providing quantified data from two independent clones.

Western blots for NEPHRIN, PODOCIN, and NEPH1 should be presented to show the specific reductions in the first two proteins.

- Western blot data is now provided that includes protein lysates prepared from OrgGloms isolated from 3 independent differentiation experiments (biological

replicates) compared to a wildtype control. Protein levels of the most significantly downregulated proteins (NEPHRIN and PODOCIN) are included.

Reviewer #4. The image of whole membranes of western blot should be presented.

Response: As described in the methods, the Western analysis was performed using the ProteinSimple Wes Capillarywestern Blot analyser. This is a more sensitive approach to Western Blotting than the conventional blot and film approach, instead using a capillary-based immunoassay platform. The results are generated in both a graphical manner (akin to a quantitative PCR plot) and a virtual blot image, which is what we displayed in our figure. This

system has been cited in over 14,000 publications since 1992 is and widely regarded as more precise than classical methods.

Reviewer #3: Importantly, detailed histological abnormalities (i.e. apico-basal distribution of slit diaphragm related proteins, electron microscopic analysis of foot processes and slit diaphragms) should be

presented. Podocytes are characterized by a unique morphology that is tightly associated with their functions and diseases, and it is known that NPHS1 mutations result in the absence of the

slit diaphragms (Ruotsalainen et al Am J Pathol 157, 1905, 2000). These data are therefore critical to the argument that the glomeruli sieved from the in vitro organoids model podocytopathies.

- We do not claim that OrgGloms have slit diaphragms in vitro. We would note that slit diaphragms are also not present in any podocyte monolayer culture currently used to model human podocyte disorder.

- Severely truncating NPHS1 mutations result in the absence of slit diaphragms as the NEPHRIN protein is integral to that structure. Milder NPHS1 mutations however do not necessarily result in loss of slit diaphragms, rather reduction in protein levels due to aberrant trafficking to the cell membrane for example. Our data shows the significant reduction in NEPHRIN protein expression within two distinct iPSC clones derived from a congenital nephrotic syndrome patient. The ease of analysis of OrgGloms by immunofluorescence would allow the interrogation of protein mis-localisation associated with disease.

- We do not claim that an OrgGlom is a perfect example of a glomerulus in vivo. However, it is not possible to access material from patients to characterise the mature structure.

We do not claim that OrgGloms can model defects of the GBM. If the presence of a slit diaphragm was imperative to a specific disease manifestation, organoids could be transplanted under the renal capsule of mice to further maturation of the OrgGloms

(van den Berg et al, 2018) prior to OrgGlom isolation by sieving. We should not be expected to show this in this manuscript as that would represent another 12 months of work and we believe what we present to date is already of importance to the research community.

Reviewer #4: I agree with the reviewer #3 that this part is important for the novelty as a publication in Nature Communications. The advantage of isolated OrgGlom over whole organoids is unclear.

Response to reviewer #4: This is a very vague request. The statement that the advantage of Org Glom over whole organoids is unclear is inaccurate. The organoids provide an environment within which many glomeruli pattern in a manner similar to what happens in vivo and can then be sieved from these structures to act as avatars of the patient's own glomeruli. It is also not possible to generate hundreds of organoids and float them in 96 well trays for analysis, whereas we show this is feasible for OrgGloms

Reviewer #4: Regarding the toxicity experiments, the glomeruli appear to simply die after drug treatment. Are the glomeruli more sensitive to the drug than other lineages in the organoids, such as renal tubules? Again, morphological abnormalities, such as foot process effacement, should also be examined.

Response: Our previous response stands: This experiment was performed to demonstrate not only the utility of OrgGloms at scale for screening purposes, but that the reporter gene expression decreases in response to even a very low dose of cytotoxic drugs. Very high doses resulted in

OrgGlom destruction which is unsurprising, however an effect was evident at the lowest dose. Analysis of foot process effacement in a high content fashion is not feasible, but we have shown in Figure 6 a capacity to measure changes in protein localisation and staining intensity using imaging analysis that would be amenable to high content screening. We believe this is sufficient evidence of potential utility.

Reviewer #4: I agree with the reviewer #3. It is not clear whether OrgGlom is better than other current tools to assess podocyte toxicity. Evaluation of apoptosis is not specific to assess podocyte-specific injury induced by doxorubicin. What is the mechanism of doxorubicin-induced podocyte injury? Due to lack of appropriate controls, the advantage of OrgGlom is not clear.

Response to reviewer #4: Our previous rebuttal to this issue was sufficient in our opinion. This reviewer is asking for mechanism around doxorubicin itself, which is outside the scope of this manuscript. As explained on numerous occasions, there is no other approach in which such a large number of conditions could be screened. This is not a paper about the specific toxicity test, but about the proof of concept that sufficient accurately patterned structures can be generated to perform such a test. We believe we adequately addressed this in our prior response to reviewer #3.

Original Reviewer Comment: The authors performed gene expression analysis using whole glomeruli, but not purified MAFB-GFP podocytes, while Sharmin et al. already showed microarray analysis of purified NPHS1-GFP podocytes (J Am Soc Nephrol 2016). It is critical to show whether podocytes induced by the authors' method are more mature than those described in the previous report and/or more similar to human podocytes in vivo.

Our previous Response: This is incorrect. In the original manuscript, we profiled podocytes from sieved glomeruli. We have now also profiled podocytes isolated using FACS and also profiled whole glomeruli after sieving from organoids. We also provide extensive proteomic analyses. The data provide in Sharmin et al was microarray data. What we present is RNA-seq. Hence, our profiling is more comprehensive.

Reviewer #4: I agree with the reviewer #3 that the current manuscript lacks definitive evidence that authors' podocytes are more matured than podocytes induced by other differentiation protocols. Humphreys lab indicated clear difference in maturation of podocytes differentiated by two different published protocols [Wu, bioRxiv, 20017], suggesting that authors' podocytes are less matured than Bonventre's. I agree that organoids contain more matured podocytes than human podocyte cell lines; however, lack of human podocyte data in figure 3 as a control makes it impossible to conclude that their induced podocytes are matured. To me, the figure 3 results demonstrate dedifferentiation of podocytes after isolation and culture because gene expression in OrgPod became similar to immortalized podocytes in comparison to OrgGlom. This result indicates that isolation of podocytes from organoids leads to less matured phenotypes than podocytes in organoids, suggesting the need of new culture methods to retain podocyte characteristics after isolation. The current manuscript does not contain any novel method for that purpose.

Response to reviewer #4: What definitive evidence is being requested here? This paper highlights the difference between monoculture podocytes and 2D vs 3D culture. We do not advocate the use of these cells in 2D. We actually make a point of showing that 2D culture is inferior. We believe we have shown definitively and comprehensively that 3D OrgGloms are a good model of a glomerulus in vitro. We present an exhaustive profiling of podocytes within organoids across time and we show functional, IF and proteomic analyses never previously present. We also show evidence of

maturation markers in the OrgGloms including basement membrane components (LAMB2 especially).

The single cell data of Wu shows evidence of podocyte patterning in our protocol and that of another. This does not disagree with our conclusions but also does not go far enough to fully characterise the maturity of the cells described. Our own single cell data published at exactly that same time also shows evidence of podocytes (Combes et al, bioRxiv, 2017), however again this was not performed to the level of detail or in a fashion that facilitates direct unbiased comparison between immortalised line, 2D culture and 3D OrgGloms. I simply do not know how much more we could have done to definitively prove the validity of this model. I would note that the data we provide to show that what we have is podocytes far exceeds that provided by Musah et al, published in Nature Biomedical Engineering (2017) and Nature Protocols (2018) in which there is **NO** transcriptional validation of the identity of the cells they claim to be podocytes, and yet they have had this data published in two Nature journals of higher profile than Nature Communications. We also note that the data presented in the Musah paper compares their model to the same conditionally immortalised cells used in this paper. Musah show that their podocyte cell model is directly comparable to the Saleem et al immortalised cells with matched expression levels of podocyte-specific markers. Given how extensively we have already shown our model to be an advance on this immortalised cell line, it would be redundant to perform yet another comparison between our OrgGloms and Musah's 2D cells on a chip.

Reviewer #3: Regarding the new Figure 5: it is unclear how the samples for RNA-seq were harvested at each time point. At a first glance, all the samples appear to represent podocytes, because MAB-GFP iPSCs were used. However, the d7+10 sample is FACS-purified podocytes, while the d7+19 sample is whole glomeruli (d7+14 remains unclear). If this is the case, it is not appropriate to compare podocytes with whole glomeruli during the time-course experiments, given that whole glomeruli contain non-podocyte lineages. Comparison of purified podocytes at each stage would be more straightforward and informative.

○ We accept that the explanation surrounding the precise isolation of cells at each time point was not clear. In response, we have provided a more precise description of the source of material for each timepoint in order to improve clarity on this point. The objective was to examine the changes in gene expression of podocytes matured within kidney organoids. In earlier cultures, the organoid glomeruli have not formed tightly packed structures and there are immature pre-podocytes that are already expressing MAFB. Once the organoid glomeruli form interdigitating foot processes, they become difficult to isolated by FACS but readily collected via sieving. Hence, at different timepoints, different approaches were used to capture all MAFB-BFP cells present. The samples at d7+10 were isolated by FACS by MAFB-reporter fluorescence as shown in Figure 5B. Samples at d7+14 were also isolated by a combination of FACS (immature cells) and sieving (increasingly mature OrgGloms) in order to maximise the podocyte population. D7+19 OrgGloms were isolated by sieving alone. The purpose of this Figure is to depict the increasing maturity of the podocytes and other cell types within OrgGloms with time.

Reviewer #4: I agree with the reviewer #3 that it is not appropriate to compare gene expression in samples obtained by different methods. The revised manuscript is still confusing and misleading.

Response to reviewer #4: We do not agree with this comment- this analysis is valid in that we have used a fluorescent tag to identify the cell type of interest and have validated the veracity of that tag. You cannot sieve until a glom has formed tight cell-cell associations. What this timecourse does is capture all MAFB+ cells across time. We had modified the text to make this clear. We do not see what else could be done here. We also object to arguments that this is

misleading. It is described in full but is now being criticised for the approach. This would seem a circular argument.

In addition, it is not clear if endothelial cells are actually present within the glomeruli or if the cells surrounding the glomeruli are contaminated in the sieved samples. These possibilities could be easily tested by staining the OrgGloms with endothelial markers. Notably, there were few endothelial cells in the glomeruli in the authors' previous study (Takasato et al., Nature 2015).

○ This data has been provided in Supplementary Figure 3E which shows immunostaining of OrgGloms for PECAM1/CD31 marking the endothelium. Also shown is staining for PDGFR β marking the mesangial population. Hence, as they mature, a proportion of OrgGloms appear to have both endothelial and mesangial signatures suggesting cellular complexity.

Reviewer #4: I suppose Supplementary Figure 3E meant Supplementary Figure 4E. Even if so, the data is not convincing enough to conclude that there are mesangial cells and endothelia in organoid glomeruli. PDGFR β staining appears positive in most of cells in glomeruli. PECAM1 staining also look positive in all cells in the structure. Co-staining with podocyte markers and quantification are necessary.

Response to reviewer #4: The IF was performed on sieved gloms to validate at the protein level the gene expression evidence for the increasing presence of these cells types within OrgGloms across time. We at no point claim there to be large populations of mesangial or endothelial cells within the OrgGloms, but the IF and profiling does suggest their presence as evidence of glomerular maturation.

Importantly, although the authors compare their organoids with 2D podocytes and immortalized cell lines, a comparison with human samples in vivo is lacking.

○ We do not have access to human fetal or adult material for this purpose. We would note that we have compared our results to whatever published data is available with this highlighting that there is little available. Since the resubmission of our manuscript, Rinschen et al (Cell Reports, 2018) have published a comprehensive transcriptomic and proteomic analysis of NPHS2 GFP-positive mouse podocytes compared to nonpodocytes from the glomerulus. They also do not provide any data from human glomeruli but concluded that existing human isolated podocyte lines are a poor model. We have provided data here on human itself, which we believe is of greater value to the field and worthy of publication.

Reviewer #4 comments: Authors demonstrated isolated glomeruli from adult human kidney in Figure 1Aiii. If the human samples are no longer available, authors should tone down their conclusions throughout the manuscript. Figure 3 simply demonstrates that isolation of podocytes from kidney organoids leads to less matured podocytes similar to immortalized human podocytes.

Response: This reviewer says we should be comparing our data to adult human glomeruli and then says that we cannot compare to existing datasets if the samples are no longer available. The image included in Figure 1Aiii of adult human glomeruli was provided by a co-author to show the phenotypic similarities between human glomeruli and human iPSC-derived OrgGloms. The proteomic data from adult OrgGloms has previously been published by the same co-author (Rachel Lennon) and it was this data to which we compared our isolated OrgGloms. The respective proteomics were performed on different platforms (the OrgGloms with more depth) and could therefore not be directly compared, hence the presentation of the data we provided. I would note that this point was clarified extensively in a previous round of reviewer's comments. We have been asked to compare our data to many other available datasets. We have done so when it is

mathematically appropriate and possible to do so in an unbiased fashion. As such, the data in Figure 1Aiii is valid.

Figure 5F and Fig 1G do not address the critical point of how similar the OrgGloms are to their counterparts in vivo.

○ We have provided extensive data on human iPSC-derived podocytes showing them to be a valid model for human podocytes. The data is extensive and of significant value. As discussed above and previously, we do not have access to such human material. We have trawled the literature for all pre-existing data on podocytes within iPSC-derived kidney tissue and there is also nothing out there. We highlight in Suppl. Figure 5 the fact that data published in Stem Cells arguing a capacity to show a phenotype using an iPSC line with a PODXL mutation was actually profiled in the undifferentiated state, making it very evidence that this is not data from podocytes and hence is of questionable relevance to the field.

We note that there is no figure 1G.

Figure 1 just says that glomeruli can be isolated from kidney organoids, so I think this is fine.

Fig 4F only shows the presence or absence of each ECM and does not show the relative expression levels, and even in this setting, the OrgGloms look different from those in vivo, including in the expression of COL4A4 (add COL4A4 data in Fig. 4D and F). The new phrase added to the abstract “comparable with human glomeruli in vivo” is therefore misleading and should be deleted.

○ It is not possible to present what is being requested here because of the nature of the prior data that was being used for comparison. The proteomic analysis in the previously published articles looking at human glomeruli and immortalised lines in culture was performed using spectral counting. The proteomics performed on the OrgGlom and OrgPT samples was analysed using the more advanced method of ion intensity. Consequently it was not possible to directly compare the precise protein levels detected between the two methods. Hence presence or absence was used.

○ As discussed in the text, the level of COL4A4 protein in the OrgGlom samples was at the limit of detection. A minimum of 3 unique peptides were used as a cut off for protein identification to ensure only adequately expressed proteins are represented in the data.

Figure 4D details all proteins identified. We have added COL4A4 into figure 4F to represent this.

Reviewer #4 comment: As authors say that it is not possible to compare values from the database to authors' result, the current Figure 4F is not appropriate. Authors should remove Figure 4F and “comparable with human glomeruli in vivo” as the reviewer #3 said. Then, Figure 4 will essentially show difference between OrgGlom and OrgPT whose conclusion will be quite different from the current one.

Response to reviewer #4: Again this is a mixed message. We are being asked to make more comparisons and then to remove others. We have substantial experience in transcriptomic and proteomic analyses. On the issue of the proteomic analyses, it is valid to compare identifications (presence/absence) in the way we have presented. Many proteomic papers make the same comparisons (eg Horton, Humphries et al, Consensus adhesome, Nat Cell Biology, 2016). Comparable is not binary- rather a spectrum. Hence we believe that the comment ‘comparable with human glomeruli in vivo’ is ok. Again, this data is the first global analysis of the matrices in kidney organoid, the results demonstrate

that this culture system will be a valuable and tractable asset for the investigation of GBM assembly and maturation.

Original reviewer comment: The authors showed reduction of only three target genes in MAFB-deficient podocytes, and failed to show any key structural abnormalities, such as foot process effacement. Transplantation experiments were not performed using MAFB-deficient podocytes. Thus, it is not convincing at present that the established cell lines combined with the authors' protocol are useful for "accurate modeling of podocytopathy".

Our previous response: Addressed above we now show disease modelling of congenital nephrotic syndrome using an iPSC line from a NPHS1 compound heterozygous patient from which organoids were generated and glomeruli isolated.

New reviewer comment: Please see the above comment. Histological examination is required, even though the cell line has been changed from MAFB-GFP to CNS iPSCs.

○ We now provide scanning electron microscopy of the glomeruli present within organoids generated from a control line and a CNS patient-derived line. This data shows clearly the presence of flattened and hypertrophied podocytes on the glomeruli, as has been described by Kuusniemi et al (Kidney International, 2006). This new SEM data is included in Supplementary Figure 4.

Kuusniemi et al demonstrated hypertrophied cell bodies by light microscopy but not by SEM. supplementary Figure 5A look quite different from what was published in Kuusniemi's paper. mutant part in the manuscript appears immature as a publication in Nature Communications. More data is required.

Original reviewer comment: Examined numbers of MAFB heterozygous and homozygous clones, as well as numbers of organoids examined per clone, should be described. Data from multiple clones should be presented in Fig 4e.

Our previous response: No longer required. Disease modelling now performed using CNS patient line
New reviewer comment:

Please see the above comment. Data to support reproducibility (such as clonal variations, sample numbers per clone, type and suitability of the control iPSCs) are required, even though the cell line has been changed from MAFB-GFP to CNS iPSCs.

The SEM images in Overall, the NPHS1

○ We have repeated the immunofluorescence and quantitative imaging for a second iPSC clone derived from the same CNS patient. We provide statistical evidence for a distinct and statistically significant reduction in protein intensity of NEPHRIN and PODOCIN and CD2AP in the patient lines. The manuscript alone shows evidence for the capacity to generate isolated organoid glomeruli from at least 5 independent iPSC lines. We have data within our laboratory for a capacity to generate kidney organoids, which contain glomeruli, from more than 15 distinct iPSC lines. We believe this represents sufficient evidence of the utility of the approach and would argue that this provides better

evidence of a capacity to characterise (gene and protein) podocytes and model a glomerular disease using an iPSC line than has been provided in the literature previously. Again we would note the transcriptional profiling data from Kim et al (Stem Cells, 2017), which reportedly modelled a

podocytopathy, does not profile those podocytes but provides data only from the undifferentiated iPSC line (see Supplementary Figure 5).

Reviewer #4: According to the following data from Kim et al (Stem Cells, 2017), the transcriptional profiling data is supposed to be generated from kidney organoids but not from undifferentiated iPSC line (I believe they used hESCs but not iPSCs). In any case, to compare gene expression profiles obtained from different sources (isolated glomeruli v.s. whole organoids) is not appropriate. Thus, supplementary Figure 3 should be removed.

<https://www.ncbi.nlm.nih.gov/geo/query/acc.cgi?acc=GSE103547>

Response to reviewer #4. We were asked to compare our data to this particular existing dataset which comes from a manuscript entitled ‘Gene-edited kidney organoids reveal mechanisms of disease in podocyte development’. We note that this manuscript was published after the first submission of our own data in mid-2017. The Kim et al data is RNAseq data performed in a way that is readily comparable with our own in a manuscript that states that it reports ‘*transcriptomic analysis of podocalyxin-knockout hPSCs and derived podocytes*’ within the abstract. While it is evident at comparison that podocyte identity is not as good as our own, we feel justified to continue to include this data as the only existing data available of relevance. We believe this clearly highlights the appropriateness of our approach. We note that the reviewer points us to the GSE file for the data within that manuscript (GSE103547). While that entry does state that the material was derived from organoids, as we had initially assumed, the manuscript states "As hPSC-podocytes are terminally differentiated cells that cannot be readily purified or expanded, these experiments were performed on undifferentiated hPSCs, which strongly express podocalyxin". To clarify what the data really does represent, we have now included RNAseq data from total organoids and from undifferentiated iPSC and note that the data is most likely undifferentiated iPSC. We have modified Supplementary Figure 3 to show the comparison and then show a PCA clarifying why the Kim et al data is not matching podocyte cell lines or our own data. It is important for the field to be aware of this.

Reviewer #4. There are other concerns such as glomeruli number. Authors say >3000 OrgGloms can be isolated from one experiment, but there is no actual data provided in the manuscript. How did they count the glomeruli number? Did they isolate kidney organoid area apparently by bright field? Did they modify differentiation protocols to the 15 distinct iPSC lines? Quality control of organoid generation is not also discussed. How do they choose successful differentiation samples?

Response to reviewer #4: Our inclusion of a comment about the number of organoids that can be generated was requested in prior reviews. The number of OrgGloms per organoid in relation to this is described in brief in the introduction, whilst we have also previously published that an organoid generated with a starting cell number of 5×10^5 cells contains approximately 100 glomeruli (Takasato et al, 2015). In any given differentiation of cells plated in a single well of a 6-well plate, we routinely generate approximately 30 organoids. This is the lower end of the spectrum and can be scaled up with ease by simply differentiating more cells on a larger surface area (e.g. multiple wells of a 6-well plate, T25 flask etc). As such differentiation of a larger number of cells can be performed with ease in order to generate a larger number of organoids if desired. Consequently, the number of organoids generated per experiment depends directly on the experimental plan for which they are being made in order to answer specific questions. If we need more, we make more. The organoids themselves can be generated with a smaller starting cell number if a smaller organoid is desired, and as such will have a smaller number of glomeruli per organoid. The full details of how each newly derived line is optimised for differentiation is described in Takasato et al, Nature Protocols, 2016. In brief, we optimise the initial duration of

CHIR induction and evaluate outcome based upon four colour immunofluorescence at Day 7 + 18 to look for evidence of glomeruli, proximal and distal nephron segments. This information is all in the public domain. To make the manuscript less definitive, we have softened the emphasis on numbers.

REVIEWERS' COMMENTS:

Reviewer #4 (Remarks to the Author):

Please see the attached word file because I included images.

Authors addressed some of concerns from reviewer #3. However, I'm afraid that there are still potential technical flaws which were pointed out by reviewer #3. As I was asked to evaluate whether authors addressed concerns from reviewer #3, I'm sorry that I have to say the answer is "No". However, because now authors provided additional data, personally I think the manuscript can be revised to the acceptable form for publication in Nature Communications, if the following remaining issues are resolved. There are many ways to address these issues, but I'll also suggest easy ways to resolve these technical flaws.

1. As the reviewer #3 raised a concern in Figure 5, due to the potential technical flaw, the authors' conclusion can be incorrect. According to the authors' comment, it was technically not possible to isolate OrgGloms from d7+10, d7+14, and d7+18/19 by the same method. So, authors isolated MAFB+ cells from d7+10 by flow sorting alone, MAFB+ cells and OrgGloms from d7+14 by both flow sorting and sieving, and OrgGloms from d7+19 by sieving alone. So, the potential technical flaw is that OrgGloms isolated by sieving contain not only MAFB+ podocytes but also PDGFRb+ mesangial cells and PECAM1+ endothelial cells according to Supplemental Figure 4E. Isolation of MAFB+ cells by flow sorting presumably provides a pure population of podocytes according to the citation 45 and 46. Thus, mRNAs from d7+10 samples represent a pure podocyte gene expression while mRNAs from d7+18 contain mixed populations including at least three different cell types.

Thus, the following sentence should be revised as follows.

In page 11, "Isolation of d7+18 OrgGloms from live organoids by sieving yielded a pure mTagBFP2+ cell population (Figure 5C, 0hr)". According to Supplementary Figure 4E, d7+18 OrgGloms isolated by sieving contain mesangial and endothelial cells, meaning d7+18 OrgGloms contain mixed populations but not a pure cell population. Hence, I suggest replacing "pure" with "enriched".

In page 12, "To capture all MAFB+ cells, both FACS sorting and sieving was performed at all timepoints and all component cells combined for RNAseq profiling". This sentence is incorrect according to the authors' following comment in the response letter.

Hence, at different timepoints, different approaches were used to capture all MAFB-BFP cells present. The samples at d7+10 were isolated by FACS by MAFB-reporter fluorescence as shown in Figure 5B. Samples at d7+14 were also isolated by a combination of FACS (immature cells) and sieving (increasingly mature OrgGloms) in order to maximise the

podocyte population. D7+19 OrgGloms were isolated by sieving alone.

I suggest revising this sentence to “To capture an enriched podocyte population, the MAFB+ cells were isolated at d7+10 by FACS sorting, MAFB+ cells and OrgGloms were isolated at d7+14 by FACS sorting and sieving, and OrgGloms were isolated at d7+19 by sieving.”.

In pages 12-13, “This suggests increasing cellular complexity and individual cellular maturity occurs within OrgGloms present in kidney organoids over time”. This conclusion may be due to the biased samples because samples at different stages of differentiation were obtained by different methods. This is the concern from the reviewer #3, and I think the revised manuscript did not address this issue yet. Ideally, as the reviewer #3 suggested, authors should more deeply characterize cells within OrgGloms. However, I think authors can simply explain this potential technical flaw and that can be enough. If authors agree to do so, I also suggest revising the labels, “d7+10”, “d7+14”, and “d7+19” in Figure 5.

2. As the reviewer #3 raised a concern in Figure 4F, there is a potential technical flaw which can mislead readers into an inaccurate conclusion. The comment from the reviewer #3 is as follows.

Fig 4F only shows the presence or absence of each ECM and does not show the relative expression levels, and even in this setting, the OrgGloms look different from those in vivo, including in the expression of COL4A4 (add COL4A4 data in Fig. 4D and F). The new phrase added to the abstract “comparable with human glomeruli in vivo” is therefore misleading and should be deleted.

Then, authors’ reply was as follows.

It is not possible to present what is being requested here because of the nature of the prior data that was being used for comparison. The proteomic analysis in the previously published articles looking at human glomeruli and immortalised lines in culture was performed using spectral counting. The proteomics performed on the OrgGlom and OrgPT samples was analysed using the more advanced method of ion intensity. Consequently it was not possible to directly compare the precise protein levels detected

After many revisions, authors’ conclusion was as follows in page 9.

Isolated organoid-derived glomeruli synthesise mature components of the human glomerular basement membrane.

To evaluate whether authors' reply is sufficient to respond to the concern from the reviewer #3, I tried to find the original data of Human GM, ciPod, ciGenC, and ciPod-ciGenC in Figure 4F but I could not find in my first review. The following sentence in the manuscript is explaining sources of those data in pages 10-11.

To achieve a wider perspective, the OrgGlom matrix data was compared to previously published proteomic data from human glomerular tissue alongside the matrices of the conditionally immortalised human podocytes (ciPod) [15] and conditionally immortalised Glomerular endothelial cells (ciGenC) [44] (Figure 4F).

The reference 15 is essentially explaining the marker expression of podocytes in a conditionally immortalized human podocyte cell line. There is no data of matrix in this publication. The reference 44 showed characterization of ciGenC. There is no reference for the data set in the method section. Then, I finally found DOI in the figure 4 legend... Authors need to add this citation and the original dataset identifier (PXD000456) to the main text and should not use DOI... Reviewers are spending much time to read and evaluate papers as volunteers. Please respect their time!

So, the DOI in page 43 is (doi: 10.1681/ASN.2013030233). This is a publication in JASN in 2014 and analyzed ECM expression in human glomeruli isolated by sieving. The result of the glomerular ECM proteome is summarized in Table 1 which I copied below.

Table 1. Basement membrane proteins in the glomerular ECM proteome

Basement Membrane Proteins	Gene Name	Molecular Mass (kDa)	Abundance (nSC)	Classification
Agrin	AGRN	215	3.075	Glycoprotein
Collagen α 1(XV) chain	COL15A1	142	0.052	Collagen
Collagen α 1(XVIII) chain	COL18A1	154	5.230	Collagen
Collagen α 1(IV) chain	COL4A1	161	12.515	Collagen
Collagen α 2(IV) chain	COL4A2	168	17.709	Collagen
Collagen α 3(IV) chain	COL4A3	162	8.155	Collagen
Collagen α 4(IV) chain	COL4A4	164	8.961	Collagen
Collagen α 5(IV) chain	COL4A5	161	3.778	Collagen
Collagen α 6(IV) chain	COL4A6	164	0.589	Collagen
Fibulin-1	FBLN1	77	0.029	Glycoprotein
Fibrillin-1	FBN1	312	1.403	Glycoprotein
Fibronectin	FN1	263	1.893	Glycoprotein
ECM protein FRAS1	FRAS1	443	0.039	Glycoprotein
Hemicentin-1	HMCN1	613	0.007	Glycoprotein
Perlecan	HSPG2	467	7.012	Proteoglycan
Laminin subunit α 2	LAMA2	343	0.043	Glycoprotein
Laminin subunit α 5	LAMA5	400	9.482	Glycoprotein
Laminin subunit β 1	LAMB1	198	0.977	Glycoprotein
Laminin subunit β 2	LAMB2	196	14.745	Glycoprotein
Laminin subunit γ 1	LAMC1	178	8.818	Glycoprotein
Nidogen-1	NID1	136	10.247	Glycoprotein
Nidogen-2	NID2	151	1.249	Glycoprotein
Tubulointerstitial nephritis antigen	TINAG	55	12.873	Glycoprotein
von Willebrand factor A domain-containing protein 1	VWA1	47	1.171	Glycoprotein

The glomerular ECM proteome was further categorized according to GO annotation. Twenty-four basement membrane proteins were identified by MS. Relative protein abundance is shown as normalized spectral counts (nSCs).

The below is the result of organoid-glumeruli by an improved method in figure 4Di from the manuscript.

The absolute values cannot be compared as the authors explain, and I agree with that. The following is the summary in figure 4F made by the authors to compare these different datasets.

I'm sorry that the resolution here is not high, but this is the actual resolution in the manuscript I received for review. It is hard to read, but it appears that LAMA1 is expressed in Human GM and OrgGloms in this summary. However, in the original Table 1, LAMA1 is not detected in Human GM. So, the summary does not reflect the original dataset, which can mislead readers to an inaccurate conclusion.

Laminin types are known to change during the development of GBM. Laminin 111 which includes LAMA1 is an immature type of laminin during the GBM development (Sharmin, JASN, 2016), which will switch to the matured type, laminin 521. The table 1 in the original publication of human GM, clearly demonstrate the expression of the matured type, laminin 521, while the figure 4Di shows sustained expression of the immature type, laminin 111. As the reviewer #3 said, the types of collagen also change during the GBM maturation. Collagen4 $\alpha1/\alpha2$ is an immature type while collagen4 $\alpha4/\alpha5$ is a matured type. As the reviewer #3 said, the mature type of collagen, $\alpha4$, is not detected in OrgGloms. Authors also said as follows in page 10 of the manuscript. Thus, OrgGloms express both immature and mature types of GBM.

The mature type IV collagen $\alpha5$ and $\alpha6$ chains were also highly abundant, these are expressed in the Bowman's capsule as the $\alpha5 \alpha5 \alpha6$ network [41], however $\alpha3$ and $\alpha4$ chains were at the limit of detection, suggesting that alternative cues are required for assembly of the $\alpha3 \alpha4 \alpha5$ network (Figure 4Di, Table 1 & Supplementary Table 2).

In conclusion here, my suggestion is to remove Figure 4F and to explain that OrgGlom is still expressing immature types of GBM with transition to the mature types. Alternatively, I suggest 1. To remove the data of Human GM in Figure 4F, 2. To rearrange the order of ECM names by categorizing into immature GBM, mature GBM, and other ECMs, 3. To focus on comparison among OrgGloms and other cell lines but not human GM.

3. As the reviewer #3 raised a concern about Figure 7, the current Figure 7 lacks appropriate

control samples such as ciPod and renal tubules. The overall conclusion in this manuscript is that OrgGloms isolated from organoids are better than other current tools such as ciPod. Thus, comparison to other cells is necessary. This is a very simple experiment to strengthen their conclusion. The original reviewer #3's comment is as follows.

Regarding the toxicity experiments, the glomeruli appear to simply die after drug treatment. Are the glomeruli more sensitive to the drug than other lineages in the organoids, such as renal tubules? Again, morphological abnormalities, such as foot process effacement, should also be examined.

Even after many revisions, the current manuscript did not address this concern from the reviewer #3. My suggestion is to simply remove Figure 7 OR to add the appropriate control to demonstrate the superiority of OrgGloms over other current tools. I also suggest testing 0.01 μ M of doxorubicin to find differences of sensitivity in OrgGloms.

4. In regard to supplementary figure 3, I simply asked the corresponding author of GSE103547 if the data was obtained from organoids or undifferentiated hPSCs (I did not talk anything about your manuscript). What authors says is actually true. The data was obtained from undifferentiated cells. So, the error needs to be corrected in GSE103547. However, I personally do not think it is appropriate to criticize mistakes of other groups' work in your manuscript. Rather, I strongly encourage you to provide productive results for researchers in this stem cell field. Thus, I suggest demonstration of the utility of OrgGloms by showing comparison between whole organoids (Takasato D25) and OrgGloms in this figure. This data will support that sieving can enrich the podocyte population.
5. In regard to western blot, according to the following web site, the ProteinSimple Wes Capillary Western Blot analyser can give you a whole membrane image. If you used a different instrument, please double-check the method in the manuscript.

<http://www.aureliabio.com/specialised-technologies/wes/>

Overall, as I said in the beginning of this letter, concerns from reviewer #3 are not fully addressed in this current manuscript. I am not suggesting to reject nor to accept the paper because I was asked to evaluate whether the manuscript addressed issues raised by the reviewer #3. In addition, I spent much time to find ways to improve the manuscript for publication in Nature Communications. As I explained above, there are potential technical flaws, and there are ways to

address issues easily. I hope my comments are useful for authors and editors. Reviewers are spending much time to review papers as volunteers. I understand reviewers' comments often make authors upset, but please be respectful to reviewers. Rather than disagreeing with reviewers, just to tone down wording can sometimes be the easiest way to get accepted.

Detailed responses to reviewer 4: (responses in blue)

Point 1. Confusion around the isolation of MAFB-expressing cells from organoids for profiling.

As the reviewer #3 raised a concern in Figure 5, due to the potential technical flaw, the authors' conclusion can be incorrect. According to the authors' comment, it was technically not possible to isolate OrgGloms from d7+10, d7+14, and d7+18/19 by the same method. So, authors isolated MAFB+ cells from d7+10 by flow sorting alone, MAFB+ cells and OrgGloms from d7+14 by both flow sorting and sieving, and OrgGloms from d7+19 by sieving alone. So, the potential technical flaw is that OrgGloms isolated by sieving contain not only MAFB+ podocytes but also PDGFRb+ mesangial cells and PECAM1+ endothelial cells according to Supplemental Figure 4E. Isolation of MAFB+ cells by flow sorting presumably provides a pure population of podocytes according to the citation 45 and 46. Thus, mRNAs from d7+10 samples represent a pure podocyte gene expression while mRNAs from d7+18 contain mixed populations including at least three different cell types.

Thus, the following sentence should be revised as follows.

In page 11, "Isolation of d7+18 OrgGloms from live organoids by sieving yielded a pure mTagBFP2+ cell population (Figure 5C, 0hr)". According to Supplementary Figure 4E, d7 +18 OrgGloms isolated by sieving contain mesangial and endothelial cells, meaning d7 +18 OrgGloms contain mixed populations but not a pure cell population. Hence, I suggest replacing "pure" with "enriched".

In page 12, "To capture all MAFB+ cells, both FACS sorting and sieving was performed at all timepoints and all component cells combined for RNAseq profiling". This sentence is incorrect according to the authors' following comment in the response letter.

Hence, at different timepoints, different approaches were used to capture all MAFB-BFP cells present. The samples at d7+10 were isolated by FACS by MAFB-reporter fluorescence as shown in Figure 5B. Samples at d7+14 were also isolated by a combination of FACS

(immature cells) and sieving (increasingly mature OrgGloms) in order to maximise the podocyte population. D7+19 OrgGloms were isolated by sieving alone.

I suggest revising this sentence to "To capture an enriched podocyte population, the MAFB+ cells were isolated at d7+10 by FACS sorting, MAFB+ cells and OrgGloms were isolated at d7+14 by FACS sorting and sieving, and OrgGloms were isolated at d7+19 by sieving."

In pages 12-13, "This suggests increasing cellular complexity and individual cellular maturity occurs within OrgGloms present in kidney organoids over time". This conclusion may be due to the biased samples because samples at different stages of differentiation were obtained by different methods. This is the concern from the reviewer #3, and I think the revised manuscript did not address this issue yet. Ideally, as the reviewer #3 suggested, authors should more deeply characterize cells within OrgGloms. However, I think authors can simply explain this potential technical flaw and that can be enough. If authors agree to do so, I also suggest revising the labels, "d7+10", "d7+14", and "d7+19" in Figure 5.

Response: There are no technical flaws in what we present or how we interpret this data. We completely describe what is being profiled in this analysis of the temporal expression of MAFB-expressing cells isolated from organoids. A kidney organoid is a model of a developing kidney. The nephrons in a kidney form via a mesenchyme to epithelial transition with the early nephrons then undergoing patterning and segmentation to form a glomerulus at one end and the distal tubule at the other. MAFB expression is known to commence, albeit at lower levels, in the proximal end of the forming nephron then increases as podocyte patterning and maturation occurs. Hence, the purpose of this entire section is to transcriptionally analyse the process of maturation of the proximal nephron across time. Hence, what we show is a timecourse of blue cells. OrgGloms do not form until after 7+14. The only way to capture all blue cells is to combine what is sieved and what is FACS sorted. We have again edited our results and methods sections to ensure this is completely clear. We have also altered the text introducing this section to make it more clear what MAFB expression identifies across time. This is important data as it represents a benchmark of gene expression against which others using iPSC-derived models of podocytes can compare their own data. We do not believe it is in any way inappropriate to be described or labelled in the way that we have done so. We note that this temporal RNAseq data has now been submitted to GEO and an accession number included in the manuscript.

Point 2. Issues with proteomic data

As the reviewer #3 raised a concern in **Figure 4F**, there is a potential technical flaw which can mislead readers into an inaccurate conclusion. The comment from the reviewer #3 is as follows.

Fig 4F only shows the presence or absence of each ECM and does not show the relative expression levels, and even in this setting, the OrgGloms look different from those in vivo, including in the expression of COL4A4 (add COL4A4 data in Fig. 4D and F). The new phrase added to the abstract “comparable with human glomeruli in vivo” is therefore misleading and should be deleted.

Then, authors’ reply was as follows.

It is not possible to present what is being requested here because of the nature of the prior data that was being used for comparison. The proteomic analysis in the previously published articles looking at human glomeruli and immortalised lines in culture was performed using spectral counting. The proteomics performed on the OrgGlom and OrgPT samples was analysed using the more advanced method of ion intensity. Consequently it was not possible to directly compare the precise protein levels detected

After many revisions, authors’ conclusion was as follows in page 9.

Isolated organoid-derived glomeruli synthesise mature components of the human glomerular basement membrane.

This is an accurate statement. It does not claim that all known components of the mature GBM are synthesised. However, it could be modified to state ‘some of the mature components’.

Response: The reviewer agrees that our statement is accurate. Nevertheless, we have modified to text to the requested wording.

To evaluate whether authors' reply is sufficient to respond to the concern from the reviewer #3, I tried to find the original data of Human GM, ciPod, ciGEnC, and ciPod-ciGEnC in Figure 4F but I could not find in my first review.

To achieve a wider perspective, the OrgGlom matrix data was compared to previously published proteomic data from human glomerular tissue alongside the matrices of the conditionally immortalised human podocytes (ciPod) [15] and conditionally immortalised Glomerular endothelial cells (ciGEnC) [44] (Figure 4F).

The reference 15 is essentially explaining the marker expression of podocytes in a conditionally immortalized human podocyte cell line. There is no data of matrix in this publication. The reference 44 showed characterization of ciGEnC. There is no reference for the data set in the method section. Then, I finally found DOI in the figure 4 legend... Authors need to add this citation and the original dataset identifier (PXD000456) to the main text and should not use DOI... Reviewers are spending much time to read and evaluate papers as volunteers. Please respect their time!

So, the DOI in page 43 is (doi: 10.1681/ASN.2013030233). This is a publication in JASN in 2014 and analyzed ECM expression in human glomeruli isolated by sieving. The result of the glomerular ECM proteome is summarized in Table 1 which I copied below.

Table 1. Basement membrane proteins in the glomerular ECM proteome

Basement Membrane Proteins	Gene Name	Molecular Mass (kDa)	Abundance (nSC)	Classification
Agrin	AGRN	215	3.075	Glycoprotein
Collagen α 1(XV) chain	COL15A1	142	0.052	Collagen
Collagen α 1(XVIII) chain	COL18A1	154	5.230	Collagen
Collagen α 1(IV) chain	COL4A1	161	12.515	Collagen
Collagen α 2(IV) chain	COL4A2	168	17.709	Collagen
Collagen α 3(IV) chain	COL4A3	162	8.155	Collagen
Collagen α 4(IV) chain	COL4A4	164	8.961	Collagen
Collagen α 5(IV) chain	COL4A5	161	3.778	Collagen
Collagen α 6(IV) chain	COL4A6	164	0.589	Collagen
Fibulin-1	FBLN1	77	0.029	Glycoprotein
Fibrillin-1	FBN1	312	1.403	Glycoprotein
Fibronectin	FN1	263	1.893	Glycoprotein
ECM protein FRAS1	FRAS1	443	0.039	Glycoprotein
Hemicentin-1	HMCN1	613	0.007	Glycoprotein
Perlecan	HSPG2	467	7.012	Proteoglycan
Laminin subunit α 2	LAMA2	343	0.043	Glycoprotein
Laminin subunit α 5	LAMA5	400	9.482	Glycoprotein
Laminin subunit β 1	LAMB1	198	0.977	Glycoprotein
Laminin subunit β 2	LAMB2	196	14.745	Glycoprotein
Laminin subunit γ 1	LAMC1	178	8.818	Glycoprotein
Nidogen-1	NID1	136	10.247	Glycoprotein
Nidogen-2	NID2	151	1.249	Glycoprotein
Tubulointerstitial nephritis antigen	TINAG	55	12.873	Glycoprotein
von Willebrand factor A domain-containing protein 1	VWA1	47	1.171	Glycoprotein

The glomerular ECM proteome was further categorized according to GO annotation. Twenty-four basement membrane proteins were identified by MS. Relative protein abundance is shown as normalized spectral counts (nSCs).

The below is the result of organoid-glumeruli by an improved method in figure 4Di from the manuscript.

The absolute values cannot be compared as the authors explain, and I agree with that. The following is the summary in figure 4F made by the authors to compare these different datasets.

I'm sorry that the resolution here is not high, but this is the actual resolution in the manuscript I received for review. It is hard to read, but it appears that LAMA1 is expressed in Human GM and OrgGloms in this summary. However, in the original Table 1, LAMA1 is not detected in Human GM. So, the summary does not reflect the original dataset, which can mislead readers to an inaccurate conclusion.

Response: We accept that the references referred to here are not the most appropriate. The original data for human glomerular matrix and cell-derived ECM proteomics are deposited in the open PRIDE resource <http://www.proteomexchange.org>. References to the PRIDE database include the following 2 relevant JASN papers:

Global Analysis Reveals the Complexity of the Human Glomerular Extracellular Matrix: Rachel Lennon, Adam Byron, Jonathan D. Humphries, Michael J. Randles, Alex Carisey, Stephanie Murphy, David Knight, Paul E. Brenchley, Roy Zent and Martin J. Humphries.

JASN May 2014, 25 (5) 939-951;
 DOI: <https://doi.org/10.1681/ASN.2013030233>
 PRIDE: **PXD000456**

Glomerular Cell Cross-Talk Influences Composition and Assembly of Extracellular Matrix: Adam Byron, Michael J. Randles, Jonathan D. Humphries, Aleksandr Mironov, Hellyeh Hamidi, Shelley Harris, Peter W. Mathieson, Moin A. Saleem, Simon C. Satchell, Roy Zent, Martin J. Humphries and Rachel Lennon

JASN May 2014, 25 (5) 953-966;
 DOI: <https://doi.org/10.1681/ASN.2013070795>
 PRIDE: **PXD000643**.

The first of these is already included in the manuscript (reference 8). We have added the second and updated the statement within the manuscript to read:

To achieve a wider perspective, the OrgGlom matrix data was compared to previously published proteomic data from human glomerular tissue (Lennon et al, 2014) alongside the matrices of the conditionally immortalised human podocytes (ciPod) and conditionally immortalised Glomerular endothelial cells (ciGEnC) (Byron et al, 2014) (Figure 4F).

Laminin types are known to change during the development of GBM. Laminin 111 which includes LAMA1 is an immature type of laminin during the GBM development (Sharmin, JASN, 2016), which will switch to the matured type, laminin 521. The table 1 in the original publication of human GM, clearly demonstrate the expression of the matured type, laminin 521, while the figure 4Di shows sustained expression of the immature type, laminin 111.

As the reviewer #3 said, the types of collagen also change during the GBM maturation. Collagen4 $\alpha1/\alpha2$ is an immature type while collagen4 $\alpha4/\alpha5$ is a matured type. As the reviewer #3 said, the mature type of collagen, $\alpha4$, is not detected in OrgGloms. Authors also said as follows in page 10 of the manuscript.

Thus, OrgGloms express both immature and mature types of GBM. The mature type IV collagen $\alpha5$ and $\alpha6$ chains were also highly abundant, these are expressed in the Bowman's capsule as the $\alpha5$ $\alpha5$ $\alpha6$ network [41], however $\alpha3$ and $\alpha4$ chains were at the limit of detection, suggesting that alternative cues are required for assembly of the $\alpha3$ $\alpha4$ $\alpha5$ network (Figure 4Di, Table 1 & Supplementary Table 2).

In conclusion here, my suggestion is to remove Figure 4F and to explain that OrgGlom is still expressing immature types of GBM with transition to the mature types. Alternatively, I suggest 1. To remove the data of Human GM in Figure 4F, 2. To rearrange the order of ECM names by categorizing into immature GBM, mature GBM, and other ECMs, 3. To focus on comparison among OrgGloms and other cell lines but not human GM.

Response: There is a single error with respect to the laminin isoforms in Figure 4F. We did not detect LAMA1 in the human glomeruli but rather LAMA2. LAMA1 but not LAMA2 was present in the organoid proteomic data.

So, the overlap in laminin chains between human gloms and organoids is as follows:
LAMA5, LAMB1, LAMB2, LAMC1

In addition, the organoids have: LAMA1, LAMA3, LAMB4

For the immortalised cells:

Endothelial cells: LAMA3, LAMA4, LAMA5, LAMB1, LAMB2, LAMB3, LAMC1, LAMC2

Podocytes: LAMA3, LAMA5, LAMB1, LAMB2, LAMB3, LAMC1, LAMC2

We haven't included all of the laminin chains expressed in the immortalised cells, as the focus is on the organoid matrix.

The overlap for all 4 is: LAMA5, LAMB1, LAMB2, LAMC1
In addition, the organoids and immortalised cells have LAMA3

Hence, we have revised / corrected Figure 4F and updated the accompanying description in the manuscript to highlight the mixed nature of the laminins being expressed (both immature and mature) in organoids, as might be expected in a developing tissue.

With respect to collagens, our representation of the collagen IV data is accurate. Our interpretation is that there is limited type IV collagen alpha 3,4,5 (we do not detect alpha 4) but we do see the alpha 1,1,2 and alpha 5,5,6 isoforms. We have noted in the manuscript that there is greater expression of collagen IV alpha 5,5,6 expression in organoids versus the immortalised cells (alpha 6 not detected).

Point 3. Proof of concept for high throughput screening of toxicity

As the reviewer #3 raised a concern about Figure 7, the current Figure 7 lacks appropriate control samples such as ciPod and renal tubules. The overall conclusion in this manuscript is that OrgGloms isolated from organoids are better than other current tools such as ciPod. Thus, comparison to other cells is necessary. This is a very simple experiment to strengthen their conclusion. The original reviewer #3's comment is as follows.

Regarding the toxicity experiments, the glomeruli appear to simply die after drug treatment. Are the glomeruli more sensitive to the drug than other lineages in the organoids, such as renal tubules? Again, morphological abnormalities, such as foot process effacement, should also be examined.

Even after many revisions, the current manuscript did not address this concern from the reviewer #3. My suggestion is to simply remove Figure 7 OR to add the appropriate control to demonstrate the superiority of OrgGloms over other current tools. I also suggest testing 0.01 μ M of doxorubicin to find differences of sensitivity in OrgGloms.

Response: We disagree with the argument made by reviewer 4 that we cannot present this data without also doing a toxicity scan of ciPod or renal tubules with the same compound. This issue here is not to compare the dose of doxorubicin likely to cause death in these other segments, but to point out that we can detect a readout in a single isolated 3D OrgGloM and hence could perform a screen of many OrgGloms rapidly, even using patient derived material. This would never be possible using ciPod or renal tubule lines. We reject the request to remove this data.

Point 4. Identity of the material profiled in Kim et al, Stem Cells.

In regard to supplementary figure 3, I simply asked the corresponding author of GSE103547 if the data was obtained from organoids or undifferentiated hPSCs (I did not talk anything about your manuscript). What authors says is actually true. The data was obtained from undifferentiated cells. So, the error needs to be corrected in GSE103547. However, I personally do not think it is appropriate to criticize mistakes of other groups' work in your manuscript. Rather, I strongly encourage you to provide productive results for researchers in this stem cell field. Thus, I suggest demonstration of the utility of OrgGloms

by showing comparison between whole organoids (Takasato D25) and OrgGloms in this figure. This data will support that sieving can enrich the podocyte population.

Response: Once again, we note that the reason we compared our data with that from Kim et al, Stem Cells, 2017, was because we were asked to. Previous reviewers argued that what we were presenting was not novel and had been presented previously in papers such as this one. When we first made this comparison, we were not aware that the Kim et al data was not podocytes. We were simply making a requested comparison. We then identified this as being from undifferentiated iPSC after careful rereading of the manuscript. We modified the manuscript to make this point. We were then told by reviewer 4 that we were incorrect because of the metadata in the GSE data referred to in that manuscript. We then illustrated that we were not incorrect, but the GSE metadata is likely incorrect. Again we reiterate that we were asked to make this comparison.

Reviewer 4 now suggests that the GSE should be corrected. It is not possible to edit the metadata of a GEO accession code unless you are the provider of that data. However, it is not the metadata but the reviewer who is incorrect. If you go to the GEO database and view that submission (GSE103547), it clearly states under Overall Design that the data represents:

‘6 biological sample were isolated from wild type and PODXL mutant human pluripotent stem cells (12 samples in total). RNA-seq was performed on each sample.’

Hence, the reviewer is referring to the title of the GEO submission, which is the title of the manuscript. This has been the problem all along as the reader is lead to believe, based on the abstract and the title, that the data in Kim et al represents organoid-derived podocytes when it does not.

We have modified the text in our manuscript to note that there is no comparable RNAseq data available from iPSC-derived podocytes given that the profiling presented in Kim et al was performed on undifferentiated iPSC. We refer here to Supplementary Figure 3.

Reviewer 4 also argues that we should remove the comparison as it places the prior publication in a poor light. The reviewer also argues that we should make constructive comments that will assist the field. My opinion is that it is important and constructive/productive to make it clear that profiling iPSC is not a valid approach for modelling podocyte disease.

We have two options and I will agree with the decision of Nature Communications. We can remove all panels except the PCA analysis that shows that the Kim et al data clusters with iPSC

OR we can also include a second panel comparing OrgGlom and the Kim et al data to make the point that, other than PODXL which is expressed in undifferentiated iPSC, this approach would not be useful for disease modelling.

This would look as follows:

Our revised version includes the latter. Please advise what is preferred.

Point 5. Western blots

In regard to Western blot, according to the following web site, the ProteinSimple Wes Capillary Western Blot analyser can give you a whole membrane image. If you used a different instrument, please double-check the method in the manuscript.

<http://www.aureliabio.com/specialised-technologies/wes/>

Response: We have provided the intensity data (raw and formatted) for all samples run on the Western from which the data in Figure 5 was generated within the Source Data file. Note that other lanes included undifferentiated iPSC which show no evidence of NEPHRIN protein.